# Fairness Overfitting in Machine Learning: An Information-Theoretic Perspective

**Firas Laakom** [1]  **Haobo Chen** [2]  **Jürgen Schmidhuber** [1,3]  **Yuheng Bu** [2]

## Abstract

Despite substantial progress in promoting fairness in high-stake applications using machine learning models, existing methods often modify the training process, such as through regularizers or other interventions, but lack formal guarantees that fairness achieved during training will generalize to unseen data. Although overfitting with respect to prediction performance has been extensively studied, overfitting in terms of fairness loss has received far less attention. This paper proposes a theoretical framework for analyzing fairness generalization error through an information-theoretic lens. Our novel bounding technique is based on Efron–Stein inequality, which allows us to derive tight information-theoretic fairness generalization bounds with both Mutual Information (MI) and Conditional Mutual Information (CMI). Our empirical results validate the tightness and practical relevance of these bounds across diverse fairness-aware learning algorithms. Our framework offers valuable insights to guide the design of algorithms improving fairness generalization.

## 1. Introduction

As machine learning advances, its deployment in high-stakes applications, such as hiring, financial lending, and criminal justice, has raised critical fairness concerns (Mehrabi et al., 2021; Li & Liu, 2022; Barocas et al., 2019). Deep learning models often inherit biases from the data they are trained on, potentially leading to inequitable outcomes for certain groups (Pessach & Shmueli, 2022; Ruggieri et al., 2023). Addressing these issues is particularly challenging because neural networks rely on high-dimensional, complex representations, which can obscure underlying biases

(Zemel et al., 2013). To mitigate these risks, various fairness interventions (Barocas et al., 2019; Zafar et al., 2017; Lee et al., 2021), particularly regularization-based in-processing methods (Kamishima et al., 2012; Baharlouei et al., 2020; Mroueh et al., 2021; Li et al., 2022; Lee et al., 2022; Alghamdi et al., 2022; Shui et al., 2022), have been proposed to ensure equitable predictions without significantly compromising model performance.

All these in-processing approaches are built on the implicit assumption that imposing fairness constraints during training will inherently generalize and maintain these fairness standards on new, unseen data. However, neural networks are known for their strong memorization capabilities, often leading to overfitting with limited training data. This suggests that the aforementioned assumption may not hold, and neural networks could exhibit 'fairness overfitting.'

**Is there a 'fairness overfitting?'** To investigate this question, we conduct experiments on the COMPAS dataset (Larson et al., 2016). We evaluate the performance of the standard Empirical Risk Minimization (ERM) approach alongside three fairness regularization techniques. The results, presented in Figure 1, reveal that all these algorithms exhibit fairness overfitting, characterized by a noticeable fairness generalization error, particularly in the low data regime. Furthermore, the impact of different fairness techniques on fairness generalization errors varies. While some approaches slightly mitigate the generalization error, others worsen it, in certain cases doubling the generalization error compared to ERM. This highlights the need for a deeper theoretical understanding of fairness generalization, which is the main goal of this paper.

One key insight from these observations is that *neural networks exhibit fairness overfitting*, and the fairness generalization depends on both the *data* and the *algorithms*. This paper introduces a theoretical framework for understanding fairness generalization through information-theoretic tools. These tools, which account for both data- and algorithm-dependent factors (Xu & Raginsky, 2017; Harutyunyan et al., 2021; Wang & Mao, 2023) rather than relying solely on model complexity as in traditional VC-dimension analysis (Sontag et al., 1998; Harvey et al., 2017), provide a rigorous foundation for understanding and characterizing the fairness-generalization behavior of machine learning

[1]Center of Excellence for Generative AI, KAUST, Saudi Arabia [2]University of Florida, Gainesville, USA [3]The Swiss AI Lab, IDSIA, USI & SUPSI, Switzerland. Correspondence to: Firas Laakom <firas.laakom@kaust.edu.sa>, Yuheng Bu <buyuheng@ufl.edu>.

*Proceedings of the 42nd International Conference on Machine Learning*, Vancouver, Canada. PMLR 267, 2025. Copyright 2025 by the author(s).

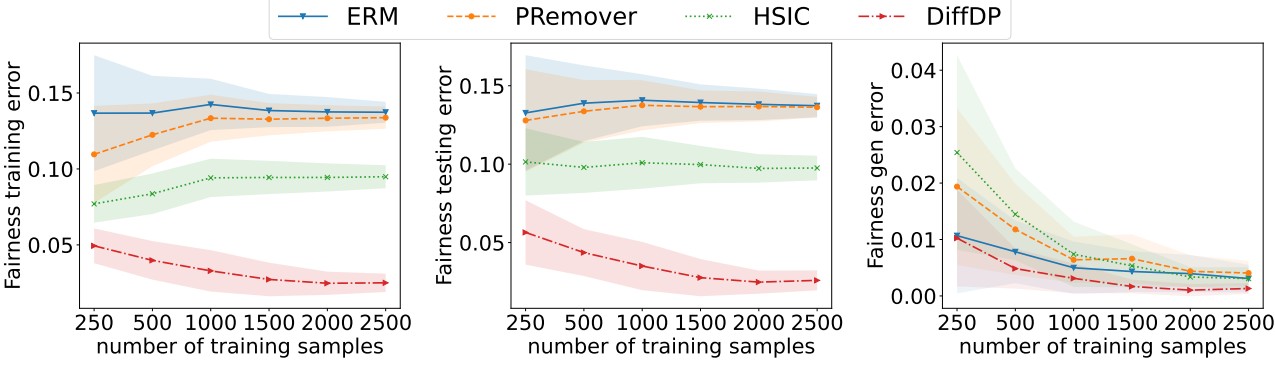

*Figure 1.* Fairness training error (left), Fairness test error (middle), and fairness generalization error (right), i.e., the difference between test fairness and training fairness error, are shown as functions of the number of training samples, using the COMPAS dataset with gender as the sensitive attribute. Experimental details are provided in Section 6.

models.

In the context of fairness generalization, prior work has derived guarantees for DP and EO within specific algorithmic frameworks and loss functions (Woodworth et al., 2017; Agarwal et al., 2018). While informative, these analyses primarily address sample complexity and do not account for broader algorithmic or data-dependent factors. More closely related is the work of Oneto et al. (2020b), which provides bounds under randomized algorithms, but their KL-based formulation is not practically computable in modern settings, e.g., deep neural network. In this paper, we aim to develop a more general and tractable framework for fairness generalization that accommodates a wider class of loss functions and learning algorithms, while capturing key data-specific and algorithmic factors. A detailed related work discussion is presented in Appendix A.

Our main contributions are as follows:

- We introduce the concept of fairness generalization to quantify and analyze how fairness properties observed during training extend to unseen test data. This concept provides a foundation for understanding the challenges posed by fairness overfitting during model training.

- We propose a novel bounding technique based on the Efron–Stein inequality, enabling the derivation of tighter information-theoretic bounds tailored for fairness generalization. Our bounds, presented in terms of Mutual Information (MI) and Conditional Mutual Information (CMI), offer rigorous insights into the different factors affecting fairness generalization.

- We conduct extensive empirical experiments on standard fairness datasets, including COMPAS and Adult. These experiments highlight the tightness of our bounds and their ability to capture the complex behavior of the fairness generalization error, providing valuable insights for future algorithm design.

**Notations:** We use upper-case letters to denote random variables, e.g., $\mathbf{Z}$, and lower-case letters to denote the realization of random variables. $\mathbb{E}_{\mathbf{Z} \sim P}$ denotes the expectation of $\mathbf{Z}$ over a distribution $P$. For a pair of random variables $\mathbf{W}$ and $\mathbf{Z}$, their joint distribution is denoted as $P_{\mathbf{W},\mathbf{Z}}$. Let $\overline{\mathbf{W}}$ be an independent copy of $\mathbf{W}$, and $\overline{\mathbf{Z}}$ be an independent copy of $\mathbf{Z}$, such that $P_{\overline{\mathbf{W}},\overline{\mathbf{Z}}} = P_{\mathbf{W}} \otimes P_{\mathbf{Z}}$. For random variables $\mathbf{X}$, $\mathbf{Y}$ and $\mathbf{Z}$, $I(\mathbf{X};\mathbf{Y}) \triangleq D(P_{\mathbf{X},\mathbf{Y}} \| P_{\mathbf{X}} \otimes P_{\mathbf{Y}})$ denotes the mutual information (MI), and $I_z(\mathbf{X};\mathbf{Y}) \triangleq D(P_{\mathbf{X},\mathbf{Y}|\mathbf{Z}=z} \| P_{\mathbf{X}|\mathbf{Z}=z} \otimes P_{\mathbf{Y}|\mathbf{Z}=z})$ denotes disintegrated conditional mutual information (CMI), and $\mathbb{E}_{\mathbf{Z}}[I_{\mathbf{Z}}(\mathbf{X};\mathbf{Y})] = I(\mathbf{X};\mathbf{Y}|\mathbf{Z})$ is the standard CMI. We will also use the notation $\mathbf{X},\mathbf{Y}|z$ to simplify $\mathbf{X},\mathbf{Y}|\mathbf{Z}=z$ when it is clear from the context.

## 2. Fairness Generalization error

### 2.1. Problem Formulation

Let $\mathcal{X}$, $\mathcal{T}$, and $\mathcal{Y}$ denote the spaces of features, sensitive attributes (e.g., race or gender), and labels, respectively, with random variables $\mathbf{X}$, $\mathbf{T}$, and $\mathbf{Y}$ taking values in these spaces. Suppose that a training set $\mathbf{S} \triangleq \{(\mathbf{X}_i, \mathbf{T}_i, \mathbf{Y}_i)\}_{i=1}^n$ contains $n$ i.i.d. samples $\mathbf{Z}_i \in \mathcal{Z}$ generated from the distribution $P_{\mathbf{Z}}$, where $\mathbf{V}_i = (\mathbf{X}_i, \mathbf{T}_i) \in \mathcal{V}$, and $\mathbf{Z}_i = (\mathbf{V}_i, \mathbf{Y}_i) = (\mathbf{X}_i, \mathbf{T}_i, \mathbf{Y}_i)$. For simplicity, we consider the case of the binary sensitive attribute, i.e., $\mathcal{T} = \{0, 1\}$. The learning algorithm $\mathcal{A}$ then takes $\mathbf{S}$ as input and produces a hypothesis $\mathbf{W} \in \mathcal{W}$, which is characterized by the conditional distribution $P_{\mathbf{W}|\mathbf{S}}$. Furthermore, given a set of indices $u = \{u_i\}_{i=1}^m \in \{1, \cdots, n\}^m$ with size $m$, let $\mathbf{Z}_u = \{\mathbf{Z}_{u_i}\}_{i=1}^m$ be the set of training samples indexed by $u$. In addition, we denote by $\mathbf{X}_u = \{(\mathbf{X}_{u_i})\}_{i=1}^m$ and $\mathbf{V}_u = \{(\mathbf{X}_{u_i}, \mathbf{T}_{u_i})\}_{i=1}^m$. Furthermore, in the remaining part of the paper, we denote by $n_0^{\mathbf{V}}$ and $n_1^{\mathbf{V}}$ the total number of samples in $\mathbf{V}$ with sensitive attribute $\mathbf{T}_i = 0$ and $\mathbf{T}_i = 1$, respectively.

Here, we illustrate the definition of fairness generalization using the widely recognized fairness metric "Demographic Parity" (DP) (Barocas et al., 2019), which aims to ensure that a machine learning model's predictions are independent of the sensitive group. In Section 5, we show how to extend our analysis to other metrics, e.g., equalized odds. Formally, demographic parity is satisfied when the probability of a specific prediction (say $\hat{y} = f(w, x)$) is invariant to the sensitive attribute, i.e., $\hat{\mathbf{Y}} \perp \mathbf{T}$. To quantify this, we define the fairness-empirical risk for DP as follows:

$$\ell_E^F(w, S) \triangleq \left| \frac{1}{n_0 + 2} \sum_{T_i = 0} f(w, x_i) \right.$$
$$\left. - \frac{1}{n_1 + 2} \sum_{T_i = 1} f(w, x_i) \right|, \quad (1)$$

where $n_0$ and $n_1$ denote the number of samples in $S$ with sensitive attribute 0 and 1, respectively. Given a model $w$ and a dataset $S$, $\ell_E^F(w, S)$ quantifies the discrepancy in predictions across sensitive groups.

*Remark* 1. In (1), we normalize using $n_t + 2$ instead of directly using $n_t$ to prevent extreme cases of division by zero when $n_0 = 0$ or $n_1 = 0$.

The fairness-population risk, which captures the expected discrepancy in model predictions across sensitive groups, averaged over the data-generating distribution $P_\mathbf{S}$, is

$$\ell_P^F(w, P_\mathbf{S}) \triangleq \mathbb{E}_{P_\mathbf{S}}[\ell_E^F(w, \mathbf{S})]. \quad (2)$$

Thus, the expected fairness generalization error, which quantifies the degree of fairness over-fitting can be defined as:

**Definition 1.** The fairness generalization error is

$$\overline{gen}_{fairness} \triangleq \mathbb{E}_{P_{\mathbf{W}, \mathbf{S}}}[\ell_P^F(\mathbf{W}, P_\mathbf{S}) - \ell_E^F(\mathbf{W}, \mathbf{S})], \quad (3)$$

where $P_{\mathbf{W}, \mathbf{S}}$ is induced by the learning algorithm $P_{\mathbf{W}|\mathbf{S}}$ and data generating distribution $P_\mathbf{S}$.

The generalization error, defined in Definition 1, measures the discrepancy between the fairness-population risk and the fairness-empirical risk. The primary goal of this paper is to understand and derive bounds for the fairness generalization error, a crucial aspect often overlooked in fairness-aware learning algorithms.

### 2.2. Key Challenges

Compared to the standard generalization error, we outline the following key challenges in analyzing the fairness generalization error:

- **Dependence on Sensitive Attributes:** The conditioning on the sensitive attribute $\mathbf{T}$ is pivotal in the fairness empirical risk, as expressed in (1). Designing bounds necessitates careful consideration of how the fairness loss function depends on the sensitive attribute, which is why we

present separate bounds for DP and equalized odds (EO) in Section 5. This dependence adds additional complexity, making the derivation of tight bounds more challenging than in standard generalization error analyses.

- **No Sample-Based Formulation:** Individual sample-based formulations (Bu et al., 2020; Wang & Mao, 2023; Laakom et al., 2024) are widely used to derive tighter bounds in information-theoretic generalization error analyses, leveraging the fact that the standard empirical risk is an average of the loss function over individual training samples. However, it is not the case for fairness generalization error, as the fairness loss $\ell_E^F(\mathbf{W}, \mathbf{S})$ is inherently group-dependent, requiring multiple samples for computation and cannot be reduced to individual sample-based terms. Consequently, existing bounding techniques (Bu et al., 2020; Wang & Mao, 2023; Laakom et al., 2024) are not directly applicable in this context.

- **Sub-Gaussian/boundedness Assumptions:** In standard empirical risk, the loss function is averaged over i.i.d. samples, and generalization bounds are commonly derived using sub-Gaussian assumptions on $f(w, \mathbf{X})$ to establish a $1/\sqrt{n}$ convergence rate (Xu & Raginsky, 2017; Steinke & Zakynthinou, 2020). However, as $\ell_P^F(\mathbf{W}, P_\mathbf{S})$ is not a simple average of $f(w, x_i)$, sub-Gaussian assumptions or even boundedness leads to loose bounds in our case.

## 3. General Methodology

The idea of deriving information-theoretic bounds using a subset of data (of size $m$) rather than the full dataset was first introduced by Harutyunyan et al. (2021), which also demonstrated that individual-based bounds ($m = 1$) provide the tightest analysis for standard generalization. Building upon this idea and to address group-based losses, Dong et al. (2024) proposed a bounding technique that decomposes the generalization error by leveraging different permutations of a subset of training data $\mathbf{Z}_u$ and rewriting the generalization error as the average over the different errors defined using these permutations. Directly leveraging this approach, Lemma 1 presents a preliminary bound for the fairness generalization error, as formalized in Definition 1:

**Lemma 1.** *Assume that* $|f| \in [0, 1]$, *then for any* $m \in \{2, \cdots, n\}$,

$$\overline{gen}_{fairness} \leq \frac{1}{|C_n^m|} \sum_{u \in C_n^m} \sqrt{2I(\mathbf{W}; \mathbf{V}_u)}, \quad (4)$$

*where* $C_n^m$ *is the set of* $m$*-combinations.*

The proof is provided in E.1. In contrast to Dong et al. (2024), where $m$ is predefined by the loss function, $m$ here is a flexible hyperparameter, similar to Harutyunyan et al. (2021), that controls the number of samples used to decompose the loss in (1) when deriving the bound.

Although Lemma 1 provides an important first step toward bounding fairness generalization error, it does not explicitly leverage the structure of fairness losses, resulting in an overly general bound but consequently too loose. This limitation can be rooted in the proof of Lemma 1, where the technique proposed by Dong et al. (2024) addresses the "No Sample-Based Formulation" challenge discussed in Section 2.2. However, its primary limitation is that it still relies on the boundedness of $\ell_P^F(\mathbf{W}, P_{\mathbf{S}})$ to bound its log-moment generating function, encountered in Donsker-Varadhan's variational representation, which yields the loose bound. Hence, while Lemma 1 offers a valuable starting point, it does not address all the challenges in Section 2.2.

In the following, we propose an alternative technique that effectively addresses the remaining challenges and derives tight bounds for the log-moment generating function. The key result, presented in Lemma 2, represents the first major contribution of this paper.

**Lemma 2.** *Let $g(\mathbf{V}) = g(\mathbf{V}_1, \mathbf{V}_2, \ldots, \mathbf{V}_m)$ be a real-valued square-integrable function of $m$ i.i.d random variables. Given fixed $v = (v_1, \ldots, v_m)$ and an index $i \in \{1, \ldots, m\}$, define $\tilde{v}^i = (\tilde{v}_1, \ldots, \tilde{v}_m)$ such that $\tilde{v}_j = v_j$ for all $j \neq i$ except on $i$, i.e., $\tilde{v}_i \neq v_i$. If $g(v)$ satisfies the following condition for a fixed $n_0^v$ (and as a consequence $n_1^v = m - n_0^v$),*

$$\sup_{v \in \mathcal{V}^m, \tilde{v}_i^i \in \mathcal{V}} |g(v) - g(\tilde{v}^i)| \leq \beta, \quad 1 \leq i \leq m, \quad (5)$$

*where $\beta$ may depend on $n_0^v$. Then, we have*

$$\mathrm{Var}(g(\mathbf{V})) \leq \frac{m}{4} \mathbb{E}[\beta^2]. \quad (6)$$

The proof of Lemma 2, provided in Appendix B.1, leverages the Efron–Stein inequality (Boucheron et al., 2013) and the law of total expectation. Lemma 2 provides a bound for the variance of any smooth function defined over independent random variables. As will be shown in the proofs of all subsequent Theorems 1-5, this result offers a flexible approach that, when combined with Hoeffding's lemma, serves as a powerful alternative to the traditional sub-gaussian/boundedness assumption. This bounded difference setup is particularly well-suited for analyzing generalization errors of loss functions beyond simple average, e.g., the fairness loss in (1).

*Remark 2.* The variational bounded difference condition introduced in (5) is similar to the condition in McDiarmid's inequality (Doob, 1940), as it captures the sensitivity of the function $g(v)$ to perturbations in its inputs. However, a notable distinction lies in the fact that in our case $\beta$ is a random variable as it depends on $n_0^{\mathbf{V}}$, as opposed to the fixed constants typically assumed in McDiarmid's inequality.

*Remark 3.* The square-integrability assumption in Lemma 2 is required for the application of the Efron-Stein inequality.

In our setting, this condition is naturally satisfied because all the random variables involved (i.e., $|f| \in [0, 1]$) are bounded. Since any bounded random variable is square-integrable, this ensures that the assumption holds in our case.

## 4. Demographic Parity

In this section, we present fairness generalization error bounds specifically tailored for the DP loss defined in (1).

### 4.1. Mutual Information-based Bounds

In order to leverage Lemma 2 in the context of Definition 1, we derive a bound on the sensitivity of $\ell_E^F(w, v_u)$ with respect to $v_u$. The main result is presented in Lemma 3.

**Lemma 3.** *Assume that $|f| \in [0, 1]$. let $g : \mathcal{V}^m \to \mathbb{R}$ be defined as $g(v) = g(v_1, \ldots, v_m) = \ell_E^F(w, v)$, where $\ell_E^F(w, v)$ is defined (1). Then, for a fixed $w \in \mathcal{W}$ and a fixed $n_0^v$ and $n_1^v$, for any $1 \leq i \leq m$, we have*

$$\sup_{v \in \mathcal{V}^m, \tilde{v}_i \in \mathcal{V}} |g(v) - g(\tilde{v}^i)| \leq \frac{1}{\mathbb{H}(n_0^v, n_1^v)}, \quad (7)$$

*where $\mathbb{H}$ denotes a shifted harmonic mean operator, i.e., $\forall \{a_i\}_{i=1}^k$, $\mathbb{H}(a_1, \cdots, a_k) = \frac{1}{\frac{1}{a_1+2}+\cdots+\frac{1}{a_k+2}}$.*

The proof is provided in Appendix C.1. Lemma 3 establishes a direct link between the sensitivity of $\ell_E^F(w, v)$ to individual perturbations in $v$ and the group sizes $n_0^v$ and $n_1^v$ for any fixed $w$. With the key component established in Lemma 3, we present the first information-theoretic bound for the fairness generalization error of DP in Definition 1. The main result is presented in Theorem 1, which provides a bound in terms of the MI between the output hypothesis $\mathbf{W}$ and a subset of the training samples $\mathbf{V}_u$.

**Theorem 1.** *Assume that $|f| \in [0, 1]$, then for any hyper-parameter $m \in \{2, \cdots, n\}$, we have*

$$\overline{gen}_{fairness} \quad (8)$$

$$\leq \frac{1}{|C_n^m|} \sum_{u \in C_n^m} \sqrt{\frac{m}{2} \mathbb{E}_{\mathbf{V}_u} \Big[ \frac{1}{\mathbb{H}(n_0^{\mathbf{V}_u}, n_1^{\mathbf{V}_u})^2} \Big] I(\mathbf{W}, \mathbf{V}_u)},$$

*where $C_n^m$ is the set of $m$-combinations.*

The proof, detailed in Appendix C.2, leverages Donsker-Varadhan's variational representation of KL divergence along with both Lemmas 2 and 3. Theorem 1 sheds light on the behavior of fairness-generalization performance, which implies that the less dependent the output hypothesis $\mathbf{W}$ is on the input samples $\mathbf{V} = (\mathbf{X}, \mathbf{T})$, the more effectively the learning algorithm generalizes.

*Remark 4.* The MI terms encountered in conventional MI-generalization error bounds, e.g., (Bu et al., 2020; Wang & Mao, 2023; Harutyunyan et al., 2021), typically involve

dependence on the label $\mathbf{Y}$, i.e., $I(\mathbf{W}, \mathbf{Z})$. However, an intriguing aspect of Theorem 1 is that the bound is label-independent, with the MI terms involving explicitly only $\mathbf{V} = (\mathbf{X}, \mathbf{T})$. This observation underscores the distinct nature of the DP fairness generalization error.

*Remark 5.* The term $\frac{1}{\mathbb{H}(n_0^{\mathbf{V}_u}, n_1^{\mathbf{Z}_u})} = \frac{1}{n_0^{\mathbf{Z}_u}+2} + \frac{1}{n_1^{\mathbf{Z}_u}+2}$ quantifies the impact of group imbalances among different sensitive attributes, encapsulating how the relative sizes of the sensitive groups influence the fairness generalization error. Notably, this term is minimized when $n_0^{\mathbf{Z}_u} = n_1^{\mathbf{Z}_u}$ for a fixed sample size. To the best of our knowledge, Theorem 1 provides the first fairness-generalization bound explicitly linking this error to group imbalance, offering new insights into this critical factor.

*Remark 6.* Unlike Dong et al. (2024), where the value of $m$ is determined by the specific loss function, Theorem 5 introduces $m > 1$ as a hyperparameter of the bound, independent of the learning algorithm and the form of the loss function. This parameter governs both the size of the subset $u$ and the number of terms considered in the bound's summation. Notably, setting $m = n$ (or $m = n - 1$) simplifies the bound to one term (or $m$ terms) that scales as $1/\sqrt{n}$ and is easy to estimate. In contrast, choosing $m < n - 1$ reduces $I(\mathbf{W}, \mathbf{V}_u)$ by incorporating more terms, albeit at the cost of increased estimation complexity.

### 4.2. Conditional Mutual Information-based Bounds

One drawback of the proposed bound in Theorem 1 is that it can be vacuous and challenging to compute in practice, due to its dependence on the potentially high-dimensional model weights $\mathbf{W}$. To address this issue, the conditional mutual information (CMI) framework, introduced by Steinke & Zakynthinou (2020), has been demonstrated in recent studies (Hellström & Durisi, 2022; Wang & Mao, 2023; Laakom et al., 2024) to yield practical generalization error bounds, even when $\mathbf{W}$ are high-dimensional and continuous.

In this section, we extend our fairness-generalization analysis using CMI with the super-sample framework. In particular, we assume that there are $n$ pairs of super-samples $\mathbf{Z}_{[2n]} = (\mathbf{Z}_1^{0,1}, \cdots, \mathbf{Z}_n^{0,1}) \in \mathcal{Z}^{2n}$ i.i.d generated from $P_{\mathbf{Z}}$. The training data $\mathbf{S} = (\mathbf{Z}_1^{\mathbf{R}_1}, \mathbf{Z}_2^{\mathbf{R}_2}, \cdots, \mathbf{Z}_n^{\mathbf{R}_n})$ are selected from $\mathbf{Z}_{[2n]}$, where $\mathbf{R} = (\mathbf{R}_1, \cdots, \mathbf{R}_n) \in \{0,1\}^n$ is the selection vector composed of $n$ independent uniform random variables. Intuitively, $\mathbf{R}_i$ selects sample $\mathbf{Z}_i^{\mathbf{R}_i}$ from the super-sample $\mathbf{Z}_i^{0,1}$ to be used in training, and the remaining $\mathbf{Z}_i^{\overline{\mathbf{R}_i}}$ is for the test. Let $\overline{\mathbf{S}} = (\mathbf{Z}_1^{\overline{\mathbf{R}_1}}, \mathbf{Z}_2^{\overline{\mathbf{R}_2}}, \cdots, \mathbf{Z}_n^{\overline{\mathbf{R}_n}})$. Therefore, analogous to Definition 1, we have a similar definition of fairness generalization error in the super-sample setting.

**Definition 2.** The fairness generalization error in the super-sample framework is:

$$\overline{gen}_{fairness} \triangleq \mathbb{E}_{P_{\mathbf{W}, \mathbf{Z}_{[2n]}, \mathbf{R}}}[\ell_E^F(\mathbf{W}, \mathbf{S}) - \ell_E^F(\mathbf{W}, \overline{\mathbf{S}})]. \quad (9)$$

**Relaxing the sub-Gaussianity.** Similar to the MI setting, we aim to provide an alternative to the sub-Gaussian condition for the fairness generalization error in the CMI framework. Lemma 4 establishes an inequality that bounds the log-moment generating function of $\ell_E^F(w, \mathbf{S}_u) - \ell_E^F(w, \overline{\mathbf{S}}_u)$ in terms of the structure of the data.

**Lemma 4.** $\forall \lambda \in \mathbb{R}$, and $|f| \in [0,1]$, we have

$$\log \left( \mathbb{E}_{\overline{\mathbf{R}}_u | \mathbf{Z}_{[2n]} = z_{[2n]}, \overline{\mathbf{W}} = w} \left[ e^{\lambda(\ell_E^F(w, \mathbf{S}_u) - \ell_E^F(w, \overline{\mathbf{S}}_u))} \right] \right)$$
$$\leq \frac{\lambda^2 m}{8} \mathbb{E}\left[ \frac{1}{\mathbb{H}(n_0^{S_u}, n_1^{S_u}, n_0^{\overline{S}_u}, n_1^{\overline{S}_u})^2} \right]. \quad (10)$$

The detailed proof is available in Appendix C.3 and is based on Lemmas 2 and 3 along with Hoeffding's lemma. Lemma 4 is a fundamental element in all the subsequent proofs, as it provides an alternative approach to tightly bound the log-moment generating function typically encountered in Donsker-Varadhan's variational representation.

Theorem 2 presents a bound for the fairness generalization error defined in Definition 2 using the disintegrated CMI between $\mathbf{W}$ and the selection variable $\mathbf{R}_u$ conditioned on super-sample $z_{[2n]}$.

**Theorem 2** (CMI bound). *Assume that $|f| \in [0,1]$, then for any $m \in \{2, \cdots, n\}$, we have*

$$\overline{gen}_{fairness} \leq \frac{1}{|C_n^m|} \mathbb{E}_{\mathbf{Z}_{[2n]}} \Big[ \sum_{u \in C_n^m} \quad (11)$$
$$\sqrt{\frac{m}{2} \mathbb{E}_{\mathbf{R}_u} \Big[ \frac{1}{\mathbb{H}(n_0^{S_u}, n_1^{S_u}, n_0^{\overline{S}_u}, n_1^{\overline{S}_u})^2} \Big] I_{z_{[2n]}}(\mathbf{W}, \mathbf{R}_u)} \Big],$$

*where $C_n^m$ is the set of $m$-combinations.*

The full proof is provided in Appendix C.4. The bound in Theorem 2 has an explicit dependency on the weights $\mathbf{W}$. This dependency highlights that fairness overfitting is influenced by the extent to which the random selection process reveals information about the model's weights.

One approach to tightening the bound in Theorem 2 is to leverage the model's predictions rather than its weights, as the $f$-CMI bound proposed by Harutyunyan et al. (2021). In this context, instead of directly using $I_{\mathbf{Z}_{[2n]}}(\mathbf{W}; \mathbf{R}_u)$, we consider the predictions $\mathbf{F}_u = (f(\mathbf{W}, \mathbf{X}_u^0), f(\mathbf{W}, \mathbf{X}_u^1))$.

**Theorem 3** ($f$-CMI bound). *Assume that $|f| \in [0,1]$, then for any $m \in \{2, \cdots, n\}$, we have*

$$\overline{gen}_{fairness} \leq \frac{1}{|C_n^m|} \mathbb{E}_{\mathbf{Z}_{[2n]}} \Big[ \sum_{u \in C_n^m} \quad (12)$$
$$\sqrt{\frac{m}{2} \mathbb{E}_{\mathbf{R}_u} \Big[ \frac{1}{\mathbb{H}(n_0^{S_u}, n_1^{S_u}, n_0^{\overline{S}_u}, n_1^{\overline{S}_u})^2} \Big] I_{z_{[2n]}}(\mathbf{F}_u; \mathbf{R}_u)} \Big],$$

*where $C_n^m$ is the set of $m$-combinations.*

To achieve even tighter bounds, inspired by Hellström & Durisi (2022), we can incorporate the the fairness loss pairs $\mathbf{L}_u = (\ell_E^F(\mathbf{W}, \mathbf{V}_u^0), \ell_E^F(\mathbf{W}, \mathbf{V}_u^1))$. This leads to the following main result established in Theorem 4.

**Theorem 4** (e-CMI bound). *Assume that $|f| \in [0,1]$, then for any $m \in \{2, \cdots, n\}$, we have*

$$\overline{gen}_{fairness} \leq \frac{1}{|C_n^m|} \mathbb{E}_{\mathbf{Z}_{[2n]}} \Big[ \sum_{u \in C_n^m} \tag{13}$$

$$\sqrt{\frac{m}{2} \mathbb{E}_{\mathbf{R}_u} \Big[ \frac{1}{\mathbb{H}\big(n_0^{S_u}, n_1^{S_u}, n_0^{\overline{S}_u}, n_1^{\overline{S}_u}\big)^2} \Big] I_{z_{[2n]}}(\mathbf{L}_u; \mathbf{R}_u)} \Big],$$

*where $C_n^m$ is the set of $m$-combinations.*

Given $\mathbf{Z}_{[2n]}$, the sequence $\mathbf{R}_u \to \mathbf{W} \to f(\mathbf{W}, \mathbf{X}_u^{0,1}) \to \mathbf{L}_u$ forms a Markov chain. Therefore, by the data processing inequality, $f$-CMI bound in Theorem 3 is tighter than CMI bound in Theorem 2, and e-CMI bound in Theorem 4 is tighter than $f$-CMI in Theorem 3.

However, the main limitation of Theorem 4 remains its computational cost. Specifically, since $\mathbf{R}_u$ is multidimensional (of size $m$) and $\mathbf{L}_u$ is both continuous and multidimensional, estimating the bound becomes challenging in practice. To address this issue, we leverage the loss-difference technique proposed in (Dong et al., 2024), which reduces the dimensionality of both terms to one.

To this end, we define $\Phi_u = \{R_{u_1} \oplus R_{u_i}\}_{i=2}^m \in \{0,1\}^{m-1}$, where $\oplus$ denotes the XOR operation. Given a binary value $b$, we define $b \otimes \Phi_u = (b, \{\Phi_{u_i} \oplus b\}_{i=1}^{m-1}) \in \{0,1\}^m$. To simplify the notation, we denote $0 \otimes \Phi_u$ and $1 \otimes \Phi_u$ as $\Phi_u^-$ and $\Phi_u^+$, respectively. $L_u^{\Phi_u} = (L^{\Phi_u^-}, L^{\Phi_u^+})$ denotes a pair of losses, while $\Delta^{\Phi_u} L_u = L_u^{\Phi_u^+} - L_u^{\Phi_u^-}$ denotes their difference. The main result is presented in Theorem 5.

**Theorem 5** ($\Delta$LCMI bound). *Assume that $|f| \in [0,1]$, then for any $m \in \{2, \cdots, n\}$, we have*

$$\overline{gen}_{fairness} \leq \frac{1}{|C_n^m|} \mathbb{E}_{\mathbf{Z}_{[2n]}} \Big[ \sum_{u \in C_n^m} \tag{14}$$

$$\sqrt{\frac{m}{2} \mathbb{E}_{\mathbf{R}_u} \Big[ \frac{1}{\mathbb{H}\big(n_0^{S_u}, n_1^{S_u}, n_0^{\overline{S}_u}, n_1^{\overline{S}_u}\big)^2} \Big] I_{z_{[2n]}}(\Delta \mathbf{L}_u^{\Phi_u}; \mathbf{R}_{u_1})} \Big],$$

*where $C_n^m$ is the set of $m$-combinations.*

Given $\mathbf{Z}_{[2n]}$, the sequence $\mathbf{R} \to (\mathbf{L}_u, \Phi_u) \to \Delta \mathbf{L}_u^{\Phi_u}$ forms a Markov chain. Using the data processing inequality and the independence of $\Phi_u$ and $\mathbf{R}_{u_1}$, we obtain

$$\begin{aligned} I_{z_{[2n]}}(\Delta \mathbf{L}_u^{\Phi_u}; \mathbf{R}_{u_1}) &\leq I_{z_{[2n]}}(\mathbf{L}_u, \Phi_u; \mathbf{R}_{u_1}) \\ &= I_{z_{[2n]}}(\mathbf{L}_u; \mathbf{R}_{u_1} | \Phi_u) \\ &= I_{z_{[2n]}}(\mathbf{L}_u; \mathbf{R}_u) - I_{z_{[2n]}}(\mathbf{L}_u; \Phi_u) \\ &\leq I_{z_{[2n]}}(\mathbf{L}_u; \mathbf{R}_u) \end{aligned} \tag{15}$$

This establishes that Theorem 5 is tighter than Theorems 3 and 4. Intuitively, the difference between two loss values, $\Delta \mathbf{L}_u^{\Phi_u}$, conveys significantly less information about the selection process $\mathbf{R}_{u_1}$ compared to the pair $\mathbf{L}_u$.

## 5. Equalized Odds

In this section, we extend our general bounding framework to address fairness under a *separation-based* notion, i.e., the Equalized Odds (EO) (Hardt et al., 2016).

### 5.1. Problem Formulation for EO

Under a *separation-based* fairness notion, the requirement is $\widehat{\mathbf{Y}} \perp \mathbf{T} | \mathbf{Y}$, indicating that the predicted label $\widehat{Y}$ is conditionally independent of the sensitive attribute $\mathbf{T}$, given the true label $\mathbf{Y}$. Extending our previous analysis of DP to this case introduces finer-grained fairness requirements by accounting for dependencies within each label class. In this section, we assume a binary label case, $\mathbf{Y} \in \{0,1\}$, which is typical in EO formulations. In addition, our hypothesis $\mathbf{W}$ produces prediction outputs $f(\mathbf{W}, \mathbf{X}) \in [0,1]$.

**Comparison with DP.** In the EO setting, we split samples not only by $\mathbf{T} \in \{0,1\}$ but also by $\mathbf{Y} \in \{0,1\}$. For $t \in \{0,1\}$, $y \in \{0,1\}$, let $n_{t,y} = \sum_{i=1}^n \mathbb{1}\{\mathbf{T}_i = t, \mathbf{Y}_i = y\}$. Then the fairness empirical risk for EO can be written as

$$\ell_E^{F_S}(w, S) = \ell_E^{F_S}(w, S | \mathbf{Y} = 0) + \ell_E^{F_S}(w, S | \mathbf{Y} = 1), \tag{16}$$

where

$$\begin{aligned} \ell_E^{F_S}(w, S | Y = y) = \Big| \frac{1}{n_{0,y} + 2} \sum_{T_i=0, Y_i=y} f(w, x_i) \\ - \frac{1}{n_{1,y} + 2} \sum_{T_i=1, Y_i=y} f(w, x_i) \Big|. \end{aligned} \tag{17}$$

The EO differs from DP in its conditional requirement: DP evaluates dependence between predictions and $\mathbf{T}$ unconditionally, while EO measures fairness by conditioning each label class $\mathbf{Y}$. This distinction refines the fairness loss to capture the influence of $\mathbf{T}$ on predictions within subpopulations defined by $\mathbf{Y}$ (a dependency reflected in the function $\ell_E^{F_S}(W, S)$, which now accounts for the *joint distribution* over both $\mathbf{T}$ and $\mathbf{Y}$). Despite these conceptual differences, the bounding methodology remains similar, utilizing Lemma 2 in the context of Definition 1 with EO loss. We maintain the same notation and subset enumeration as in the DP setting to ensure continuity in the analysis.

### 5.2. Generalization Error Bounds for EO

We will show that $\ell_E^{F_S}(\mathbf{W}, \mathbf{S})$ satisfies a similar bounded-differences condition as in the DP setting, but with subgroup sizes partitioned by both $\mathbf{T}$ and $\mathbf{Y}$.

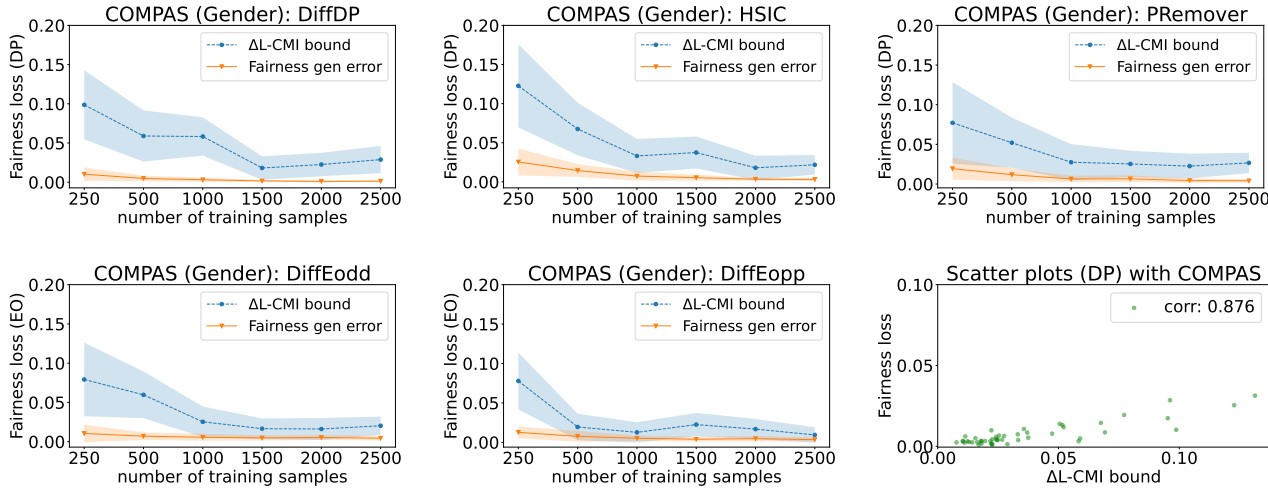

*Figure 2.* Evaluation of fairness generalization bounds on the COMPAS dataset (gender as sensitive attribute) as a function of training set size $n$. Top: DP methods and corresponding bound from Theorem 5. Bottom-left&middle: EO methods and corresponding bound from Theorem 7. Bottom-right: Scatter plot showing the correlation between our DP bound and observed fairness generalization error. The results confirm the tightness and reliability of our bounds across different methods.

**Lemma 5.** *Fix $w \in \mathcal{W}$ and a fixed $n_{0,0}^{Z_u}, n_{1,0}^{Z_u}, n_{0,1}^{Z_u}, n_{1,1}^{Z_u}$. Let $g : \mathcal{Z}^m \to \mathbb{R}$ be defined as follows: $g(z_1, \ldots, z_m) = \ell_E^{F_S}(w, z_u)$, where $\ell_E^{F_S}(w, z_u)$ is defined (16). Then, for any $1 \leq i \leq n$, we have*

$$\sup_{z_u, \tilde{z}_u^i} \left| g(z_u) - g(\tilde{z}_u^i) \right| \leq \frac{2}{\min_{(t,y)}\left(n_{t,y}^{Z_u} + 2\right)}, \quad (18)$$

*where $|f(w, x_i)|$ is bounded by 1 in the binary case, and $\min_{(t,y)}(n_{t,y} + 2)$ is the smallest subgroup size across all $t$ and $y$.*

The proof is provided in Appendix D.1. Lemma 5 employs a similar bounding methodology as Lemma 3, which extends the bounded-differences property to the EO setting. Both results are grounded in analyzing the sensitivity of the function $g(\cdot)$ with input perturbations to derive an upper bound that incorporates the number of subgroups. Notably, Lemma 2 remains applicable across different fairness measures, serving as a general tool for bounding sensitivity to individual sample perturbations, while adapting to distinct loss functions and partitioning structures. In the case of EO, the fairness risk $\ell_E^{F_S}$ is determined by the smallest $(\mathbf{T}, \mathbf{Y})$-specific subgroup. This adaptation refines the bounded-differences property to account for the joint structure of $\mathbf{T}$ and $\mathbf{Y}$.

Building on this result, we analyze the EO fairness generalization gap, leveraging both mutual information and loss-difference dependencies to quantify the impact of subgroup imbalances on fairness generalization error.

**Theorem 6.** *(EO Fairness MI Bound) Assume $|f| \in [0, 1]$.*

*For any $m \in \{2, \cdots, n\}$,*

$$\overline{gen}_{fairness} \quad (19)$$
$$\leq \frac{1}{|C_n^m|} \sum_{u \in C_n^m} \sqrt{2m\mathbb{E}_{\mathbf{Z}_u}\left[\frac{1}{\min_{(t,y)}\left(n_{t,y}^{Z_u} + 2\right)^2}\right] I(\mathbf{W}, \mathbf{Z}_u)}.$$

Compared to the DP bound in Theorem 1, the bound in Theorem 6 depends on the smallest subgroup across both $\mathbf{T}$ and $\mathbf{Y}$, while also capturing the dependence between $\mathbf{W}$ and the joint distribution of $(\mathbf{T}, \mathbf{Y})$.

**Theorem 7.** *(EO Fairness $\Delta\mathbf{L}$ Bound) Assume $|f| \in [0, 1]$. For any $m \in \{4, \cdots, n\}$, let*

$$\overline{\mathbb{H}}_{\text{CMI}} = \mathbb{H}\left(\min_{(t,y)}\left(n_{t,y}^{S_u}\right), \min_{(t,y)}\left(n_{t,y}^{\overline{S}_u}\right)\right),$$

*where $\min_{(t,y)}\left(n_{t,y}^{S_u}\right)$ and $\min_{(t,y)}\left(n_{t,y}^{\overline{S}_u}\right)$ are the smallest subgroup sizes in $t, y$ for the training set $S_u$ and test set $\overline{S}_u$, then*

$$\overline{gen}_{fairness} \leq \frac{1}{|C_n^m|} \mathbb{E}_{Z_{[2n]}}\left[\sum_{u \in C_n^m}\right. \quad (20)$$
$$\left. \sqrt{2m \, \mathbb{E}_{\mathbf{R}_u}\left[\frac{1}{\overline{\mathbb{H}}_{\text{CMI}}^2}\right] I_{z_{[2n]}}\left(\Delta\mathbf{L}_u^{\Phi_u}; \mathbf{R}_{u_1}\right)}\right],$$

*Remark 7.* Theorem 7 quantifies the dependence between $\mathbf{R}_u$ and the **EO** loss difference ($\Delta\mathbf{L}_u^{\Phi_u}$) via CMI, isolating $\mathbf{T}$'s influence on predictions *given* $\mathbf{Y}$. More details can be found in Appendix D.3.

As we show here, the proposed bound remains valid across different subgroup structures and loss formulations, highlighting the framework's wide applicability. Unlike DP,

where bounds depend solely on group sizes defined by $\mathbf{T}$, our EO analysis explicitly accounts for the joint distribution of $(\mathbf{T}, \mathbf{Y})$. As a result, the bound captures subgroup imbalances (via $\min_{(t,y)} n_{t,y}$ and $\Delta \mathbf{L}_u^{\Phi_u}$) and their impact on fairness generalization errors.

# 6. Experiments

**Empirical Setup** In this section, we empirically assess the effectiveness of our fairness generalization error bounds. Specifically, we evaluate the bounds from Theorems 4 and 7 in the context of deep neural networks. Following Han et al. (2024), we conduct experiments using on two widely studied datasets in fairness research with gender or race used as the sensitive attribute: i) The **COMPAS** (Larson et al., 2016) dataset, which involves recidivism prediction based on criminal and demographic records. ii) The **Adult** (Kohavi & Becker, 1996) dataset, derived from U.S. Census data, which focuses on income prediction.

We compare multiple fairness algorithms, including Empirical Risk Minimization (ERM), three DP in-processing approaches (Demographic Hilbert-Schmidt Independence Criterion (HSIC) (Baharlouei et al., 2020; Li et al., 2022), DiffDP (Mroueh et al., 2021), and Prejudice Remover (PRemover) (Kamishima et al., 2012)), as well as two EO methods (DiffOdd and DiffEopp (Mroueh et al., 2021)). All approaches follow the same training protocol (architectures, hyperparameters, etc.) as in Han et al. (2024).

To evaluate our bounds, we adopt the same setup as in Harutyunyan et al. (2021), reporting the mean and standard deviation over the $\mathbf{Z}_{[2n]}$ realizations. The CMI terms in Theorems 5 and 7, which involve one discrete and one continuous variable, are estimated using the method from Ross (2014). Additional details about the experimental setup are provided in Appendix F.1. Each data point in our experiments corresponds to a total of 1050 runs.

**Bound tightness:** In Figure 2, we present the fairness generalization error of models trained with different algorithms on the COMPAS dataset with gender as the sensitive attribute. The top row reports the results for different DP approaches along with our corresponding bound (Theorem 5), while the first two columns of the second row show the results for different EO approaches along with our corresponding bound (Theorem 7). As can be seen, the proposed bounds consistently capture the true fairness generalization error across the various settings. These findings validate the theoretical results and underscore the utility of our framework in providing meaningful guarantees for fairness generalization in real-world applications. The additional results on the other datasets, provided in Appendix F.2, are consistent with these findings.

**Bound-error correlation analysis:** To further analyze the relationship between our bounds and fairness generalization error, we generate scatter plots comparing the bound from Theorem 5 with the true observed DP fairness error on the COMPAS datasets. The plot, in the bottom-right corner of Figure 2, reveals that our bound exhibits a strong linear relationship with the true fairness generalization, reinforcing its reliability as an indicator of generalization performance.

**Batch Balancing:** One of the key insights of Theorem 1 is the connection between fairness generalization error and group imbalance, highlighting that balancing groups in training data can result in tighter generalization bounds. Motivated by this result, we introduce a simple batch-balancing technique, where we ensure that the different groups are proportionally balanced within each *training batch*. In particular, we sample half of the batch samples from each sensitive group (e.g., gender: Male and Female), ensuring equal representation in each batch, thereby mitigating the impact of group imbalance.

To evaluate the effectiveness of this technique, we conduct extensive experiments with various methods, measuring its impact on DP over unseen test data. The results, summarized in Table 1 for COMPAS and Table 2 (Appendix F.3) for Adult, demonstrate that our approach consistently reduces DP on test data across all methods for dataset sizes. Remarkably, in some cases, it decreases the error by a factor of 10, highlighting its impact on improving the performance of different fairness methods. These findings not only validate our theoretical results but also highlight their practical relevance in developing effective strategies for enhancing fairness in real-world applications.

# 7. Multiclass Extension

In this paper, our analysis mainly focuses on the binary classification setting, where the label space $\mathcal{Y} = \{0, 1\}$. In this context, we adopt the widely used group fairness criteria: demographic parity defined as $P(\hat{\mathbf{Y}} \mid \mathbf{T}{=}0) = P(\hat{\mathbf{Y}} \mid \mathbf{T}{=}1)$ and equalized odds $P(\hat{\mathbf{Y}}{=}1 \mid \mathbf{T}{=}1, \mathbf{Y}{=}y) = P(\hat{\mathbf{Y}}{=}0 \mid \mathbf{T} = 0, \mathbf{Y} = y)$ for $y \in \{0, 1\}$. These criteria are equivalent to the notions of independence and separation, respectively, as formalized in prior work (Mroueh et al., 2021; Madras et al., 2018).

Extending DP and EO beyond binary labels to multiclass settings is nontrivial and remains an open challenge, with no universally accepted definition of the loss function. Several approaches have been proposed, including information-theoretic formulations based on mutual information (Gupta et al., 2021), dependence measures such as distance correlation (Guo et al., 2022), and direct multiclass analogues of binary fairness criteria (Denis et al., 2021). To demonstrate the flexibility of our framework, we consider a concrete multiclass extension aligned with the definition in (Denis

Table 1. Effect of batch balancing on DP errors with the COMPAS dataset. We report DP on test data for various approaches, with and without our proposed batch-balancing strategy. The results show that balancing consistently reduces DP by an order of magnitude. This empirically supports the theoretical insights of Theorems 1- 5 regarding the role of group imbalance in fairness generalization error.

| | sensitive attribute: gender | | | | | | sensitive attribute: race | | | | | |
|---|---|---|---|---|---|---|---|---|---|---|---|---|
| | 250 | 500 | 1000 | 1500 | 2000 | 2500 | 250 | 500 | 1000 | 1500 | 2000 | 2500 |
| DiffDP | 0.056 | 0.044 | 0.035 | 0.028 | 0.025 | 0.026 | 0.053 | 0.037 | 0.026 | 0.020 | 0.017 | 0.017 |
| DiffDP (ours) | 0.042 | 0.025 | 0.009 | 0.005 | 0.003 | 0.003 | 0.045 | 0.024 | 0.011 | 0.007 | 0.006 | 0.005 |
| HSIC | 0.101 | 0.098 | 0.101 | 0.099 | 0.097 | 0.098 | 0.088 | 0.072 | 0.062 | 0.056 | 0.055 | 0.055 |
| HSIC (ours) | 0.059 | 0.044 | 0.022 | 0.016 | 0.010 | 0.008 | 0.069 | 0.048 | 0.027 | 0.022 | 0.018 | 0.016 |
| PRremover | 0.128 | 0.134 | 0.138 | 0.137 | 0.137 | 0.136 | 0.159 | 0.162 | 0.154 | 0.149 | 0.150 | 0.151 |
| PRremover (ours) | 0.106 | 0.119 | 0.125 | 0.126 | 0.126 | 0.126 | 0.144 | 0.149 | 0.147 | 0.142 | 0.143 | 0.142 |

et al., 2021), showing that our proof technique is general and easily extends to more complex output spaces.

In the multiclass case, the predictor $\hat{y} = f(w; x)$ maps into a broader output range, specifically $[0, a]$ rather than $\{0, 1\}$. In the context of EO, consider our use of the Total Variation (TV) loss as a fairness measure to quantify the difference between prediction distributions conditioned on different labels. Specifically, we use $\mathcal{Y} = \{0, 1, 2, 3\}$ as a demonstration.

For each true label $y \in \{0, 1, 2, 3\}$, we define the TV loss as follows:

$$\ell_E^{F_S}(\mathbf{W}, \mathbf{S} \mid \mathbf{Y} = y) = \frac{1}{2} \sum_{c=0}^{3} \Big| P(\hat{\mathbf{Y}} = c \mid \mathbf{Y} = y, \mathbf{T} = 0) - P(\hat{\mathbf{Y}} = c \mid \mathbf{Y} = y, \mathbf{T} = 1) \Big|, \quad (21)$$

which, in practice, can be approximated by

$$\ell_E^{F_S}(\mathbf{W}, \mathbf{S} \mid \mathbf{Y} = y) = \frac{1}{2} \sum_{c=0}^{3} \left| \frac{n_{0,y,c}}{n_{0,y} + 2} - \frac{n_{1,y,c}}{n_{1,y} + 2} \right|. \quad (22)$$

Here, $n_{t,y,c} = \sum_{i=1}^{n} \mathbb{1}\{\mathbf{T}_i = t, \mathbf{Y}_i = y, \hat{\mathbf{Y}} = c\}$ counts the number of specific pairs in the training data.

We then define an aggregate function over the true labels:

$$g(\mathbf{Z}_u) = \sum_{y \in \{0,1,2,3\}} \ell_E^{F_S}(\mathbf{W}, \mathbf{Z}_u \mid |rmY = y). \quad (23)$$

Based on this multi-class extension of the EO loss, similar to the proof of Lemma 5, we can establish the following

$$\sup_{z_u, \tilde{z}_u^i} |g(z_u) - g(\tilde{z}_u^i)| \leq \frac{2}{\min_{(t,y)}\{n_{t,y}^{Z_u} + 2\}}. \quad (24)$$

Thus, our technical arguments extend naturally to the multiclass setting, and similar bounds to Theorems 6 and 7 can be

obtained in this case. Interestingly, the value of $a$, i.e., the number of classes, does not affect (24) and the final bound. The key point is that, even if the prediction function $f$ is allowed to take any value in $[0, a]$, changing one sample affects the probability estimates (and hence the TV loss) by at most a fixed amount (i.e., at most 1) regardless of $a$. In other words, while $f$'s output may be scaled by $a$, the impact on the fairness loss (measured in terms of probability differences) remains unchanged.

## 8. Conclusion & Future Work

Our theoretical analysis of fairness generalization reveals key interactions between sensitive group imbalances, model capacity, and fairness overfitting through MI and CMI bounds. Additionally, we introduce a new variance-based bounding technique using Efron–Stein inequality that improves upon traditional sub-Gaussian assumptions, leading to tighter results. Empirical evaluations validate both the effectiveness and tightness of the proposed bounds.

Crucially, our framework provides a unified approach for analyzing concentration properties across different loss structures. By leveraging Lemma 2 and Hoeffding's lemma, we ensure that the variance is bounded by sensitivity to individual sample perturbations. Integrating variance control with information-theoretic mutual information bounds, our framework provides guarantees for generalization across various fairness criteria. This demonstrates the wide applicability and the potential of our bounding approach.

Future work could involve extending our theoretical framework to cases where the sensitive attributes are non-binary sensitive attributes or continuous. Another potential direction is leveraging our bounding techniques in other contexts that have similar challenges, i.e., group-based, non-i.i.d loss functions.

## Acknowledgment

The research reported in this publication was supported by funding from King Abdullah University of Science and Technology (KAUST) - Center of Excellence for Generative AI, under award number 5940.

## Impact Statement

This paper presents work whose goal is to advance the field of Machine Learning. There are many potential societal consequences of our work, none of which we feel must be specifically highlighted here.

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

# A. Extra Related Work

**Information-theoretic Bounds:** Information-theoretic bounds have gained significant attention in recent years for characterizing the generalization behavior of learning algorithms (Neu et al., 2021; Wu et al., 2020; Modak et al., 2021; Wang & Mao, 2021; Shui et al., 2020; Wang et al., 2023; Alabdulmohsin, 2020). In the context of supervised learning, various generalization error bounds have been proposed, utilizing different information measures, such as KL divergence (Laakom et al., 2024; Zhou et al., 2023), Wasserstein distance (Rodríguez Gálvez et al., 2021), and mutual information between the samples and model weights (Xu & Raginsky, 2017; Bu et al., 2020). More recently, it has been demonstrated that tighter generalization bounds can be derived using conditional mutual information (CMI) (Steinke & Zakynthinou, 2020; Zhou et al., 2022). Building on this framework, Harutyunyan et al. (2021) introduced $f$-CMI bounds based on model outputs. Additionally, Hellström & Durisi (2022) proposed tighter bounds leveraging CMI of the loss function, further improved by Wang & Mao (2023) through the use of $\Delta L$ CMI. An exact characterization of the generalization error for the Gibbs algorithm is provided in (Aminian et al., 2021). Beyond standard supervised settings, Information-theoretic bounds have been used to study contrastive learning (Dong et al., 2024), meta-learning (Chen et al., 2021; Jose & Simeone, 2021; Rezazadeh et al., 2021), transfer learning (Wu et al., 2020; 2024), and class-generalization error (Laakom et al., 2024).

**Algorithmic Fairness:** Ensuring fairness in machine learning has become a critical area of research, aiming to mitigate biases that may disadvantage certain individuals or groups. Fairness is broadly categorized into *group fairness* (Dwork et al., 2012; Hardt et al., 2016; Corbett-Davies et al., 2017), which seeks to ensure equitable treatment across predefined demographic groups, and *individual fairness* (Dwork et al., 2012; Sharifi-Malvajerdi et al., 2019), which enforces the principle that similar individuals should receive similar predictions. Bias mitigation strategies are typically classified into three approaches: *pre-processing* methods that modify the data before training (Kamiran & Calders, 2012; Calmon et al., 2017), *post-processing* approaches that adjust model predictions after training (Hardt et al., 2016; Jiang et al., 2020), and *in-processing* techniques, the main focus of this paper, that integrate fairness constraints within the learning algorithm (Kamishima et al., 2012; Baharlouei et al., 2020; Mroueh et al., 2021; Lee et al., 2021; 2022; Shah et al., 2022; Alghamdi et al., 2022; Shui et al., 2022; Mehrotra & Vishnoi, 2022).

**Fairness Theory:** In the theoretical analysis of fairness, prior works have primarily focused on the context of domain adaptation, imposing strong assumptions on distributional shifts (Chen et al., 2022; Singh et al., 2021; Coston et al., 2019; Rezaei et al., 2021). For instance, several studies have considered fairness under covariate shifts (Singh et al., 2021; Coston et al., 2019; Rezaei et al., 2021), demographic shifts (Giguere et al., 2022), and prior probability shifts (Biswas & Mukherjee, 2021). However, these assumptions may not always hold in real-world scenarios, limiting their practical applicability (Chen et al., 2022; Singh et al., 2021; Coston et al., 2019; Rezaei et al., 2021; Oneto et al., 2020a; Schumann et al., 2019; Yoon et al., 2020). Beyond domain adaptation, Pham et al. (2023) provides a theoretical analysis of invariant representation learning for domain generalization, establishing upper bounds on both prediction error and unfairness in terms of the Jensen-Shannon (JS) distance. Additionally, Huang & Liu (2024) investigates the convergence rate between empirical and population-level conditional distance covariances, which measure the statistical dependence between model predictions and sensitive attributes. In this paper, we study a different problem, i.e., fairness overfitting in a supervised learning setting, and derive information-theoretic bounds for generalization errors of different fairness losses. Another line of research has focused on the impossibility of satisfying multiple group-level fairness criteria simultaneously Chouldechova (2017); Kleinberg et al. (2016); Tang & Zhang (2022), highlighting the fundamental trade-offs between different fairness notions. In contrast to these impossibility results, which establish lower bounds on population-level fairness risk, our work provides upper bounds on fairness generalization error. Notably, a model can have low generalization error but still perform poorly in terms of fairness on the population level.

**Fairness Generalization:** In the context of fairness generalization, it is worth highlighting the work of Woodworth et al. (2017) and Agarwal et al. (2018), which derive generalization guarantees for loss functions corresponding to DP and EO within specific algorithms. In contrast, our work targets a more general algorithmic framework in the DP and EO setting, accommodating a wider range of loss functions. Moreover, while their analysis primarily focuses on overall sample complexity, our bounds account for additional factors, including properties of the learning algorithm, the choice of loss function, dataset characteristics, and, importantly, the group balance in the data. A more closely related work is that of Oneto et al. (2020b), which derives a generalization bound (Theorem 1) for fairness under randomized algorithms. However, their KL-divergence-based bound is not computable in practice for realistic settings. In contrast, our bounds, particularly Theorem 5, are computable even for modern deep neural networks. This makes our results not only theoretically sound but also practically applicable, enabling the study of fairness generalization in real-world scenarios.

# B. Proofs of the Theorems/Lemmas in Section 3

This appendix includes the proofs of the results presented in the main text in Section 3.

**Lemma 6.** *Let* $\mathbf{X}$ *be a bounded random variable, i.e.,* $\mathbf{X} \in [a, b]$ *almost surely. If* $\mathbb{E}[\mathbf{X}] = 0$*, then* $\mathbf{X}$ *is* $(b-a)$*-sub-gaussian and we have:*

$$\mathbb{E}[e^{\lambda \mathbf{X}}] \leq e^{\frac{\lambda^2 (b-a)^2}{8}}, \quad \forall \lambda \in \mathbb{R}. \tag{25}$$

**Lemma 7.** *(Efron–Stein Inequality) (Boucheron et al. (2013, Theorem 3.1)) Let* $X_1, X_2, \ldots, X_n$ *be independent random variables, and let* $\mathbf{Z} = f(X_1, X_2, \ldots, X_n)$ *be a real-valued square-integrable function of these variables. For each* $i$*, let* $X_i'$ *be an independent copy of* $X_i$*, and define for every* $i$*:*

$$\mathbf{Z}_i' = f(X_1, \ldots, X_{i-1}, X_i', X_{i+1}, \ldots, X_n). \tag{26}$$

*Then, the variance of* $\mathbf{Z} = f(X_1, X_2, \ldots, X_n)$ *satisfies the following inequality:*

$$\mathrm{Var}(\mathbf{Z}) \leq \frac{1}{2} \sum_{i=1}^{n} \mathbb{E}\left[(\mathbf{Z} - \mathbf{Z}_i')^2\right] = \inf_{\mathbf{Z}_i} \sum_{i=1}^{n} \mathbb{E}\left[(\mathbf{Z} - \mathbf{Z}_i)^2\right], \tag{27}$$

*where the infimum is over* $\mathbf{Z}_i = g_i(X_1, \ldots, X_{i-1}, X_{i+1}, \ldots, X_n)$ *the class of all measurable functions* $g_i : \mathcal{X}^{n-1} \to \mathbb{R}$*.*

## B.1. Proof of Lemma 2

**Lemma 2** (restated) Let $g(\mathbf{V}) = g(\mathbf{V}_1, \mathbf{V}_2, \ldots, \mathbf{V}_m)$ be a real-valued function of $m$ i.i.d random variables. Given fixed $v = (v_1, \ldots, v_m)$ and an index $i \in \{1, \ldots, m\}$, define $\tilde{v}^i = (\tilde{v}_1, \ldots, \tilde{v}_m)$ such that $\tilde{v}_j = v_j$ for all $j \neq i$ except on $i$, i.e., $\tilde{v}_i \neq v_j$. If $g(v)$ satisfies the following condition for a fixed $n_0^v$ (and as a consequence $n_1^v = m - n_0^v$),

$$\sup_{v \in \mathcal{V}^m, \tilde{v}_i \in \mathcal{V}} |g(v) - g(\tilde{v}^i)| \leq \beta, \quad 1 \leq i \leq m, \tag{28}$$

where $\beta$ is a random variable that may depend on $n_0^v$ and hence is a random variable. Then, we have

$$\mathrm{Var}(g(\mathbf{V})) \leq \frac{m}{4} \mathbb{E}[\beta^2]. \tag{29}$$

*Proof.* We will use Efron-Stein inequality in Lemma 7 to show this result. Using the Lemma statement on $g(\mathbf{V})$, we have

$$\mathrm{Var}(g(\mathbf{V})) \leq \inf_{g_i} \sum_{i=1}^{m} \mathbb{E}\left[\left(g(\mathbf{V}) - g_i(\mathbf{V}_0, \ldots, \mathbf{V}_{i-1}, \mathbf{V}_{i+1}, \ldots, \mathbf{V}_m)\right)^2\right] \tag{30}$$

where $g_i$ is an arbitrary function. Using the law of total expectation in (30), we have

$$\mathrm{Var}(g(\mathbf{V})) \leq \inf_{g_i} \sum_{i=1}^{m} \mathbb{E}_{n_0^\mathbf{V}} \mathbb{E}\left[\left(g(\mathbf{V}_u) - g_i(\mathbf{V}_0, \ldots, \mathbf{V}_{i-1}, \mathbf{V}_{i+1}, \ldots, \mathbf{V}_m)\right)^2 | n_0^V\right] \tag{31}$$

Note that by conditioning on $n_0^\mathbf{V}$, we have also $n_1^\mathbf{V} = m - n_0^\mathbf{V}$ is fixed. As this is true for any arbitrary $g_i$ depending on $(\mathbf{V}_1, \ldots, \mathbf{V}_{i-1}, \mathbf{V}_{i+1}, \ldots, \mathbf{V}_m)$, we select

$$g_i(\mathbf{V}_1, \ldots, \mathbf{V}_{i-1}, \mathbf{V}_{i+1}, \mathbf{V}_m) = \frac{1}{2}\Big( \sup_{x_i' \in \mathcal{X}} g(\mathbf{V}_1, \ldots, \mathbf{V}_{i-1}, x_i', \mathbf{V}_{i+1}, \ldots, \mathbf{V}_m) \tag{32}$$

$$+ \inf_{x_i' \in \mathcal{X}} g(\mathbf{V}_1, \ldots, \mathbf{V}_{i-1}, x_i', \mathbf{V}_{i+1}, \ldots, \mathbf{V}_m)\Big) \tag{33}$$

Hence, as in the inner expectation in (31), $n_0^\mathbf{V}$ and $n_1^\mathbf{V}$ are fixed, using the main condition, we have

$$|g(\mathbf{V}) - g_i(\mathbf{V}_0, \ldots, \mathbf{V}_{i-1}, \mathbf{V}_{i+1}, \ldots, \mathbf{V}_m)| \leq \frac{1}{2}\beta \tag{34}$$

Taking the square on both sides,

$$(g(\mathbf{V}) - g_i(\mathbf{V}_0, \ldots, \mathbf{V}_{i-1}, \mathbf{V}_{i+1}, \ldots, \mathbf{V}_m))^2 \leq \frac{1}{4}\beta^2 \tag{35}$$

Replacing in the inner expectation in (31), we have

$$\mathrm{Var}(g(\mathbf{V})) \leq \sum_{i=1}^{m} \mathbb{E}[\frac{1}{4}\beta^2] = \frac{m}{4}\mathbb{E}[\beta^2] \tag{36}$$

$\square$

## C. Proof of the Theorems/Lemmas in Section 4

### C.1. Proof of Lemma 3

**Lemma 3** (restated) Assume that $|f| \in [0,1]$. let $g : \mathcal{V}^m \to \mathbb{R}$ be defined as $g(v) = g(v_1, \ldots, v_m) = \ell_E^F(w, v)$, where $\ell_E^F(w, v)$ is defined (1). Then, for a fixed $w \in \mathcal{W}$ and a fixed $n_0^v$ and $n_1^v$, for any $1 \leq i \leq m$, we have

$$\sup_{v \in \mathcal{V}^m, \tilde{v}_i \in \mathcal{V}} |g(v) - g(\tilde{v}^i)| \leq \frac{1}{\mathbb{H}(n_0^v, n_1^v)}, \tag{37}$$

where $\mathbb{H}$ denotes a shifted harmonic mean operator, i.e., $\forall \{a_i\}_{i=1}^k$, $\mathbb{H}(a_1, \cdots, a_k) = \frac{1}{\frac{1}{a_1+2} + \cdots + \frac{1}{a_k+2}}$.

*Proof.* For $1 \leq i \leq m$, we have

$$|g(v) - g(\tilde{v}^i)| = \Big|\Big|\frac{1}{n_0^v + 2}\sum_{T_j=0} f(w, x_j) - \frac{1}{n_1^v + 2}\sum_{T_j=1} f(w, x_j)| \tag{38}$$

$$- |\frac{1}{n_0^{\tilde{v}^i} + 2}\sum_{\tilde{T}_j=0} f(w, \tilde{x}_j) - \frac{1}{n_1^{\tilde{v}^i} + 2}\sum_{\tilde{T}_j=1} f(w, \tilde{x}_j)|\Big| \tag{39}$$

$$\leq \Big|\frac{1}{n_0^v + 2}\sum_{T_j=0} f(w, x_j) - \frac{1}{n_1^v + 2}\sum_{T_j=1} f(w, x_j) \tag{40}$$

$$- \frac{1}{n_0^{\tilde{v}^i} + 2}\sum_{\tilde{T}_j=0} f(w, \tilde{x}_j) + \frac{1}{n_1^{\tilde{v}^i} + 2}\sum_{\tilde{T}_j=1} f(w, \tilde{x}_j)\Big|, \tag{41}$$

where the inequality comes from the fact $|| \cdot | - | \cdot || \leq | \cdot - \cdot |$. Furthermore, as $\tilde{v}_j = v_j$ for all $j \neq i$, based on the values of $t_i$ and $\tilde{t}_i$, we have 4 different cases:

**1st case** $t_i = \tilde{t}_i = 0$: In this case, we have $n_0^v = n_0^{\tilde{v}^i}$, $n_1^v = n_1^{\tilde{v}^i}$, and $f(w, x_j) = f(w, \tilde{x}_j) \forall j \neq i$. Hence, based on (41), we have:

$$|g(v) - g(\tilde{v}^i)| \leq |\frac{1}{n_0^v + 2}f(w, x_i) - \frac{1}{n_0^v + 2}f(w, \tilde{x}_i)| \tag{42}$$

$$\leq \frac{1}{n_0^v + 2}|f(w, x_i) - f(w, \tilde{x}_i)| \leq \frac{1}{n_0^v + 2} \tag{43}$$

**2nd case** $t_i = \tilde{t}_i = 1$: Similar to the previous case, we can show that

$$|g(v) - g(\tilde{v}^i)| \leq \frac{1}{n_1^v + 2} \tag{44}$$

**3rd case** $t_i = 0, \tilde{t}_i = 1$: In this case, we have $n_0^{\tilde{v}} = n_0^v - 1$, $n_1^{\tilde{v}} = n_1^v + 1$. Hence, we have:

$$|g(v) - g(\tilde{v}^i)| \leq \left| \frac{1}{n_0^v + 2} \sum_{T_j = 0} f(w, x_j) - \frac{1}{n_1^v + 2} \sum_{T_j = 1} f(w, x_j) \right| \tag{45}$$

$$- \frac{1}{n_0^v + 1} \sum_{\tilde{T}_j = 0} f(w, \tilde{x}_j) + \frac{1}{n_1^v + 3} \sum_{\tilde{T}_j = 1} f(w, \tilde{x}_j) \Bigg| \tag{46}$$

$$= \left| \left( \frac{1}{n_0^v + 2} - \frac{1}{n_0^v + 1} \right) \sum_{t_j = 0, j \neq i} f(w, x_j) + \left( \frac{1}{n_1^v + 3} - \frac{1}{n_1^v + 2} \right) \sum_{t_j = 1, j \neq i} f(w, x_j) \right.$$

$$\left. + \frac{1}{n_0^v + 2} f(w, x_i) + \frac{1}{n_1^v + 3} f(w, \tilde{x}_i) \right| \tag{47}$$

$$= \left| \frac{-1}{(n_0^v + 2)(n_0^v + 1)} \sum_{t_j = 0, j \neq i} f(w, x_j) + \frac{-1}{(n_1^v + 3)(n_1^v + 2)} \sum_{t_j = 1, j \neq i} f(w, x_j) \right.$$

$$\left. + \frac{1}{n_0^v + 2} f(w, x_i) + \frac{1}{n_1^v + 3} f(w, \tilde{x}_i) \right| \tag{48}$$

$$= \left| \left( \frac{1}{n_0^v + 2} f(w, x_i) - \frac{1}{(n_0^v + 2)(n_0^v + 1)} \sum_{t_j = 0, j \neq i} f(w, x_j) \right) \right.$$

$$\left. + \left( \frac{1}{n_1^v + 3} f(w, \tilde{x}_i) - \frac{1}{(n_1^v + 3)(n_1^v + 2)} \sum_{t_j = 1, j \neq i} f(w, x_j) \right) \right| \tag{49}$$

$$\leq \left| \frac{1}{n_0^v + 2} f(w, x_i) - \frac{1}{(n_0^v + 2)(n_0^v + 1)} \sum_{t_j = 0, j \neq i} f(w, x_j) \right| \tag{50}$$

$$+ \left| \frac{1}{n_1^v + 3} f(w, \tilde{x}_i) - \frac{1}{(n_1^v + 3)(n_1^v + 2)} \sum_{t_j = 1, j \neq i} f(w, x_j) \right| \tag{51}$$

where the last inequality comes from the fact that $|| \cdot | - | \cdot || \leq | \cdot - \cdot |$. Next as $f(w, \cdot) \in [0, 1]$ is positive, we have:

$$|g(v) - g(\tilde{v}^i)|$$

$$\leq \max \left( \frac{1}{n_0^v + 2} f(w, x_i), \frac{1}{(n_0^v + 2)(n_0^v + 1)} \sum_{t_j = 0, j \neq i} f(w, x_j) \right)$$

$$+ \max \left( \frac{1}{n_1^v + 3} f(w, \tilde{x}_i), \frac{1}{(n_1^v + 3)(n_1^v + 2)} \sum_{t_j = 1, j \neq i} f(w, x_j) \right) \tag{52}$$

$$\leq \max \left( \frac{1}{n_0^v + 2}, \frac{1}{(n_0^v + 2)(n_0^v + 1)} \sum_{t_j = 0, j \neq i} 1 \right)$$

$$+ \max \left( \frac{1}{n_1^v + 3}, \frac{1}{(n_1^v + 3)(n_1^v + 2)} \sum_{t_j = 1, j \neq i} 1 \right) \tag{53}$$

$$= \max \left( \frac{1}{n_0^v + 2}, \frac{1}{(n_0^v + 2)(n_0^v + 1)} (n_0^v - 1) \right)$$

$$+ \max \left( \frac{1}{n_1^v + 3}, \frac{1}{(n_1^v + 3)(n_1^v + 2)} (n_1^v - 1) \right) \tag{54}$$

As, $\forall n_0^v, n_1^v$, we have $\frac{n_0^v - 1}{n_0^v + 1} \leq 1$ and $\frac{n_1^v - 1}{n_1^v + 2} \leq 1$. Hence, we have from (54),

$$|g(v) - g(\tilde{v}^i)| \leq \max \left( \frac{1}{n_0^v + 2}, \frac{1}{n_0^v + 2} \right) + \max \left( \frac{1}{n_1^v + 3}, \frac{1}{n_1^v + 3} \right) \tag{55}$$

$$\leq \frac{1}{n_0^v + 2} + \frac{1}{n_1^v + 3} \leq \frac{1}{n_0^v + 2} + \frac{1}{n_1^v + 2} \tag{56}$$

**4th case** $t_i = 1, \tilde{t}_i = 0$**:**   Similarly, we can show that

$$|g(v) - g(\tilde{v}^i)| \leq \frac{1}{n_0^v + 3} + \frac{1}{n_1^v + 2} \leq \frac{1}{n_0^v + 2} + \frac{1}{n_1^v + 2} \tag{57}$$

Hence, using the cases discussed above, for a fixed $w$, we always have

$$|g(v) - g(\tilde{v}^i)| \leq \frac{1}{n_0^v + 2} + \frac{1}{n_1^v + 2} \quad 1 \leq i \leq m \tag{58}$$

For a fixed $n_0^v$ and $n_1^v$, taking the supremum with respect $v, \tilde{v}^i$ in both sides of (58), we have

$$\sup_{v \in \mathcal{V}^m, \tilde{v}_i^i \in \mathcal{V}} |g(v) - g(\tilde{v}^i)| \leq \frac{1}{n_0^v + 2} + \frac{1}{n_1^v + 2} \quad 1 \leq i \leq n \tag{59}$$

which completes the proof. $\qquad\square$

## C.2. Proof of Theorem 1

**Theorem 1** (restated) Assume that $|f| \in [0, 1]$, then for any $m \in \{2, \cdots, n\}$

$$\overline{gen}_{fairness} \leq \frac{1}{|C_n^m|} \sum_{u \in C_n^m} \sqrt{\frac{m}{2} \mathbb{E}_{\mathbf{V}_u}[(\frac{1}{n_0^{\mathbf{V}_u} + 2} + \frac{1}{n_1^{\mathbf{V}_u} + 2})^2] I(\mathbf{W}, \mathbf{V}_u)} \tag{60}$$

where $C_n^m$ is the set of $m$-combinations.

*Proof.* Let $m \in \{2, \cdots, n\}$ be a fixed number and let $P_n^m$ be the set of $m$-permutations of $n$. For a fixed $u \in P_n^m$, the corresponding $\ell_E^F(w, Z_u)$ for the set $Z_u \sim \mu^m$ is as follows:

$$\ell_E^F(w, Z_u) \triangleq \left| \frac{1}{n_0^{V_u} + 2} \sum_{T_i=0} f(w, x_i) - \frac{1}{n_1^{V_u} + 2} \sum_{T_i=1} f(w, x_i) \right| \tag{61}$$

Then, using the notion of $\ell_E^F(\mathbf{W}, \mathbf{Z}_u)$, starting from Definition 1 we have:

$$\overline{gen}_{fairness} = |\mathbb{E}_{P_{\mathbf{W}} \otimes P_{\mathbf{S}}}[\ell_E^F(\mathbf{W}, \mathbf{S})] - \mathbb{E}_{P_{\mathbf{W},\mathbf{s}}}[\ell_E^F(\mathbf{W}, \mathbf{S})]| \tag{62}$$

$$= \left| \mathbb{E}_{P_{\overline{\mathbf{W}}, \overline{\mathbf{z}}_u}}[\ell_E^F(\overline{\mathbf{W}}, \overline{\mathbf{Z}}_u)] - \frac{1}{|P_n^m|} \sum_{u \in P_n^m} \mathbb{E}_{P_{\mathbf{W}, \mathbf{z}_u}}[\ell_E^F(\mathbf{W}, \mathbf{Z}_u)] \right| \tag{63}$$

$$\leq \frac{1}{|P_n^m|} \sum_{u \in P_n^m} \left| \mathbb{E}_{P_{\overline{\mathbf{W}}, \overline{\mathbf{z}}_u}}[\ell_E^F(\overline{\mathbf{W}}, \overline{\mathbf{Z}}_u)] - \mathbb{E}_{P_{\mathbf{W}, \mathbf{z}_u}}[\ell_E^F(\mathbf{W}, \mathbf{Z}_u)] \right| \tag{64}$$

Hence, to bound $\overline{gen}_{fairness}$, we only need to bound $\left| \mathbb{E}_{P_{\overline{\mathbf{W}}, \overline{\mathbf{z}}_u}}[\ell_E^F(\overline{\mathbf{W}}, \overline{\mathbf{Z}}_u)] - \mathbb{E}_{P_{\mathbf{W}, \mathbf{z}_u}}[\ell_E^F(\mathbf{W}, \mathbf{Z}_u)] \right|$.

Using the Donsker–Varadhan variational representation of the relative entropy, we have for any function $h(\cdot)$ and for every $\lambda$

$$I(\mathbf{W}, \mathbf{V}_u) \geq \mathbb{E}_{P_{\mathbf{W}, \mathbf{V}_u}}[\lambda h(\mathbf{W}, \mathbf{V}_u)] - \log \mathbb{E}_{P_{\overline{\mathbf{W}}} \otimes P_{\overline{\mathbf{V}}_u}}[e^{\lambda h(\overline{\mathbf{W}}, \overline{V_u})}], \forall \lambda \in \mathbb{R}. \tag{65}$$

By taking $h(w, v_u) = \ell_E^F(w, v_u) - \mathbb{E}_{P_{\overline{\mathbf{V}}_u}}[\ell_E^F(w, \overline{\mathbf{V}}_u)]$

$$I(\mathbf{W}, \mathbf{V}_u) \geq \mathbb{E}_{P_{\mathbf{W}, \mathbf{V}_u}}[\lambda h(\mathbf{W}, \mathbf{V}_u)] - \log \mathbb{E}_{P_{\overline{\mathbf{W}}} \otimes P_{\overline{\mathbf{V}}_u}}[e^{\lambda h(\overline{\mathbf{W}}, \overline{V_u})}], \forall \lambda \in \mathbb{R}. \tag{66}$$

$$I(\mathbf{W}, \mathbf{V}_u) \geq \mathbb{E}_{P_{\mathbf{W}, \mathbf{V}_u}}\left[ \lambda(\ell_E^F(\mathbf{W}, \mathbf{V}_u) - \mathbb{E}_{P_{\overline{V}_u}}[\ell_E^F(\mathbf{W}, \overline{\mathbf{V}}_u)]) \right]$$
$$- \log \mathbb{E}_{P_{\overline{\mathbf{W}}} \otimes P_{\overline{\mathbf{V}}_u}}[e^{\lambda(\ell_E^F(\overline{\mathbf{W}}, \overline{\mathbf{V}}_u) - \mathbb{E}_{P_{\overline{V}_u}}[\ell_E^F(\overline{\mathbf{W}}, \overline{V_u})])}], \forall \lambda \in \mathbb{R}. \tag{67}$$

$$I(\mathbf{W}, \mathbf{V}_u) \geq \mathbb{E}_{P_{\mathbf{W}, \mathbf{V}_u}}\left[ \lambda(\ell_E^F(\mathbf{W}, \mathbf{V}_u) - \mathbb{E}_{P_{\overline{V}_u}}[\ell_E^F(\mathbf{W}, \overline{\mathbf{V}}_u)]) \right]$$
$$- \log \mathbb{E}_{P_{\overline{\mathbf{W}}}}\left[ \mathbb{E}_{P_{\overline{V}_u}}[e^{\lambda(\ell_E^F(w, \overline{V_u}) - \mathbb{E}_{P_{\overline{V}_u}}[\ell_E^F(w, \overline{V_u})])}]|\overline{\mathbf{W}} = w \right], \forall \lambda \in \mathbb{R}. \tag{68}$$

Next, for a fixed $w$, we will bound $\mathbb{E}_{P_{\overline{\mathbf{V}_u}}}[e^{\lambda(\ell_E^F(w,\overline{\mathbf{V}_u})-\mathbb{E}_{P_{\overline{V_u}}}[\ell_E^F(w,\overline{V_u})])}]$. To this end, we want to bound the variance of $\ell_E^F(w,\overline{\mathbf{V}_u})-\mathbb{E}_{P_{\overline{V_u}}}[\ell_E^F(w,\overline{V_u})]$. As $\mathrm{Var}(\mathbf{X}+s)=\mathrm{Var}(\mathbf{X}), \forall s \in sR$ and $\mathbb{E}[\ell_E^F(w,\mathbf{V}_u)]$ is constant for a fixed $w$, we have

$$\mathrm{Var}\Big(\ell_E^F(w,\overline{\mathbf{V}_u})-\mathbb{E}[\ell_E^F(\mathbf{W},\overline{\mathbf{V}_u})]\Big)=\mathrm{Var}\big(\ell_E^F(w,\overline{\mathbf{V}_u})\big) \tag{69}$$

Hence, it is sufficient to bound the variance $\ell_E^F(w,\overline{\mathbf{V}_u})$. We will use the results of Lemma 3 and Lemma 2 to upper bound the aforementioned variance.

As $|f| \in [0,1]$, we have

$$\ell(w,v_u)=\Big|\frac{1}{n_0+2}\sum_{T_i=0}f(w,x_i)-\frac{1}{n_1+2}\sum_{T_i=1}f(w,x_i)\Big| \leq \max\left(\Big|\frac{1}{n_0+2}\sum_{T_i=0}f(w,x_i)\Big|,\Big|\frac{1}{n_1+2}\sum_{T_i=1}f(w,x_i)\Big|\right) \tag{70}$$

$$\leq \max(\frac{n_0}{n_0+2},\frac{n_1}{n_1+2}) \leq \max(1,1)=1 \tag{71}$$

Hence, $\ell(w,v_u)$ is bounded and therefore square-integrable. Applying Lemma 3 with $g(v_u)=\ell(w,v_u)$, we have for fixed $w, n_0^{V_u}$ :

$$\sup_{v_u \in \mathcal{V}^m, \tilde{v}_i \in \mathcal{V}}|g(v_u)-g(\tilde{v}_u^i)| \leq \frac{1}{n_0^{V_u}+2}+\frac{1}{n_1^{V_u}+2} \quad 1 \leq i \leq n \tag{72}$$

Hence, $g(v_u)=\ell(w,v_u)$ verifies the condition in Lemma 2 with $\beta=\frac{1}{n_0^{V_u}+2}+\frac{1}{n_1^{V_u}+2}$. Thus, using Lemma 2, the variance of the random variable $\ell_E^F(w,\overline{\mathbf{V}_u})$ can be bounded as

$$\mathrm{Var}(\ell_E^F(w,\overline{\mathbf{V}_u})) \leq \frac{m}{4}\mathbb{E}[(\frac{1}{n_0^{\mathbf{V}_u}+2}+\frac{1}{n_1^{\mathbf{V}_u}+2})^2], \tag{73}$$

for every $w \in \mathcal{W}$. Thus, from (69),we have

$$\mathrm{Var}\big(\ell_E^F(w,\mathbf{V}_u)-\mathbb{E}[\ell_E^F(\mathbf{W},\mathbf{V}_u)]\big)=\mathrm{Var}(\ell_E^F(w,\mathbf{V}_u)) \leq \frac{m}{4}\mathbb{E}[(\frac{1}{n_0^{\mathbf{V}_u}+2}+\frac{1}{n_1^{\mathbf{V}_u}+2})^2] \tag{74}$$

So, the random variable $\ell_E^F(w,\mathbf{V}_u)-\mathbb{E}[\ell_E^F(w,\mathbf{V}_u)]$ has a bounded variance and an expectation $\mathbb{E}_{P_{\overline{\mathbf{V}_u}}}[\ell_E^F(w,\mathbf{V}_u)-\mathbb{E}[\ell_E^F(w,\mathbf{V}_u)]]=0$. Hence, using Hoeffding's lemma, we have for every $w \in \mathcal{W}$:

$$\mathbb{E}_{P_{\overline{\mathbf{V}_u}}}[e^{\lambda(\ell_E^F(w,\overline{\mathbf{V}_u})-\mathbb{E}_{P_{\overline{V_u}}}[\ell_E^F(w,\overline{V_u})])}] \leq e^{\frac{\lambda^2 m}{8}\mathbb{E}[(\frac{1}{n_0^{\mathbf{V}_u}+2}+\frac{1}{n_1^{\mathbf{V}_u}+2})^2]} \tag{75}$$

Replacing in (68), we have

$$I(\mathbf{W},\mathbf{V}_u) \geq \mathbb{E}_{P_{\mathbf{W},\mathbf{V}_u}}\Big[\lambda\big(\ell_E^F(\mathbf{W},\mathbf{V}_u)-\mathbb{E}_{P_{\overline{V_u}}}[\ell_E^F(\mathbf{W},\overline{V_u})]\big)\Big]$$
$$-\log\mathbb{E}_{P_{\overline{\mathbf{W}}}}\Big[e^{\frac{\lambda^2 m}{8}\mathbb{E}[(\frac{1}{n_0^{\mathbf{V}_u}+2}+\frac{1}{n_1^{\mathbf{V}_u}+2})^2]}\Big|\overline{\mathbf{W}}=w\Big], \forall \lambda \in \mathbb{R}. \tag{76}$$

$$I(\mathbf{W},\mathbf{V}_u) \geq \mathbb{E}_{P_{\mathbf{W},\mathbf{V}_u}}\Big[\lambda\big(\ell_E^F(\mathbf{W},\mathbf{V}_u)-\mathbb{E}_{P_{\overline{V_u}}}[\ell_E^F(\mathbf{W},\overline{V_u})]\big)\Big]$$
$$-\frac{\lambda^2 m}{8}\mathbb{E}[(\frac{1}{n_0^{\mathbf{V}_u}+2}+\frac{1}{n_1^{\mathbf{V}_u}+2})^2], \forall \lambda \in \mathbb{R}. \tag{77}$$

where the last inequality comes from the fact that $\mathbb{E}[(\frac{1}{n_0^{\mathbf{V}_u}+2}+\frac{1}{n_1^{\mathbf{V}_u}+2})^2]$ is independent of $\mathbf{W}$. From (77), we have

$$\frac{\lambda^2 m}{8}\mathbb{E}[(\frac{1}{n_0^{\mathbf{V}_u}+2}+\frac{1}{n_1^{\mathbf{V}_u}+2})^2]-\lambda\Big(\mathbb{E}_{P_{\mathbf{W},\overline{\mathbf{V}}_u}}[\ell_E^F(\mathbf{W},\overline{\mathbf{V}}_u)]-\mathbb{E}_{P_{\mathbf{W},\mathbf{V}_u}}[\ell_E^F(\mathbf{W},\mathbf{V}_u)]\Big)$$
$$+I(\mathbf{W},\mathbf{V}_u) \geq 0, \forall \lambda \in \mathbb{R}. \tag{78}$$

So, the parabola (with respect to $\lambda$) on the left side of (78) is always positive. Hence, its discriminant must be negative:

$$\left| \mathbb{E}_{P_{\mathbf{W},\overline{\mathbf{Z}}_u}}[\ell_E^F(\mathbf{W}, \overline{\mathbf{Z}}_u)] - \mathbb{E}_{P_{\mathbf{W},\mathbf{Z}_u}}[\ell_E^F(\mathbf{W}, \mathbf{Z}_u)] \right| \leq \sqrt{\frac{m}{2} \mathbb{E}[(\frac{1}{n_0^{V_u}+2} + \frac{1}{n_1^{V_u}+2})^2] I(\mathbf{W}, \mathbf{V}_u)} \tag{79}$$

Replacing in (64), we have

$$\overline{gen}_{fairness} \leq \frac{1}{|P_n^m|} \sum_{u \in P_n^m} \sqrt{\frac{m}{2} \mathbb{E}[(\frac{1}{n_0^{\mathbf{V}_u}+2} + \frac{1}{n_1^{\mathbf{V}_u}+2})^2] I(\mathbf{W}, \mathbf{V}_u)} \tag{80}$$

In addition, as mutual information is invariant to sample permutations, we have

$$\overline{gen}_{fairness} \leq \frac{1}{|C_n^m|} \sum_{u \in C_n^m} \sqrt{\frac{m}{2} \mathbb{E}[(\frac{1}{n_0^{\mathbf{V}_u}+2} + \frac{1}{n_1^{\mathbf{V}_u}+2})^2] I(\mathbf{W}, \mathbf{V}_u)} \tag{81}$$

where $C_n^m$ is the set of $m$-combinations.

$\square$

### C.3. Proof of Lemma 4

**Lemma 4** (restated) $\forall \lambda \in \mathbb{R}$, we have

$$\mathbb{E}_{\overline{\mathbf{R}}_u|\mathbf{Z}_{[2n]}=z_{[2n]}, \overline{\mathbf{W}}=w}\left[e^{\lambda(\ell_E^F(w,\mathbf{S}_u) - \ell_E^F(w,\overline{\mathbf{S}}_u))}\right] \leq e^{\frac{\lambda^2 m}{8} \mathbb{E}[\left(\frac{1}{n_0^{S_u}+2} + \frac{1}{n_1^{S_u}+2} + \frac{1}{n_0^{\overline{S}_u}+2} + \frac{1}{n_1^{\overline{S}_u}+2}\right)^2]} \tag{82}$$

*Proof.* we will bound $\log \mathbb{E}_{\overline{\mathbf{R}}_u|\mathbf{Z}_{[2n]}=z_{[2n]}, \overline{\mathbf{W}}=w}[e^{\lambda(\ell_E^F(w,\mathbf{S}_u) - \ell_E^F(w,\overline{\mathbf{S}}_u))}]$ by bounding its variance using Lemma 3 and Lemma 2.

Given a fixed $z_{[2n]} \in \mathcal{Z}^{2n}$ and $w \in \mathcal{W}$, we define $g_{z_{[2n]}} : \mathcal{V}^n \to \mathbb{R}$ as follows: $g_{z_{[2n]}}(\mathbf{R}_u) = \ell_E^F(w, \mathbf{S}_u) - \ell_E^F(w, \overline{\mathbf{S}}_u)$. Furthermore, given a set $r_u = (r_1, \ldots, r_m)$ and an index $i \in \{1, \ldots, m\}$, define $\tilde{r}_u^i = (\tilde{r}_1, \ldots, \tilde{r}_m)$ such that $\tilde{r}_j = r_j$ for all $j \neq i$ except on $i$, i.e., $\tilde{r}_i \neq r_j$. The goal is to upper-bound $|g_{z_{[2n]}}(r_u) - g_{z_{[2n]}}(\tilde{r}_u)|$ for $i \in \{1, \ldots, m\}$.

For $i \in [1, m]$, we have

$$|g_{z_{[2n]}}(r_u) - g_{z_{[2n]}}(\tilde{r}_u)| = |\ell_E^F(w, S_u) - \ell_E^F(w, \overline{S}_u) - \ell_E^F(w, \tilde{S}_u) + \ell_E^F(w, \tilde{\overline{S}}_u)| \tag{83}$$

$$\leq |\ell_E^F(w, S_u) - \ell_E^F(w, \tilde{S}_u)| + |\ell_E^F(w, \overline{S}_u) - \ell_E^F(w, \tilde{\overline{S}}_u)| \tag{84}$$

Hence, we can bound each term separately using Lemma 3. It follows that:

$$|g_{z_{[2n]}}(r_u) - g_{z_{[2n]}}(\tilde{r}_u^i)| \leq \left(\frac{1}{n_0^{S_u}+2} + \frac{1}{n_1^{S_u}+2}\right) + \left(\frac{1}{n_0^{\overline{S}_u}+2} + \frac{1}{n_1^{\overline{S}_u}+2}\right) \tag{85}$$

$$\tag{86}$$

Hence we have

$$|g_{z_{[2n]}}(r_u) - g_{z_{[2n]}}(\tilde{r}_u^i)| \leq \frac{1}{n_0^{S_u}+2} + \frac{1}{n_1^{S_u}+2} + \frac{1}{n_0^{\overline{S}_u}+2} + \frac{1}{n_1^{\overline{S}_u}+2} \quad 1 \leq i \leq m \tag{87}$$

Furthermore, $g_{z_{[2n]}}(\mathbf{R}_u)$, being the difference of two bounded random variables (see (70)), is itself bounded and hence square-integrable. Now, we have all the ingredients to apply Lemma 2. Using the Lemma statement on $g_{z_{[2n]}}(\mathbf{R}_u) = \ell_E^F(w, \mathbf{S}_u) - \ell_E^F(w, \overline{\mathbf{S}}_u)$, we have

$$\text{Var}\left(\ell_E^F(w, \mathbf{S}_u) - \ell_E^F(w, \overline{\mathbf{S}}_u)\right) \leq \frac{m}{4} \mathbb{E}[(\frac{1}{n_0^{S_u} + 2} + \frac{1}{n_1^{S_u} + 2} + \frac{1}{n_0^{\overline{S}_u} + 2} + \frac{1}{n_1^{\overline{S}_u} + 2})^2] \tag{88}$$

Furthermore, as $\overline{\mathbf{R}}_i$ are independent Rademacher variables, we have

$$\mathbb{E}_{\overline{\mathbf{R}}_u | \mathbf{Z}_{[2n]} = z_{[2n]}, \overline{\mathbf{W}} = w} [\ell_E^F(w, \mathbf{S}_u) - \ell_E^F(w, \overline{\mathbf{S}}_u)] = 0 \tag{89}$$

Hence, the random variable $\ell_E^F(w, \mathbf{S}_u) - \ell_E^F(w, \overline{\mathbf{S}}_u)$ is centered and with bounded variance (as shown in (88)). It follows that using Hoeffding's lemma, we have the final results:

$$\mathbb{E}\left[e^{\lambda(\ell_E^F(w, \mathbf{S}_u) - \ell_E^F(w, \overline{\mathbf{S}}_u))}\right] \leq e^{\frac{\lambda^2 m}{8} \mathbb{E}[(\frac{1}{n_0^{S_u} + 2} + \frac{1}{n_1^{S_u} + 2} + \frac{1}{n_0^{\overline{S}_u} + 2} + \frac{1}{n_1^{\overline{S}_u} + 2})^2]} \tag{90}$$

$\square$

## C.4. Proof of Theorem 2

**Theorem 2** (restated) Assume that $|f| \in [0, 1]$, then for any $m \in \{2, \cdots, n\}$

$$\overline{gen}_{fairness} \leq \frac{1}{|C_n^m|} \mathbb{E}_{\mathbf{Z}_{[2n]}} \left[ \sum_{u \in C_n^m} \sqrt{\frac{m}{2} \mathbb{E}_{\mathbf{R}_u}[(\frac{1}{n_0^{S_u} + 2} + \frac{1}{n_1^{S_u} + 2} + \frac{1}{n_0^{\overline{S}_u} + 2} + \frac{1}{n_1^{\overline{S}_u} + 2})^2] I_{z_{[2n]}}(\mathbf{W}, \mathbf{R}_u)} \right] \tag{91}$$

where $C_n^m$ is the set of $m$-combinations.

*Proof.* Similar to the MI setting in Theorem 1, the fairness generalization in the supersample setting, can be bounded as follows:

$$\overline{gen}_{fairness} = |\mathbb{E}_{P_{\mathbf{W}, \mathbf{z}_{[2n]}, \mathbf{R}}} [\ell_E^F(\mathbf{W}, \mathbf{S}) - \ell_E^F(\mathbf{W}, \overline{\mathbf{S}})]| \tag{92}$$

$$\leq \mathbb{E}_{P_{\mathbf{Z}_{[2n]}}} |\mathbb{E}_{P_{\mathbf{W}, \mathbf{R} | \mathbf{Z}_{[2n]}}} [\ell_E^F(\mathbf{W}, \mathbf{S}) - \ell_E^F(\mathbf{W}, \overline{\mathbf{S}})]| \tag{93}$$

$$\leq \frac{1}{|P_n^m|} \sum_{u \in P_n^m} \mathbb{E}_{P_{\mathbf{Z}_{[2n]}}} |\mathbb{E}_{P_{\mathbf{W}, \mathbf{R} | \mathbf{Z}_{[2n]}}} [\ell_E^F(\mathbf{W}, \mathbf{S}_u) - \ell_E^F(\mathbf{W}, \overline{\mathbf{S}}_u)]| \tag{94}$$

For a fixed $u \in P_n^m$, let $(\overline{\mathbf{W}}, \overline{\mathbf{R}}_u)$ be an independent copy of $(\mathbf{W}, \mathbf{R}_u)$. The disintegrated mutual information $I_{z_{[2n]}}(\mathbf{W}, \mathbf{R}_u)$ is equal to:

$$I_{z_{[2n]}}(\mathbf{W}, \mathbf{R}_u) = D\big(P_{\mathbf{W}, \mathbf{R}_u | \mathbf{Z}_{[2n]} = z_{[2n]}} \| P_{\mathbf{W} | \mathbf{Z}_{[2n]} = z_{[2n]}} P_{\mathbf{R}_u}\big), \tag{95}$$

Thus, by the Donsker–Varadhan variational representation of KL divergence, $\forall \lambda \in \mathbb{R}$, we have

$$I_{z_{[2n]}}(\mathbf{W}; \mathbf{R}_u) \geq \lambda \mathbb{E}_{\mathbf{W}, \mathbf{R}_u | \mathbf{Z}_{[2n]} = z_{[2n]}} [\ell_E^F(\mathbf{W}, \mathbf{S}_u) - \ell_E^F(\mathbf{W}, \overline{\mathbf{S}}_u)]$$
$$- \log \mathbb{E}_{\overline{\mathbf{W}}, \overline{\mathbf{R}}_u | \mathbf{Z}_{[2n]} = z_{[2n]}} [e^{\lambda(\ell_E^F(\overline{\mathbf{W}}, \mathbf{S}_u) - \ell_E^F(\overline{\mathbf{W}}, \overline{\mathbf{S}}_u))}]. \tag{96}$$

$$I_{z_{[2n]}}(\mathbf{W}; \mathbf{R}_u) \geq \lambda \mathbb{E}_{\mathbf{W}, \mathbf{R}_u | \mathbf{Z}_{[2n]} = z_{[2n]}} [\ell_E^F(\mathbf{W}, \mathbf{S}_u) - \ell_E^F(\mathbf{W}, \overline{\mathbf{S}}_u)]$$
$$- \log \mathbb{E}_{\overline{\mathbf{W}}} \left[ \mathbb{E}_{\overline{\mathbf{R}}_u | \mathbf{Z}_{[2n]} = z_{[2n]}, \overline{\mathbf{W}} = w} [e^{\lambda(\ell_E^F(w, \mathbf{S}_u) - \ell_E^F(w, \overline{\mathbf{S}}_u))}] \right]. \tag{97}$$

Using Lemma 4, we have

$$I_{z_{[2n]}}(\mathbf{W}; \mathbf{R}_u) \geq \lambda \mathbb{E}_{\mathbf{W}, \mathbf{R}_u | \mathbf{Z}_{[2n]} = z_{[2n]}} [\ell_E^F(\mathbf{W}, \mathbf{S}_u) - \ell_E^F(\mathbf{W}, \overline{\mathbf{S}}_u)]$$
$$- \frac{\lambda^2 m}{8} \mathbb{E}[(\frac{1}{n_0^{S_u} + 2} + \frac{1}{n_1^{S_u} + 2} + \frac{1}{n_0^{\overline{S}_u} + 2} + \frac{1}{n_1^{\overline{S}_u} + 2})^2]. \tag{98}$$

So the parabola (function of $\lambda$)

$$I_{z_{[2n]}}(\mathbf{W}; \mathbf{R}_u) - \lambda \mathbb{E}_{\mathbf{W}, \mathbf{R}_u | \mathbf{Z}_{[2n]} = z_{[2n]}}[\ell_E^F(\mathbf{W}, \mathbf{S}_u) - \ell_E^F(\mathbf{W}, \overline{\mathbf{S}}_u)]$$
$$+ \frac{\lambda^2 m}{8} \mathbb{E}[(\frac{1}{n_0^{S_u} + 2} + \frac{1}{n_1^{S_u} + 2} + \frac{1}{n_0^{\overline{S}_u} + 2} + \frac{1}{n_1^{\overline{S}_u} + 2})^2] \geq 0, \quad \forall \lambda \in \mathbb{R} \quad (99)$$

is always positive. Hence, its discriminant is always negative. It follows:

$$\left| \mathbb{E}_{\mathbf{W}, \mathbf{R}_u | \mathbf{Z}_{[2n]} = z_{[2n]}}[\ell_E^F(\mathbf{W}, \mathbf{S}_u) - \ell_E^F(\mathbf{W}, \overline{\mathbf{S}}_u)] \right| \leq \sqrt{\frac{m}{2} \mathbb{E}[(\frac{1}{n_0^{S_u} + 2} + \frac{1}{n_1^{S_u} + 2} + \frac{1}{n_0^{\overline{S}_u} + 2} + \frac{1}{n_1^{\overline{S}_u} + 2})^2] I_{z_{[2n]}}(\mathbf{W}; \mathbf{R}_u)}$$
$$(100)$$

Replacing in (94), we have

$$\overline{gen}_{fairness} \leq \frac{1}{|P_n^m|} \mathbb{E}_{\mathbf{Z}_{[2n]}} \Big[ \sum_{u \in P_n^m} \sqrt{\frac{m}{2} \mathbb{E}_{\mathbf{R}_u}[(\frac{1}{n_0^{S_u} + 2} + \frac{1}{n_1^{S_u} + 2} + \frac{1}{n_0^{\overline{S}_u} + 2} + \frac{1}{n_1^{\overline{S}_u} + 2})^2] I_{z_{[2n]}}(\mathbf{W}; \mathbf{R}_u)} \Big] \quad (101)$$

As mutual information is invariant to permutations, we have the final results. $\qquad \square$

### C.5. Proof of Theorem 3

**Theorem 3** (restated) Assume that $|f| \in [0, 1]$, then for any $m \in \{2, \cdots, n\}$, we have

$$\overline{gen}_{fairness} \leq \frac{1}{|C_n^m|} \mathbb{E}_{\mathbf{Z}_{[2n]}} \Big[ \sum_{u \in C_n^m} \tag{102}$$
$$\sqrt{\frac{m}{2} \mathbb{E}_{\mathbf{R}_u} \Big[ \frac{1}{\mathbb{H}(n_0^{S_u}, n_1^{S_u}, n_0^{\overline{S}_u}, n_1^{\overline{S}_u})^2} \Big] I_{z_{[2n]}}(\mathbf{F}_u; \mathbf{R}_u)} \Big],$$

where $C_n^m$ is the set of $m$-combinations.

*Proof.* We have

$$\overline{gen}_{fairness} = |\mathbb{E}_{P_{\mathbf{W}, \mathbf{Z}_{[2n]}, \mathbf{R}}}[\ell_E^F(\mathbf{W}, \mathbf{S}) - \ell_E^F(\mathbf{W}, \overline{\mathbf{S}})]| \tag{103}$$

$$\leq \mathbb{E}_{P_{\mathbf{Z}_{[2n]}}} |\mathbb{E}_{P_{\mathbf{W}, \mathbf{R} | \mathbf{Z}_{[2n]}}}[\ell_E^F(\mathbf{W}, \mathbf{S}) - \ell_E^F(\mathbf{W}, \overline{\mathbf{S}})]| \tag{104}$$

$$\leq \frac{1}{|P_n^m|} \sum_{u \in P_n^m} \mathbb{E}_{P_{\mathbf{Z}_{[2n]}}} |\mathbb{E}_{P_{\mathbf{W}, \mathbf{R} | \mathbf{Z}_{[2n]}}}[\ell_E^F(\mathbf{W}, \mathbf{S}_u) - \ell_E^F(\mathbf{W}, \overline{\mathbf{S}}_u)| \tag{105}$$

$$= \frac{1}{|P_n^m|} \sum_{u \in P_n^m} \mathbb{E}_{P_{\mathbf{Z}_{[2n]}}} |\mathbb{E}_{P_{\mathbf{W}, \mathbf{R} | \mathbf{Z}_{[2n]}}} \Big[ \Big| \frac{1}{n_0^{\mathbf{S}_u} + 2} \sum_{\mathbf{S}_u, T_i = 0} f(w, x_i) - \frac{1}{n_1^{\mathbf{S}_u} + 2} \sum_{\mathbf{S}_u, T_i = 1} f(w, x_i) \Big| \tag{106}$$

$$- \Big| \frac{1}{n_0^{\overline{\mathbf{S}}} + 2} \sum_{\overline{\mathbf{S}}, T_i = 0} f(w, x_i) - \frac{1}{n_1^{\overline{\mathbf{S}}} + 2} \sum_{\overline{\mathbf{S}}, T_i = 1} f(w, x_i) \Big| \Big] \tag{107}$$

$$= \frac{1}{|P_n^m|} \sum_{u \in P_n^m} \mathbb{E}_{P_{\mathbf{Z}_{[2n]}}} |\mathbb{E}_{P_{\mathbf{F}_u, \mathbf{R} | \mathbf{Z}_{[2n]}}} \Big[ \Big| \frac{1}{n_0^{\mathbf{S}_u} + 2} \sum_{\mathbf{S}_u, T_i = 0} \mathbf{F}_i - \frac{1}{n_1^{\mathbf{S}_u} + 2} \sum_{\mathbf{S}_u, T_i = 1} \mathbf{F}_i \Big| \tag{108}$$

$$- \Big| \frac{1}{n_0^{\overline{\mathbf{S}}} + 2} \sum_{\overline{\mathbf{S}}, T_i = 0} \mathbf{F}_i - \frac{1}{n_1^{\overline{\mathbf{S}}} + 2} \sum_{\overline{\mathbf{S}}, T_i = 1} \overline{\mathbf{F}}_i \Big| \Big] \tag{109}$$

For a fixed $u \in P_n^m$, let $(\overline{\mathbf{F}}_u, \overline{\mathbf{R}}_u)$ be an independent copy of $(\mathbf{F}_u, \mathbf{R}_u)$.

By the Donsker–Varadhan variational representation of KL divergence, $\forall \lambda \in \mathbb{R}$, we have

$$I_{z_{[2n]}}(\mathbf{F}_u; \mathbf{R}_u) \geq \lambda \mathbb{E}_{\mathbf{F}_u, \mathbf{R}_u | \mathbf{Z}_{[2n]} = z_{[2n]}}[\ell_E^F(\mathbf{W}, \mathbf{S}_u) - \ell_E^F(\mathbf{W}, \overline{\mathbf{S}}_u)]$$
$$- \log \mathbb{E}_{\overline{\mathbf{F}_u}, \overline{\mathbf{R}}_u | \mathbf{Z}_{[2n]} = z_{[2n]}}[e^{\lambda(\ell_E^F(\overline{\mathbf{W}}, \mathbf{S}_u) - \ell_E^F(\overline{\mathbf{W}}, \overline{\mathbf{S}}_u))}]. \quad (110)$$

$$I_{z_{[2n]}}(\mathbf{F}_u; \mathbf{R}_u) \geq \lambda \mathbb{E}_{\mathbf{F}_u, \mathbf{R}_u | \mathbf{Z}_{[2n]} = z_{[2n]}}[\ell_E^F(\mathbf{W}, \mathbf{S}_u) - \ell_E^F(\mathbf{W}, \overline{\mathbf{S}}_u)]$$
$$- \log \mathbb{E}_{\overline{\mathbf{F}_u}}\Big[\mathbb{E}_{\overline{\mathbf{R}}_u | \mathbf{Z}_{[2n]} = z_{[2n]}, \overline{\mathbf{F}_u} = F_u}[e^{\lambda(\ell_E^F(w, \mathbf{S}_u) - \ell_E^F(w, \overline{\mathbf{S}}_u))}]\Big]. \quad (111)$$

Using the result of Lemma 4, we have

$$I_{z_{[2n]}}(\mathbf{F}_u; \mathbf{R}_u) \geq \lambda \mathbb{E}_{\mathbf{F}_u, \mathbf{R}_u | \mathbf{Z}_{[2n]} = z_{[2n]}}[\ell_E^F(\mathbf{W}, \mathbf{S}_u) - \ell_E^F(\mathbf{W}, \overline{\mathbf{S}}_u)]$$
$$- \frac{\lambda^2 m}{8} \mathbb{E}[(\frac{1}{n_0^{S_u} + 2} + \frac{1}{n_1^{S_u} + 2} + \frac{1}{n_0^{\overline{S}_u} + 2} + \frac{1}{n_1^{\overline{S}_u} + 2})^2] \quad (112)$$

Hence, the discriminant is negative:

$$\overline{gen}_{fairness} \leq \frac{1}{|P_n^m|} \mathbb{E}_{\mathbf{Z}_{[2n]}}\Big[\sum_{u \in P_n^m} \sqrt{\frac{m}{2} \mathbb{E}_{\mathbf{R}_u}[(\frac{1}{n_0^{S_u} + 2} + \frac{1}{n_1^{S_u} + 2} + \frac{1}{n_0^{\overline{S}_u} + 2} + \frac{1}{n_1^{\overline{S}_u} + 2})^2] I_{z_{[2n]}}(\mathbf{F}_u; \mathbf{R}_u)}\Big] \quad (113)$$

As mutual information is invariant to permutations, we have the final results. $\square$

## C.6. Proof of Theorem 4

**Theorem 4** (restated) Assume that $|f| \in [0, 1]$, then for any $m \in \{2, \cdots, n\}$

$$\overline{gen}_{fairness} \leq \frac{1}{|C_n^m|} \mathbb{E}_{\mathbf{Z}_{[2n]}}\Big[\sum_{u \in C_n^m} \sqrt{\frac{m}{2} \mathbb{E}_{\mathbf{R}_u}[(\frac{1}{n_0^{S_u} + 2} + \frac{1}{n_1^{S_u} + 2} + \frac{1}{n_0^{\overline{S}_u} + 2} + \frac{1}{n_1^{\overline{S}_u} + 2})^2] I_{z_{[2n]}}(\mathbf{L}_u; \mathbf{R}_u)}\Big] \quad (114)$$

where $C_n^m$ is the set of $m$-combinations.

*Proof.* We have

$$\overline{gen}_{fairness} = |\mathbb{E}_{P_{\mathbf{W}, \mathbf{Z}_{[2n]}, \mathbf{R}}}[\ell_E^F(\mathbf{W}, \mathbf{S}) - \ell_E^F(\mathbf{W}, \overline{\mathbf{S}})]| \quad (115)$$

$$\leq \mathbb{E}_{P_{\mathbf{Z}_{[2n]}}} |\mathbb{E}_{P_{\mathbf{W}, \mathbf{R} | \mathbf{Z}_{[2n]}}}[\ell_E^F(\mathbf{W}, \mathbf{S}) - \ell_E^F(\mathbf{W}, \overline{\mathbf{S}})]| \quad (116)$$

$$\leq \frac{1}{|P_n^m|} \sum_{u \in P_n^m} \mathbb{E}_{P_{\mathbf{Z}_{[2n]}}} |\mathbb{E}_{P_{\mathbf{W}, \mathbf{R} | \mathbf{Z}_{[2n]}}}[\ell_E^F(\mathbf{W}, \mathbf{S}_u) - \ell_E^F(\mathbf{W}, \overline{\mathbf{S}}_u)]| \quad (117)$$

$$\leq \frac{1}{|P_n^m|} \sum_{u \in P_n^m} \mathbb{E}_{P_{\mathbf{Z}_{[2n]}}} |\mathbb{E}_{P_{\mathbf{L}_u, \mathbf{R}_u | \mathbf{Z}_{[2n]}}}[\mathbf{L}_u^{\overline{\mathbf{R}}} - \mathbf{L}_u^{\mathbf{R}}]| \quad (118)$$

where $\mathbf{L}_u = (\mathbf{L}_u^{\overline{\mathbf{R}}}, \mathbf{L}_u^{\mathbf{R}}) = (\ell_E^F(\mathbf{W}, \overline{\mathbf{S}}_u), \ell_E^F(\mathbf{W}, \mathbf{S}_u))$. By the Donsker–Varadhan variational representation of KL divergence, $\forall \lambda \in \mathbb{R}$, we have

$$I_{z_{[2n]}}(\mathbf{L}_u; \mathbf{R}_u) \geq \lambda \mathbb{E}_{\mathbf{L}_u, \mathbf{R}_u | \mathbf{Z}_{[2n]} = z_{[2n]}}[\mathbf{L}_u^{\overline{\mathbf{R}}} - \mathbf{L}_u^{\mathbf{R}}]$$
$$- \log \mathbb{E}_{\overline{\mathbf{L}_u}, \overline{\mathbf{R}}_u | \mathbf{Z}_{[2n]} = z_{[2n]}}[e^{\lambda(\overline{\mathbf{L}_u^{\overline{\mathbf{R}}}} - \overline{\mathbf{L}_u^{\mathbf{R}}})}]. \quad (119)$$

$$I_{z_{[2n]}}(\mathbf{L}_u; \mathbf{R}_u) \geq \lambda \mathbb{E}_{\mathbf{L}_u, \mathbf{R}_u | \mathbf{Z}_{[2n]} = z_{[2n]}}[\mathbf{L}_u^{\overline{\mathbf{R}}} - \mathbf{L}_u^{\mathbf{R}}]$$
$$- \log \mathbb{E}_{\overline{\mathbf{W}}}\Big[\mathbb{E}_{\overline{\mathbf{R}}_u | \mathbf{Z}_{[2n]} = z_{[2n]}, \overline{\mathbf{W}} = w}[e^{\lambda(\mathbf{L}_u^{\overline{\mathbf{R}}} - \mathbf{L}_u^{\mathbf{R}})}]\Big]. \quad (120)$$

Using the result of Lemma 4, we have

$$I_{z_{[2n]}}(\mathbf{L}_u; \mathbf{R}_u) \geq \lambda \mathbb{E}_{\mathbf{L}_u, \mathbf{R}_u | \mathbf{Z}_{[2n]} = z_{[2n]}} [\ell_E^F(\mathbf{W}, \mathbf{S}_u) - \ell_E^F(\mathbf{W}, \overline{\mathbf{S}}_u)]$$
$$- \frac{\lambda^2 m}{8} \mathbb{E}[(\frac{1}{n_0^{S_u} + 2} + \frac{1}{n_1^{S_u} + 2} + \frac{1}{n_0^{\overline{S}_u} + 2} + \frac{1}{n_1^{\overline{S}_u} + 2})^2] \tag{121}$$

Hence, the discriminant is negative:

$$\overline{gen}_{fairness} \leq \frac{1}{|P_n^m|} \mathbb{E}_{\mathbf{Z}_{[2n]}} \Big[ \sum_{u \in P_n^m} \sqrt{\frac{m}{2} \mathbb{E}_{\mathbf{R}_u} [(\frac{1}{n_0^{S_u} + 2} + \frac{1}{n_1^{S_u} + 2} + \frac{1}{n_0^{\overline{S}_u} + 2} + \frac{1}{n_1^{\overline{S}_u} + 2})^2] I_{z_{[2n]}}(\mathbf{L}_u; \mathbf{R}_u)} \Big] \tag{122}$$

As mutual information is invariant to permutations, we have the final results.

$\square$

## C.7. Proof of Theorem 5

Let $\mathbf{R}_u = \{\mathbf{R}_{u_i}\}_{i=1}^m \in \mathcal{B}^m$ be the sequence of supersample variables indexed by $u$, and let $\Phi_u = \{R_{u_1} \oplus R_{u_i}\}_{i=2}^m \in \mathcal{B}^{m-1}$, where $\oplus$ denotes the XOR operation. Given a binary value $b \in \{0, 1\}$, we define $b \otimes \Phi_u = (b, \{\Phi_{u_i} \oplus b\}_{i=1}^{m-1}) \in \mathcal{B}^m$. To simplify the notation, we denote $0 \otimes \Phi_u$ and $1 \otimes \Phi_u$ as $\Phi_u^-$ and $\Phi_u^+$, respectively. The notation $L_u^{\Phi_u} = (L^{\Phi_u^-}, L^{\Phi_u^+})$ then represents a pair of losses, while $\Delta^{\Phi_u} L_u = L_u^{\Phi_u^+} - L_u^{\Phi_u^-}$ denotes their difference.

**Theorem 5** (restated) Assume that $|f| \in [0, 1]$, then for any $m \in \{2, \cdots, n\}$

$$\overline{gen}_{fairness} \leq \frac{1}{|C_n^m|} \mathbb{E}_{\mathbf{Z}_{[2n]}} \Big[ \sum_{u \in C_n^m} \sqrt{\frac{m}{2} \mathbb{E}_{\mathbf{R}_u} [(\frac{1}{n_0^{S_u} + 2} + \frac{1}{n_1^{S_u} + 2} + \frac{1}{n_0^{\overline{S}_u} + 2} + \frac{1}{n_1^{\overline{S}_u} + 2})^2] I_{z_{[2n]}}(\Delta \mathbf{L}_u^{\Phi_u}; \mathbf{R}_{u_1})} \Big] \tag{123}$$

where $C_n^m$ is the set of $m$-combinations.

*Proof.* We have

$$\overline{gen}_{fairness} = |\mathbb{E}_{P_{\mathbf{W}, \mathbf{Z}_{[2n]}, \mathbf{R}}} [\ell_E^F(\mathbf{W}, \mathbf{S}) - \ell_E^F(\mathbf{W}, \overline{\mathbf{S}})]| \tag{124}$$

$$\leq \mathbb{E}_{P_{\mathbf{Z}_{[2n]}}} |\mathbb{E}_{P_{\mathbf{W}, \mathbf{R} | \mathbf{Z}_{[2n]}}} [\ell_E^F(\mathbf{W}, \mathbf{S}) - \ell_E^F(\mathbf{W}, \overline{\mathbf{S}})]| \tag{125}$$

$$\leq \frac{1}{|P_n^m|} \sum_{u \in P_n^m} \mathbb{E}_{P_{\mathbf{Z}_{[2n]}}} |\mathbb{E}_{P_{\mathbf{W}, \mathbf{R} | \mathbf{Z}_{[2n]}}} [\ell_E^F(\mathbf{W}, \mathbf{S}_u) - \ell_E^F(\mathbf{W}, \overline{\mathbf{S}}_u)]| \tag{126}$$

$$\leq \frac{1}{|P_n^m|} \sum_{u \in P_n^m} \mathbb{E}_{P_{\mathbf{Z}_{[2n]}}} |\mathbb{E}_{P_{L_u^{\Phi_u}, \mathbf{R}_{u_1} | \mathbf{Z}_{[2n]}}} [(-1)^{\mathbf{R}_{u_1}} (\mathbf{L}_u^{\Phi_u^+} - \mathbf{L}_u^{\Phi_u^-})]| \tag{127}$$

$$\leq \frac{1}{|P_n^m|} \sum_{u \in P_n^m} \mathbb{E}_{P_{\mathbf{Z}_{[2n]}}} |\mathbb{E}_{P_{\Delta \mathbf{L}_u^{\Phi_u}, \mathbf{R}_u | \mathbf{Z}_{[2n]}}} [(-1)^{\mathbf{R}_{u_1}} \Delta \mathbf{L}_u^{\Phi_u}]| \tag{128}$$

By the Donsker–Varadhan variational representation of KL divergence, $\forall \lambda \in \mathbb{R}$, we have

$$I_{z_{[2n]}}(\Delta \mathbf{L}_u^{\Phi_u}; \mathbf{R}_{u_1}) \geq \lambda \mathbb{E}_{\Delta \mathbf{L}_u^{\Phi_u}, \mathbf{R}_{u_1}|\mathbf{Z}_{[2n]}=z_{[2n]}}[(-1)^{\mathbf{R}_{u_1}} \Delta \mathbf{L}_u^{\Phi_u}]$$
$$- \log \mathbb{E}_{\overline{\Delta \mathbf{L}_u^{\Phi_u}}, \overline{\mathbf{R}_{u_1}}|\mathbf{Z}_{[2n]}=z_{[2n]}}[e^{\lambda\left((-1)^{\overline{\mathbf{R}_{u_1}}} \overline{\Delta \mathbf{L}_u^{\Phi_u}}\right)}]. \tag{129}$$

$$I_{z_{[2n]}}(\Delta \mathbf{L}_u^{\Phi_u}; \mathbf{R}_{u_1}) \geq \lambda \mathbb{E}_{\Delta \mathbf{L}_u^{\Phi_u}, \mathbf{R}_{u_1}|\mathbf{Z}_{[2n]}=z_{[2n]}}[(-1)^{\mathbf{R}_{u_1}} \Delta \mathbf{L}_u^{\Phi_u}]$$
$$- \log \mathbb{E}_{\overline{\mathbf{W}}, \overline{\mathbf{R}_u}|\mathbf{Z}_{[2n]}=z_{[2n]}}[e^{\lambda\left(\ell_E^F(\overline{\mathbf{W}}, \mathbf{S}_u) - \ell_E^F(\overline{\mathbf{W}}, \overline{\mathbf{S}}_u)\right)}]. \tag{130}$$

$$I_{z_{[2n]}}(\Delta \mathbf{L}_u^{\Phi_u}; \mathbf{R}_{u_1}) \geq \lambda \mathbb{E}_{\Delta \mathbf{L}_u^{\Phi_u}, \mathbf{R}_{u_1}|\mathbf{Z}_{[2n]}=z_{[2n]}}[(-1)^{\mathbf{R}_{u_1}} \Delta \mathbf{L}_u^{\Phi_u}]$$
$$- \log \mathbb{E}_{\overline{\mathbf{W}}}\left[\mathbb{E}_{\overline{\mathbf{R}}_u|\mathbf{Z}_{[2n]}=z_{[2n]}, \overline{\mathbf{W}}=w}[e^{\lambda(\ell_E^F(w, \mathbf{S}_u) - \ell_E^F(w, \overline{\mathbf{S}}_u))}]\right]. \tag{131}$$

$$\tag{132}$$

Using the result of Lemma 4, we have

$$I_{z_{[2n]}}(\Delta \mathbf{L}_u^{\Phi_u}; \mathbf{R}_{u_1}) \geq \lambda \mathbb{E}_{\Delta \mathbf{L}_u^{\Phi_u}, \mathbf{R}_{u_1}|\mathbf{Z}_{[2n]}=z_{[2n]}}[(-1)^{\mathbf{R}_{u_1}} \Delta \mathbf{L}_u^{\Phi_u}]$$
$$- \frac{\lambda^2 m}{8} \mathbb{E}[(\frac{1}{n_0^{S_u}+2} + \frac{1}{n_1^{S_u}+2} + \frac{1}{n_0^{\overline{S}_u}+2} + \frac{1}{n_1^{\overline{S}_u}+2})^2] \tag{133}$$

Hence, the discriminant is negative:

$$\overline{gen}_{fairness} \leq \frac{1}{|P_n^m|} \mathbb{E}_{\mathbf{Z}_{[2n]}}\left[\sum_{u \in P_n^m} \sqrt{\frac{m}{2} \mathbb{E}_{\mathbf{R}_u}[(\frac{1}{n_0^{S_u}+2} + \frac{1}{n_1^{S_u}+2} + \frac{1}{n_0^{\overline{S}_u}+2} + \frac{1}{n_1^{\overline{S}_u}+2})^2] I_{z_{[2n]}}(\Delta \mathbf{L}_u^{\Phi_u}; \mathbf{R}_{u_1})}\right] \tag{134}$$

As mutual information is invariant to permutations, we have the final results.

$\square$

## D. Proof of the Theorems/Lemmas in Section 5

**EO Generalization Error.** Analogously to the DP setting, we define the population-level separation fairness loss as

$$\ell_P^{F_S}(\mathbf{W}, P_{\mathbf{S}}) = \mathbb{E}_{P_{\mathbf{S}}}\left[\ell_E^{F_S}(\mathbf{W}, \mathbf{S})\right], \tag{135}$$

and the separation-based fairness generalization gap becomes

$$\overline{gen}_{\text{fairness}} = \mathbb{E}_{P_{\mathbf{W},\mathbf{s}}}\left[\ell_P^{F_S}(\mathbf{W}, P_{\mathbf{S}}) - \ell_E^{F_S}(\mathbf{W}, \mathbf{S})\right]. \tag{136}$$

### D.1. Proof of Lemma 5

**Lemma 5** (restated)
Fix $w \in \mathcal{W}$ and a fixed $n_{0,0}^{Z_u}, n_{1,0}^{Z_u}, n_{0,1}^{Z_u}, n_{1,1}^{Z_u}$. Let $g : \mathcal{Z}^m \to \mathbb{R}$ be defined as follows: $g(z_1, \ldots, z_m) = \ell_E^{F_S}(w, z_u)$, where $\ell_E^{F_S}(w, z_u)$ is defined (16). Then, for any $1 \leq i \leq n$, we have

$$\sup_{z_u, \tilde{z}_u^i} \left|g(z_u) - g(\tilde{z}_u^i)\right| \leq \frac{2}{\min_{(t,y)}\left(n_{t,y}^{Z_u}+2\right)}, \tag{137}$$

where $|f(w, x_i)|$ is bounded by 1 and $\min_{(t,y)}(n_{t,y}+2)$ is the smallest subgroup size across both $\mathbf{T}$ and $\mathbf{Y}$.

*Proof.* For $1 \leq i \leq m$, consider the difference:

$$|g(z_u) - g(\tilde{z}_u^i)| = \left|\left|\frac{1}{n_{t=0,y=0}^{Z_u} + 2} \sum_{\substack{T_j=0 \\ Y_j=0}} f(w, x_j) - \frac{1}{n_{t=1,y=0}^{Z_u} + 2} \sum_{\substack{T_j=1 \\ Y_j=0}} f(w, x_j)\right|\right.$$

$$\left.- \left|\frac{1}{n_{t=0,y=0}^{\tilde{Z}_u} + 2} \sum_{\substack{\tilde{T}_j=0 \\ \tilde{Y}_j=0}} f(w, \tilde{x}_j) - \frac{1}{n_{t=1,y=0}^{\tilde{Z}_u} + 2} \sum_{\substack{\tilde{T}_j=1 \\ \tilde{Y}_j=0}} f(w, \tilde{x}_j)\right|\right| \tag{138}$$

$$+ \left|\left|\frac{1}{n_{t=0,y=1}^{Z_u} + 2} \sum_{\substack{T_j=0 \\ Y_j=1}} f(w, x_j) - \frac{1}{n_{t=1,y=1}^{Z_u} + 2} \sum_{\substack{T_j=1 \\ Y_j=1}} f(w, x_j)\right|\right.$$

$$\left.- \left|\frac{1}{n_{t=0,y=1}^{\tilde{Z}_u} + 2} \sum_{\substack{\tilde{T}_j=0 \\ \tilde{Y}_j=1}} f(w, \tilde{x}_j) - \frac{1}{n_{t=1,y=1}^{\tilde{Z}_u} + 2} \sum_{\substack{\tilde{T}_j=1 \\ \tilde{Y}_j=1}} f(w, \tilde{x}_j)\right|\right| \tag{139}$$

$$\leq \left|\frac{1}{n_{t=0,y=0}^{Z_u} + 2} \sum_{\substack{T_j=0 \\ Y_j=0}} f(w, x_j) - \frac{1}{n_{t=1,y=0}^{Z_u} + 2} \sum_{\substack{T_j=1 \\ Y_j=0}} f(w, x_j)\right.$$

$$\left.- \frac{1}{n_{t=0,y=0}^{\tilde{Z}_u} + 2} \sum_{\substack{\tilde{T}_j=0 \\ \tilde{Y}_j=0}} f(w, \tilde{x}_j) + \frac{1}{n_{t=1,y=0}^{\tilde{Z}_u} + 2} \sum_{\substack{\tilde{T}_j=1 \\ \tilde{Y}_j=0}} f(w, \tilde{x}_j)\right| \tag{140}$$

$$+ \left|\frac{1}{n_{t=0,y=1}^{Z_u} + 2} \sum_{\substack{T_j=0 \\ Y_j=1}} f(w, x_j) - \frac{1}{n_{t=1,y=1}^{Z_u} + 2} \sum_{\substack{T_j=1 \\ Y_j=1}} f(w, x_j)\right.$$

$$\left.+ \frac{1}{n_{t=0,y=1}^{\tilde{Z}_u} + 2} \sum_{\substack{\tilde{T}_j=0 \\ \tilde{Y}_j=1}} f(w, \tilde{x}_j) + \frac{1}{n_{t=1,y=1}^{\tilde{Z}_u} + 2} \sum_{\substack{\tilde{T}_j=1 \\ \tilde{Y}_j=1}} f(w, \tilde{x}_j)\right| \tag{141}$$

$$\tag{142}$$

The inequality in (139) follows from the triangle inequality $||a| - |b|| \leq |a - b|$.

Since $\tilde{z}_j = z_j$ for all $j \neq i$, the change affects only the $i$-th component. Based on the values of $t_i$ and $\tilde{t}_i$, we have 8 different subcases within 4 main cases:

**Case 1.1:** $y_i = 0$ **and** $\tilde{y}_i = 1$ **while** $t_i = 0$. Before the change:

$$L_0(z_u) = |\ell_{t=0,0}(z_u) - \ell_{t=1,0}(z_u)|. \tag{143}$$

After the change, $n_{t=0,0}^{Z_u} = n_{t=0,0}^{Z_u} - 1$ and $n_{t=0,1}^{Z_u} = n_{t=0,1}^{Z_u} + 1$:

$$L_0(\tilde{z}_u^i) = \left|\frac{1}{n_{t=0,0}^{Z_u} + 1} \sum_{\substack{T_j=0 \\ Y_j=0 \\ j \neq i}} f(w, x_j) - \ell_{t=1,0}(z_u)\right|, \tag{144}$$

$$L_1(\tilde{z}_u^i) = \left|\frac{1}{n_{t=0,1}^{Z_u} + 3} \left(\sum_{\substack{T_j=0 \\ Y_j=1}} f(w, x_j) + f(w, \tilde{x}_i)\right) - \ell_{t=1,1}(z_u)\right|. \tag{145}$$

The change in $g(z_u)$ is:

$$|g(z_u) - g(\tilde{z}_u^i)| = |L_0(z_u) - L_0(\tilde{z}_u^i)| + |L_1(z_u) - L_1(\tilde{z}_u^i)|$$

$$\leq \left| \frac{1}{n_{t=0,0}^{Z_u} + 2} \sum_{\substack{T_j=0 \\ Y_j=0}} f(w, x_j) - \frac{1}{n_{t=0,0}^{Z_u} + 1} \sum_{\substack{T_j=0 \\ Y_j=0 \\ j \neq i}} f(w, x_j) \right|$$

$$+ \left| \frac{1}{n_{t=0,1}^{Z_u} + 2} \sum_{\substack{T_j=0 \\ Y_j=1}} f(w, x_j) - \frac{1}{n_{t=0,1}^{Z_u} + 3} \left( \sum_{\substack{T_j=0 \\ Y_j=1}} f(w, x_j) + f(w, \tilde{x}_i) \right) \right|$$

$$\leq \frac{1}{n_{t=0,0}^{Z_u} + 2} + \frac{1}{n_{t=0,1}^{Z_u} + 2}. \tag{146}$$

**Case 1.2:** $y_i = 1$ and $\tilde{y}_i = 0$ **while** $t_i = 0$.   This scenario is symmetric to Case 1.1. The bound is identical:

$$|g(z_u) - g(\tilde{z}_u^i)| \leq \frac{1}{n_{t=0,0}^{Z_u} + 2} + \frac{1}{n_{t=0,1}^{Z_u} + 2}. \tag{147}$$

**Case 2.1:** $y_i = 0$ and $\tilde{y}_i = 1$ **while** $t_i = 1$.   Before the change:

$$L_0(z_u) = |\ell_{t=0,0}(z_u) - \ell_{t=1,0}(z_u)|. \tag{148}$$

After the change, $n_{t=1,0}^{Z_u} = n_{t=1,0}^{Z_u} - 1$ and $n_{t=1,1}^{Z_u} = n_{t=1,1}^{Z_u} + 1$:

$$L_0(\tilde{z}_u^i) = \left| \ell_{t=0,0}(z_u) - \frac{1}{n_{t=1,0}^{Z_u} + 1} \sum_{\substack{T_j=1 \\ Y_j=0 \\ j \neq i}} f(w, x_j) \right|, \tag{149}$$

$$L_1(\tilde{z}_u^i) = \left| \frac{1}{n_{t=1,1}^{Z_u} + 3} \left( \sum_{\substack{T_j=1 \\ Y_j=1}} f(w, x_j) + f(w, \tilde{x}_i) \right) - \ell_{t=1,1}(z_u) \right|. \tag{150}$$

The change in $g(z_u)$ is:

$$|g(z_u) - g(\tilde{z}_u^i)| = |L_0(z_u) - L_0(\tilde{z}_u^i)| + |L_1(z_u) - L_1(\tilde{z}_u^i)|$$

$$\leq \left| \frac{1}{n_{t=1,0}^{Z_u} + 2} \sum_{\substack{T_j=1 \\ Y_j=0}} f(w, x_j) - \frac{1}{n_{t=1,0}^{Z_u} + 1} \sum_{\substack{T_j=1 \\ Y_j=0 \\ j \neq i}} f(w, x_j) \right|$$

$$+ \left| \frac{1}{n_{t=1,1}^{Z_u} + 2} \sum_{\substack{T_j=1 \\ Y_j=1}} f(w, x_j) - \frac{1}{n_{t=1,1}^{Z_u} + 3} \left( \sum_{\substack{T_j=1 \\ Y_j=1}} f(w, x_j) + f(w, \tilde{x}_i) \right) \right|$$

$$\leq \frac{1}{n_{t=1,0}^{Z_u} + 2} + \frac{1}{n_{t=1,1}^{Z_u} + 2}. \tag{151}$$

**Case 2.2:** $y_i = 1$ **and** $\tilde{y}_i = 0$ **while** $t_i = 1$. This scenario is symmetric to Case 2.1. The bound is identical:

$$|g(z_u) - g(\tilde{z}_u^i)| \leq \frac{1}{n_{t=1,0}^{Z_u} + 2} + \frac{1}{n_{t=1,1}^{Z_u} + 2}. \tag{152}$$

**Case 3.1:** $t_i = 0$ **and** $\tilde{t}_i = 1$ **while** $y_i = 0$. Before the change:

$$L_0(z_u) = |\ell_{t=0,0}(z_u) - \ell_{t=1,0}(z_u)|. \tag{153}$$

After the change, $n_{t=0,0}^{Z_u} = n_{t=0,0}^{Z_u} - 1$ and $n_{t=1,0}^{Z_u} = n_{t=1,0}^{Z_u} + 1$:

$$L_0(\tilde{z}_u^i) = \left| \frac{1}{n_{t=0,0}^{Z_u} + 1} \sum_{\substack{T_j=0 \\ Y_j=0 \\ j \neq i}} f(w, x_j) - \frac{1}{n_{t=1,0}^{Z_u} + 3} \sum_{\substack{T_j=1 \\ Y_j=0}} f(w, x_j) \right|. \tag{154}$$

The change in $g(z_u)$ is:

$$|g(z_u) - g(\tilde{z}_u^i)| = |L_0(z_u) - L_0(\tilde{z}_u^i)|$$

$$\leq \left| \frac{1}{n_{t=0,0}^{Z_u} + 2} \sum_{\substack{T_j=0 \\ Y_j=0}} f(w, x_j) - \frac{1}{n_{t=0,0}^{Z_u} + 1} \sum_{\substack{T_j=0 \\ Y_j=0 \\ j \neq i}} f(w, x_j) \right|$$

$$+ \left| \frac{1}{n_{t=1,0}^{Z_u} + 2} \sum_{\substack{T_j=1 \\ Y_j=0}} f(w, x_j) - \frac{1}{n_{t=1,0}^{Z_u} + 3} \left( \sum_{\substack{T_j=1 \\ Y_j=0}} f(w, x_j) + f(w, \tilde{x}_i) \right) \right|$$

$$\leq \frac{1}{n_{t=0,0}^{Z_u} + 2} + \frac{1}{n_{t=1,0}^{Z_u} + 2}. \tag{155}$$

**Case 3.2:** $t_i = 1$ **and** $\tilde{t}_i = 0$ **while** $y_i = 0$. This scenario is similar to Case 3.1. The bound is identical:

$$|g(z_u) - g(\tilde{z}_u^i)| \leq \frac{1}{n_{t=0,0}^{Z_u} + 2} + \frac{1}{n_{t=1,0}^{Z_u} + 2}. \tag{156}$$

**Case 4.1:** $t_i = 1$ **and** $\tilde{t}_i = 0$ **while** $y_i = 1$. Before the change:

$$L_1(z_u) = |\ell_{t=0,1}(z_u) - \ell_{t=1,1}(z_u)|. \tag{157}$$

After the change, $n_{t=1,1}^{Z_u} = n_{t=1,1}^{Z_u} - 1$ and $n_{t=0,1}^{Z_u} = n_{t=0,1}^{Z_u} + 1$:

$$L_1(\tilde{z}_u^i) = \left| \frac{1}{n_{t=0,1}^{Z_u} + 3} \left( \sum_{\substack{T_j=0 \\ Y_j=1}} f(w, x_j) + f(w, \tilde{x}_i) \right) - \frac{1}{n_{t=1,1}^{Z_u} + 1} \sum_{\substack{T_j=1 \\ Y_j=1 \\ j \neq i}} f(w, x_j) \right|. \tag{158}$$

The change in $g(z_u)$ is:

$$|g(z_u) - g(\tilde{z}_u^i)| = |L_1(z_u) - L_1(\tilde{z}_u^i)|$$

$$\leq \left| \frac{1}{n_{t=0,1}^{Z_u} + 2} \sum_{\substack{T_j=0 \\ Y_j=1}} f(w, x_j) - \frac{1}{n_{t=0,1}^{Z_u} + 3} \left( \sum_{\substack{T_j=0 \\ Y_j=1}} f(w, x_j) + f(w, \tilde{x}_i) \right) \right|$$

$$+ \left| \frac{1}{n_{t=1,1}^{Z_u} + 2} \sum_{\substack{T_j=1 \\ Y_j=1}} f(w, x_j) - \frac{1}{n_{t=1,1}^{Z_u} + 1} \sum_{\substack{T_j=1 \\ Y_j=1 \\ j \neq i}} f(w, x_j) \right|$$

$$\leq \frac{1}{n_{t=0,1}^{Z_u} + 2} + \frac{1}{n_{t=1,1}^{Z_u} + 2}. \tag{159}$$

**Case 4.2: $t_i = 0$ and $\tilde{t}_i = 1$ while $y_i = 1$.** This scenario is similar to Case 4.1. The bound is identical:

$$|g(z_u) - g(\tilde{z}_u^i)| \leq \frac{1}{n_{t=0,1}^{Z_u} + 2} + \frac{1}{n_{t=1,1}^{Z_u} + 2}. \tag{160}$$

**Case 5: $\tilde{t}_i = t_i$ or $\tilde{y}_i = y_i$.** In this case, the change affects only one group, either in $t$ or $y$, while the other attribute remains unchanged. T For both scenarios, the counts are updated as follows:

$$|g(z_u) - g(\tilde{z}_u^i)| \leq \frac{1}{n_{t,y}^{Z_u} + 2}. \tag{161}$$

**Aggregate the Bounds** From Cases 1.1 to 5, in each scenario where a change occurs, we can write the following:

$$\sup_{z_u \in \mathcal{Z}^m, \tilde{z}_i \in \mathcal{Z}} |g(z_u) - g(\tilde{z}_u^i)| \leq \max_{\substack{(t,y),(t',y') \\ (t,y) \neq (t',y')}} \left( \frac{1}{n_{t,y}^{Z_u} + 2} + \frac{1}{n_{t',y'}^{Z_u} + 2} \right) \tag{162}$$

$$\leq 2 \max_{(t,y)} \frac{1}{n_{t,y}^{Z_u} + 2} \tag{163}$$

$$= \frac{2}{\min_{(t,y)} n_{t,y}^{Z_u} + 2} \tag{164}$$

This completes the proof.

$\square$

### D.2. Proof of Theorem 6

**Theorem 6** (restated) Assume that $|f| \in [0, 1]$, then for any $m \in \{2, \cdots, n\}$

$$\overline{gen}_{\text{fairness}} \leq \frac{1}{|P_n^m|} \sum_{u \in P_n^m} \sqrt{2m \mathbb{E}[(\frac{1}{\min_{(t,y)}(n_{t,y} + 2)})^2] I(\mathbf{W}, \mathbf{Z}_u)} \tag{165}$$

And the label with fewer samples will typically dominate the maximum variance, so the upper bound depends not only on the expectation over the sensitive attribute but also on the number of samples for each label.

$$\overline{gen}_{fairness} \triangleq \mathbb{E}_{P_{\mathbf{W},\mathbf{S}}}[\ell_P^F(\mathbf{W}, P_{\mathbf{S}}) - \ell_E^F(\mathbf{W}, \mathbf{S})] \tag{166}$$

*Proof.*

$$\text{Var}(\ell_Y(W, V_{\text{aug}}|Y)) \leq \frac{m}{4}\mathbb{E}[\max_{\substack{t,y,t',y' \\ (t,y)\neq(t',y')}} (\frac{1}{n_{t=t,Y=y}^{Z_u}+2} + \frac{1}{n_{t=t',Y=y'}^{Z_u}+2})^2] \tag{167}$$

$$\overline{gen}_{fairness} == \left| \mathbb{E}_{P_W \otimes P_S}[\ell_E^{F_S}(W, V^{\text{aug}}|Y)] - \mathbb{E}_{P_{W,S}}[\ell_E^{F_S}(W, V^{\text{aug}}|Y)] \right| \tag{168}$$

$$= \left| \mathbb{E}_{P_{\overline{W},\overline{Z}_u}}[\ell_E^{F_S}(\overline{W}, \overline{Z}_u)] - \frac{1}{|P_n^m|}\sum_{u \in P_n^m} \mathbb{E}_{P_{W,Z_u}}[\ell_E^{F_S}(W, Z_u)] \right| \tag{169}$$

$$\leq \frac{1}{|P_n^m|}\sum_{u \in P_n^m} \left| \mathbb{E}_{P_{\overline{W},\overline{Z}_u}}[\ell_E^{F_S}(\overline{W}, \overline{Z}_u)] - \mathbb{E}_{P_{W,Z_u}}[\ell_E^{F_S}(W, Z_u)] \right| \tag{170}$$

$$\leq \frac{1}{|P_n^m|}\sum_{u \in P_n^m} \sqrt{\frac{m}{2}\mathbb{E}[\max_{\substack{t,y,t',y' \\ (t,y)\neq(t',y')}} (\frac{1}{n_{t=t,Y=y}^{Z_u}+2} + \frac{1}{n_{t=t',Y=y'}^{Z_u}+2})^2]I(\mathbf{W}, \mathbf{Z}_u)} \tag{171}$$

$$\leq \frac{1}{|P_n^m|}\sum_{u \in P_n^m} \sqrt{2m\mathbb{E}[(\frac{1}{\min_{(t,y)} n_{t,y}+2})^2]I(\mathbf{W}, \mathbf{Z}_u)} \tag{172}$$

In addition, as mutual information is invariant to sample permutations, we have

$$\overline{gen}_{fairness} \leq \frac{1}{|C_n^m|}\sum_{u \in C_n^m} \sqrt{2m\mathbb{E}[(\frac{1}{\min_{(t,y)} n_{t,y}+2})^2]I(\mathbf{W}, \mathbf{Z}_u)} \tag{173}$$

where $C_n^m$ is the set of $m$-combinations.

And we have $a = 1$ for the binary label case. This completes the proof.

$\square$

**Lemma 8.** *Assume that $|f| \in [0, 1]$, then for any $m \in \{2, \cdots, n\}$*

$$\overline{gen}_{\text{fairness}} \leq \frac{1}{|P_n^m|}\mathbb{E}_{\mathbf{Z}_{[2n]}}\left[ \sum_{u \in P_n^m} \sqrt{2m\mathbb{E}_{\mathbf{R}_u}[(\frac{1}{\min_{(t,y)} n_{t,y}^{Z_u}+2} + \frac{1}{\min_{(t,y)} n_{t,y}^{\overline{Z}_u}+2})^2]I_{z_{[2n]}}(\mathbf{W}; \mathbf{R}_u)} \right] \tag{174}$$

Similar to the result of Lemma 4, by applying the separation loss to the variance bound, we obtain:

$$\text{Var}\left(\ell_E^{F_S}(w, \mathbf{S}_u) - \ell_E^{F_S}(w, \overline{\mathbf{S}}_u)\right) \leq \frac{m}{4}\mathbb{E}\left[\left(\frac{2}{\min_{(t,y)} n_{t,y}^{S_u}+2} + \frac{2}{\min_{(t,y)} n_{t,y}^{\overline{S}_u}+2}\right)^2\right] \tag{175}$$

Finally, using Hoeffding's Lemma, we get:

$$\mathbb{E}\left[e^{\lambda\left(\ell_E^{F_S}(w,\mathbf{S}_u)-\ell_E^{F_S}(w,\overline{\mathbf{S}}_u)\right)}\right] \leq e^{\frac{\lambda^2 m}{8}\mathbb{E}\left[\left(\frac{2}{\min_{(t,y)} n_{t,y}^{S_u}+2}+\frac{2}{\min_{(t,y)} n_{t,y}^{\overline{S}_u}+2}\right)^2\right]} \tag{176}$$

For a fixed $u \in P_n^m$, let $(\overline{\mathbf{W}}, \overline{\mathbf{R}}_u)$ be an independent copy of $(\mathbf{W}, \mathbf{R}_u)$. The disintegrated mutual information $I_{z_{[2n]}}(\mathbf{W}, \mathbf{R}_u)$ is equal to:

$$I_{z_{[2n]}}(\mathbf{W}, \mathbf{R}_u) = D\left(P_{\mathbf{W},\mathbf{R}_u|\mathbf{Z}_{[2n]}=z_{[2n]}} \| P_{\mathbf{W}|\mathbf{Z}_{[2n]}=z_{[2n]}} P_{\mathbf{R}_u}\right), \tag{177}$$

And here we only change the independence loss function to the separation-based loss

$$\overline{gen}_{fairness} = |\mathbb{E}_{P_{\mathbf{W},\mathbf{Z}_{[2n]},\mathbf{R}}}[\ell_E^{F_S}(\mathbf{W}, \mathbf{S}) - \ell_E^{F_S}(\mathbf{W}, \overline{\mathbf{S}})]| \tag{178}$$

$$\leq \mathbb{E}_{P_{\mathbf{Z}_{[2n]}}}|\mathbb{E}_{P_{\mathbf{W},\mathbf{R}|\mathbf{Z}_{[2n]}}}[\ell_E^{F_S}(\mathbf{W}, \mathbf{S}) - \ell_E^{F_S}(\mathbf{W}, \overline{\mathbf{S}})]| \tag{179}$$

$$\leq \frac{1}{|P_n^m|}\sum_{u \in P_n^m} \mathbb{E}_{P_{\mathbf{Z}_{[2n]}}}|\mathbb{E}_{P_{\mathbf{W},\mathbf{R}|\mathbf{Z}_{[2n]}}}[\ell_E^{F_S}(\mathbf{W}, \mathbf{S}_u) - \ell_E^{F_S}(\mathbf{W}, \overline{\mathbf{S}}_u)]| \tag{180}$$

Similar the proof in Theorem 2, we can have

$$\overline{gen}_{\text{fairness}} \leq \frac{1}{|P_n^m|} \mathbb{E}_{\mathbf{Z}_{[2n]}} \Big[ \sum_{u \in P_n^m} \sqrt{2m\mathbb{E}_{\mathbf{R}_u}[(\frac{1}{\min\limits_{(t,y)} n_{t,y}^{S_u} + 2} + \frac{1}{\min\limits_{(t,y)} n_{t,y}^{\overline{S}_u} + 2})^2] I_{z_{[2n]}}(\mathbf{W}; \mathbf{R}_u)} \Big] \tag{181}$$

As mutual information is invariant to permutations, we have the final results.

### D.3. Proof of Theorem 7

**Theorem 7** (restated) Assume that $|f| \in [0,1]$, then for any $m \in \{2, \cdots, n\}$

$$\overline{gen}_{fairness} \leq \frac{1}{|C_n^m|} \mathbb{E}_{\mathbf{Z}_{[2n]}} \Big[ \sum_{u \in C_n^m} \sqrt{2m\mathbb{E}_{\mathbf{R}_u}[(\frac{1}{\min_{(t,y)} n_{t,y}^{S_u} + 2} + \frac{1}{\min_{(t,y)} n_{t,y}^{\overline{S}_u} + 2})^2] I_{z_{[2n]}}(\Delta \mathbf{L}_u^{\Phi_u}; \mathbf{R}_{u_1})} \Big] \tag{182}$$

*Proof.* We have

$$\overline{gen}_{fairness} = |\mathbb{E}_{P_{\mathbf{W}, \mathbf{Z}_{[2n]}, \mathbf{R}}}[\ell_E^F(\mathbf{W}, \mathbf{S}) - \ell_E^F(\mathbf{W}, \overline{\mathbf{S}})]| \tag{183}$$

$$\leq \mathbb{E}_{P_{\mathbf{Z}_{[2n]}}} |\mathbb{E}_{P_{\mathbf{W}, \mathbf{R}|\mathbf{Z}_{[2n]}}}[\ell_E^F(\mathbf{W}, \mathbf{S}) - \ell_E^F(\mathbf{W}, \overline{\mathbf{S}})]| \tag{184}$$

$$\leq \frac{1}{|P_n^m|} \sum_{u \in P_n^m} \mathbb{E}_{P_{\mathbf{Z}_{[2n]}}} |\mathbb{E}_{P_{\mathbf{W}, \mathbf{R}|\mathbf{Z}_{[2n]}}}[\ell_E^F(\mathbf{W}, \mathbf{S}_u) - \ell_E^F(\mathbf{W}, \overline{\mathbf{S}}_u)]| \tag{185}$$

$$\leq \frac{1}{|P_n^m|} \sum_{u \in P_n^m} \mathbb{E}_{P_{\mathbf{Z}_{[2n]}}} |\mathbb{E}_{P_{L_u^{\Phi_u}, \mathbf{R}_{u_1}|\mathbf{Z}_{[2n]}}}[(-1)^{\mathbf{R}_{u_1}} (\mathbf{L}_u^{\Phi_u^+} - \mathbf{L}_u^{\Phi_u^-})]| \tag{186}$$

$$\leq \frac{1}{|P_n^m|} \sum_{u \in P_n^m} \mathbb{E}_{P_{\mathbf{Z}_{[2n]}}} |\mathbb{E}_{P_{\Delta \mathbf{L}_u^{\Phi_u}, \mathbf{R}_u|\mathbf{Z}_{[2n]}}}[(-1)^{\mathbf{R}_{u_1}} \Delta \mathbf{L}_u^{\Phi_u}]| \tag{187}$$

By the Donsker–Varadhan variational representation of KL divergence, $\forall \lambda \in \mathbb{R}$, we have

$$I_{z_{[2n]}}(\Delta \mathbf{L}_u^{\Phi_u}; \mathbf{R}_{u_1}) \geq \lambda \mathbb{E}_{\Delta \mathbf{L}_u^{\Phi_u}, \mathbf{R}_{u_1}|\mathbf{Z}_{[2n]}=z_{[2n]}}[(-1)^{\mathbf{R}_{u_1}} \Delta \mathbf{L}_u^{\Phi_u}]$$
$$- \log \mathbb{E}_{\overline{\Delta \mathbf{L}_u^{\Phi_u}}, \overline{\mathbf{R}_{u_1}}|\mathbf{Z}_{[2n]}=z_{[2n]}}[e^{\lambda((-1)^{\overline{\mathbf{R}_{u_1}}} \overline{\Delta \mathbf{L}_u^{\Phi_u}})}]. \tag{188}$$

$$I_{z_{[2n]}}(\Delta \mathbf{L}_u^{\Phi_u}; \mathbf{R}_{u_1}) \geq \lambda \mathbb{E}_{\Delta \mathbf{L}_u^{\Phi_u}, \mathbf{R}_{u_1}|\mathbf{Z}_{[2n]}=z_{[2n]}}[(-1)^{\mathbf{R}_{u_1}} \Delta \mathbf{L}_u^{\Phi_u}]$$
$$- \log \mathbb{E}_{\overline{\mathbf{W}}, \overline{\mathbf{R}_u}|\mathbf{Z}_{[2n]}=z_{[2n]}}[e^{\lambda(\ell_E^F(\overline{\mathbf{W}}, \mathbf{S}_u) - \ell_E^F(\overline{\mathbf{W}}, \overline{\mathbf{S}}_u))}]. \tag{189}$$

$$I_{z_{[2n]}}(\Delta \mathbf{L}_u^{\Phi_u}; \mathbf{R}_{u_1}) \geq \lambda \mathbb{E}_{\Delta \mathbf{L}_u^{\Phi_u}, \mathbf{R}_{u_1}|\mathbf{Z}_{[2n]}=z_{[2n]}}[(-1)^{\mathbf{R}_{u_1}} \Delta \mathbf{L}_u^{\Phi_u}]$$
$$- \log \mathbb{E}_{\overline{\mathbf{W}}} \Big[ \mathbb{E}_{\overline{\mathbf{R}_u}|\mathbf{Z}_{[2n]}=z_{[2n]}, \overline{\mathbf{W}}=w}[e^{\lambda(\ell_E^F(w, \mathbf{S}_u) - \ell_E^F(w, \overline{\mathbf{S}}_u))}] \Big]. \tag{190}$$

$$\tag{191}$$

Using the result of Lemma 8, we have

$$I_{z_{[2n]}}(\Delta \mathbf{L}_u^{\Phi_u}; \mathbf{R}_{u_1}) \geq \lambda \mathbb{E}_{\Delta \mathbf{L}_u^{\Phi_u}, \mathbf{R}_{u_1}|\mathbf{Z}_{[2n]}=z_{[2n]}}[(-1)^{\mathbf{R}_{u_1}} \Delta \mathbf{L}_u^{\Phi_u}]$$
$$- \frac{\lambda^2 m}{8} \mathbb{E}_{\mathbf{R}_u}[(\frac{1}{\min\limits_{(t,y)} n_{t,y}^{S_u} + 2} + \frac{1}{\min\limits_{(t,y)} n_{t,y}^{\overline{S}_u} + 2})^2] \tag{192}$$

Hence, the discriminant is negative:

$$\overline{gen}_{fairness} \leq \frac{1}{|P_n^m|} \mathbb{E}_{\mathbf{Z}_{[2n]}} \Big[ \sum_{u \in P_n^m} \sqrt{2\,m\,a^2 \mathbb{E}_{\mathbf{R}_u}[(\frac{1}{\min\limits_{(t,y)} n_{t,y}^{S_u} + 2} + \frac{1}{\min\limits_{(t,y)} n_{t,y}^{\overline{S}_u} + 2})^2] I_{z_{[2n]}}(\Delta \mathbf{L}_u^{\Phi_u}; \mathbf{R}_{u_1})} \Big] \tag{193}$$

As mutual information is invariant to permutations and we have $a = 1$ for the binary label case, we have the final results.

$$\square$$

## E. Bounds based on prior techniques

**Lemma 9.** *Assume that $|f| \in [0, 1]$, then*

$$\overline{gen}_{fairness} \leq \sqrt{2I(\mathbf{W}, \mathbf{V}_u)} \tag{194}$$

*Proof.* Using the Donsker–Varadhan variational representation of the relative entropy, we have

$$I(\mathbf{W}, \mathbf{V}_u) \geq \mathbb{E}_{P_{\mathbf{W},\mathbf{S}}}[\lambda \ell_E(\mathbf{W}, \mathbf{V}_u)] - \log \mathbb{E}_{P_{\overline{\mathbf{W}}} \otimes P_{\overline{\mathbf{V}_u}}}[e^{\lambda \ell_E(\mathbf{W}, \mathbf{V}_u)}], \forall \lambda \in \mathbb{R}. \tag{195}$$

We have $|f| \in [0, 1]$, Thus

$$\ell_E(w, S) = \left| \frac{1}{n_0} \sum_{T_i=0} f(w, x_i) - \frac{1}{n_1} \sum_{T_i=1} f(w, x_i) \right| \leq \max \left( \left| \frac{1}{n_0} \sum_{T_i=0} f(w, x_i) \right|, \left| \frac{1}{n_1} \sum_{T_i=1} f(w, x_i) \right| \right) \tag{196}$$

$$\leq \max(1, 1) = 1 \tag{197}$$

So $F_E(w, X'_u) \in [0, 1]$ is bounded. Thus, using Hoeffding's lemma, we have:

$$\log \mathbb{E}_{P_{\overline{\mathbf{W}}} \otimes P_{\overline{\mathbf{V}_u}}}[e^{\lambda \ell_E(\mathbf{W}, \mathbf{V}_u)}] \leq \lambda \mathbb{E}_{P_{\overline{\mathbf{W}}} \otimes P_{\overline{\mathbf{V}_u}}}[\ell_E(\mathbf{W}, \mathbf{V}_u)] + \lambda^2 \frac{1}{2} \tag{198}$$

By replacing in 195, we have:

$$I(\mathbf{W}, \mathbf{V}_u) - \lambda(\mathbb{E}_{P_{\mathbf{W},\mathbf{V}_u}}[\ell_E(\mathbf{W}, \mathbf{V}_u)] - \mathbb{E}_{P_{\overline{\mathbf{W}}} \otimes P_{\overline{\mathbf{V}_u}}}[\ell_E(\mathbf{W}, \mathbf{V}_u)]) + \lambda^2 \frac{1}{2} \geq 0 \forall \lambda \in \mathbb{R}. \tag{199}$$

Thus, (199) must have a non-positive discriminant.

$$|\mathbb{E}_{P_{\mathbf{W},\mathbf{V}_u}}[\ell_E(\mathbf{W}, \mathbf{V}_u)] - \mathbb{E}_{P_{\overline{\mathbf{W}}} \otimes P_{\overline{\mathbf{V}_u}}}[\ell_E(\mathbf{W}, \mathbf{V}_u)]| \leq \sqrt{2I(\mathbf{W}, \mathbf{V}_u)} \tag{200}$$

$\square$

### E.1. Proof of Lemma 1

**Lemma 1** (restated) Assume that $|f| \in [0, 1]$, then for any $m \in \{2, \cdots, n\}$

$$\overline{gen}_{fairness} \leq \frac{1}{|C_n^m|} \sum_{u \in C_n^m} \sqrt{2I(\mathbf{W}; \mathbf{V}_u)} \tag{201}$$

*Proof.*

$$\overline{gen}_{fairness} = \mathbb{E}_{P_{\mathbf{W}} \otimes P_{\mathbf{S}}}[\ell_E^F(\mathbf{W}, \mathbf{S})] - \mathbb{E}_{P_{\mathbf{W},\mathbf{S}}}[\ell_E^F(\mathbf{W}, \mathbf{S})] \tag{202}$$

Let $m \in \{2, \cdots, n\}$ be a fixed number. Given the set $Z_{1:m} \sim \mu^m$, the corresponding $\ell_E^F(w, Z_{1:m})$ is as follows:

$$\ell_E^F(w, Z_{1:m}) \triangleq \left| \frac{1}{n_0^{Z_{1:m}}} \sum_{T_i=0} f(w, x_i) - \frac{1}{n_1^{Z_{1:m}}} \sum_{T_i=1} f(w, x_i) \right| \tag{203}$$

Given a sequence of indices $u = \{u_i\}_{i=1}^m \in [1, n]^m$, let $Z_u = \{Z_{u_i}\}_{i=1}^m$ be the sequence of training samples indexed by $u$. Let $P_n^m$ be the set of $m$-permutations of $n$. Then, using the notion of $\ell_E^F(\mathbf{W}, \mathbf{Z}_{1:m})$ we have:

$$\overline{gen}_{fairness} = |\mathbb{E}_{P_{\mathbf{W}} \otimes P_{\mathbf{S}}}[\ell_E^F(\mathbf{W}, \mathbf{S})] - \mathbb{E}_{P_{\mathbf{W},\mathbf{S}}}[\ell_E^F(\mathbf{W}, \mathbf{S})]| \tag{204}$$

$$= \left| \mathbb{E}_{P_{\mathbf{W},\overline{\mathbf{z}}_{1:m}}}[\ell_E^F(\mathbf{W}, \overline{\mathbf{Z}}_{1:m})] - \frac{1}{|P_n^m|} \sum_{u \in P_n^m} \mathbb{E}_{P_{\mathbf{W},\mathbf{z}_u}}[\ell_E^F(\mathbf{W}, \mathbf{Z}_u)] \right| \tag{205}$$

$$\leq \frac{1}{|P_n^m|} \sum_{u \in P_n^m} \left| \mathbb{E}_{P_{\mathbf{W},\overline{\mathbf{z}}_{1:m}}}[\ell_E^F(\mathbf{W}, \overline{\mathbf{Z}}_{1:m})] - \mathbb{E}_{P_{\mathbf{W},\mathbf{z}_u}}[\ell_E^F(\mathbf{W}, \mathbf{Z}_u)] \right| \tag{206}$$

Assuming that $|f| \in [0, 1]$, Similar to Lemma 9, we can show that

$$\mathbb{E}_{P_{\mathbf{W}, \overline{\mathbf{Z}}_{1:m}}}[\ell_E^F(\mathbf{W}, \overline{\mathbf{Z}}_{1:m})] - \mathbb{E}_{P_{\mathbf{W}, \mathbf{Z}_u}}[\ell_E^F(\mathbf{W}, \mathbf{Z}_u)] \leq \sqrt{2I(\mathbf{W}; \mathbf{V}_u)} \tag{207}$$

where $\mathbf{V}_u = (\mathbf{X}_u, \mathbf{T}_u)$

By plugging (207) into (206), we have

$$\overline{gen}_{fairness} \leq \frac{1}{|P_n^m|} \sum_{u \in P_n^m} \left| \mathbb{E}_{P_{\mathbf{W}, \overline{\mathbf{Z}}_{1:m}}}[\ell_E^F(\mathbf{W}, \overline{\mathbf{Z}}_{1:m})] - \mathbb{E}_{P_{\mathbf{W}, \mathbf{z}_u}}[\ell_E^F(\mathbf{W}, \mathbf{Z}_u)] \right| \tag{208}$$

$$\leq \frac{1}{|P_n^m|} \sum_{u \in P_n^m} \sqrt{2I(\mathbf{W}; \mathbf{V}_u)} \tag{209}$$

However, as mutual information is invariant to sample permutations, we have

$$\overline{gen}_{fairness} \leq \frac{1}{|C_n^m|} \sum_{u \in C_n^m} \sqrt{2I(\mathbf{W}; \mathbf{V}_u)}, \tag{210}$$

where $C_n^m$ is the set of $m$-combinations.

$\square$

# F. Additional Empirical results

### F.1. Experimental details

In all the experiments in this paper, similar to Harutyunyan et al. (2021), for every number of training data $n$, we randomly sample $m_1 = 21$ different $2n$ samples from the original dataset. Next, for each $z_{[2n]}$, we draw $m_2 = 50$ different train/test splits, i.e., $m_2$ random realizations of $\mathbf{R}$. Hence, each data point in the figures corresponds to a total of $m_1 m_2 = 1050$ experiments. We report the mean and standard deviation on the $m_1$ results.

We conducted experiments using the following datasets:

- **COMPAS** (Larson et al., 2016): The COMPAS dataset contains records of criminal defendants and is used to predict whether the defendant will recidivate within two years. The dataset includes attributes such as criminal history and demographic information, including race and gender.

- **Adult** (Kohavi & Becker, 1996): The Adult dataset is based on the 1994 U.S. Census and is widely used in machine learning research. The goal is to predict whether an individual earns more than $50K annually based on demographic and financial data. In fairness studies, gender is often used as the sensitive (binary) attribute.

The following fairness methods were evaluated in our experiments:

- **ERM**: Empirical Risk Minimization (ERM) is a standard machine learning approach that minimizes the empirical risk on the training data. It serves as a baseline for fairness methods.

- **DiffDP, DiffEopp, DiffEodd** (Mroueh et al., 2021): These methods apply gap regularization for demographic parity, equalized opportunity, and equalized odds, respectively. Since these fairness metrics cannot be optimized directly, gap regularization provides a differentiable alternative loss that can be optimized using gradient descent.

- **PRemover (PrejudiceRemover)** (Kamishima et al., 2012): This method minimizes the mutual information between prediction accuracy and sensitive attributes.

- **HSIC** (Baharlouei et al., 2020; Li et al., 2022): The Hilbert-Schmidt Independence Criterion (HSIC) is minimized to reduce the dependence between prediction accuracy and sensitive attributes.

### F.2. Extra Numerical results for our bound in Theorems 5

We report the experimental results of fairness generalization error and our bounds in Theorem 5 (DP) and Theorem 7 (EO) as a function of the the total number of training samples $n$ on with COMPAS dataset with gender and race as sensitive attribute in Figure 3 and Figure 4, respectively. In Figure 5 and Figure 6, we report the results of our bound in Theorem 5 (DP) on Adult with Gender and Race as sensitive attributes, respectively.

The CMI terms in Theorems 5 (DP) and Theorem 7 (EO) involve both a continuous and a discrete variable. We evaluate different estimators for these terms. Specifically, in addition to the main estimator used in this paper, i.e., Ross (2014), we experiment with the estimators proposed in Gao et al. (2017), Darbellay & Vajda (1999), and Kraskov et al. (2004). The main results of these comparisons are presented in Figure 7, showing that Ross (2014) is the most robust estimator in our case.

### F.3. Extra Numerical results for batch balancing

In Table 2, we report the results for our batch balancing approach on the Adult dataset. The results are consistent with Table 1, highlighting the strength of our approach and its ability to improve fairness test errors of different algorithms.

In Figures 8, 9, and 10, we report the mean of test accuracy, the mean of fairness generalization error, and the mean of fairness test errors of the different approaches using COMPAS (gender) dataset with different number of training samples $n$.

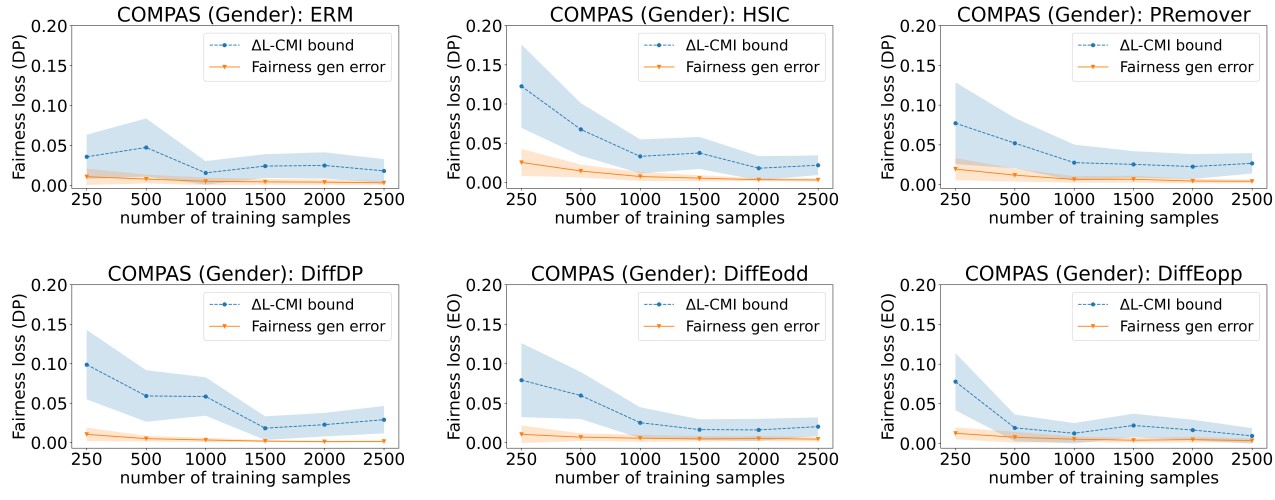

*Figure 3.* Experimental results with COMPAS dataset (gender as sensitive attribute) of fairness generalization error and our bounds in Theorems 5 (DP) and Theorem 7 (EO) as a function of the total number of training samples $n$.

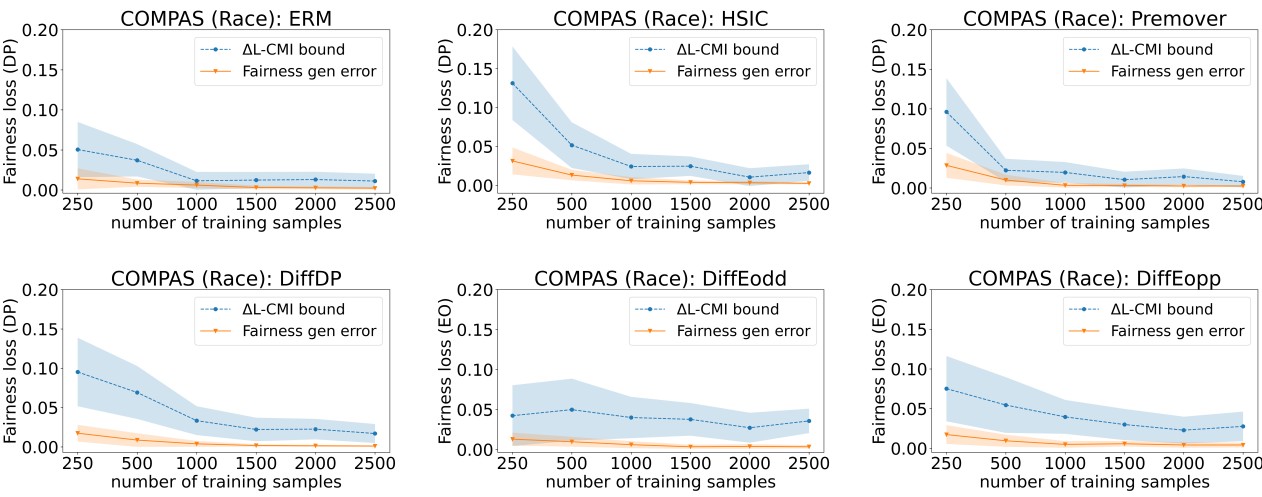

*Figure 4.* Experimental results with COMPAS dataset (Race as sensitive attribute) of fairness generalization error and our bounds in Theorems 5 (DP) and Theorem 7 (EO) as a function of the total number of training samples $n$.

*Table 2.* Effect of batch balancing on DP errors with the Adult dataset. We report DP on test data for various approaches, with and without our proposed batch-balancing strategy. The results show that balancing consistently reduces DP by an order of magnitude. This empirically supports the theoretical insights of Theorems 1- 5 regarding the role of group imbalance in fairness generalization error.

| | sensitive attribute: gender | | | | sensitive attribute: race | | | |
| --- | --- | --- | --- | --- | --- | --- | --- | --- |
| | 7500 | 10000 | 15000 | 20000 | 7500 | 10000 | 15000 | 20000 |
| DiffDP | 0.038 | 0.037 | 0.039 | 0.038 | 0.083 | 0.083 | 0.084 | 0.084 |
| DiffDP (ours) | 0.020 | 0.020 | 0.021 | 0.020 | 0.063 | 0.063 | 0.064 | 0.063 |
| HSIC | 0.026 | 0.025 | 0.025 | 0.026 | 0.087 | 0.086 | 0.086 | 0.086 |
| HSIC (ours) | 0.009 | 0.005 | 0.007 | 0.007 | 0.026 | 0.023 | 0.019 | 0.0178 |
| PRremover | 0.027 | 0.026 | 0.026 | 0.026 | 0.035 | 0.033 | 0.032 | 0.031 |
| PRremover (ours) | 0.008 | 0.007 | 0.007 | 0.007 | 0.011 | 0.009 | 0.007 | 0.006 |

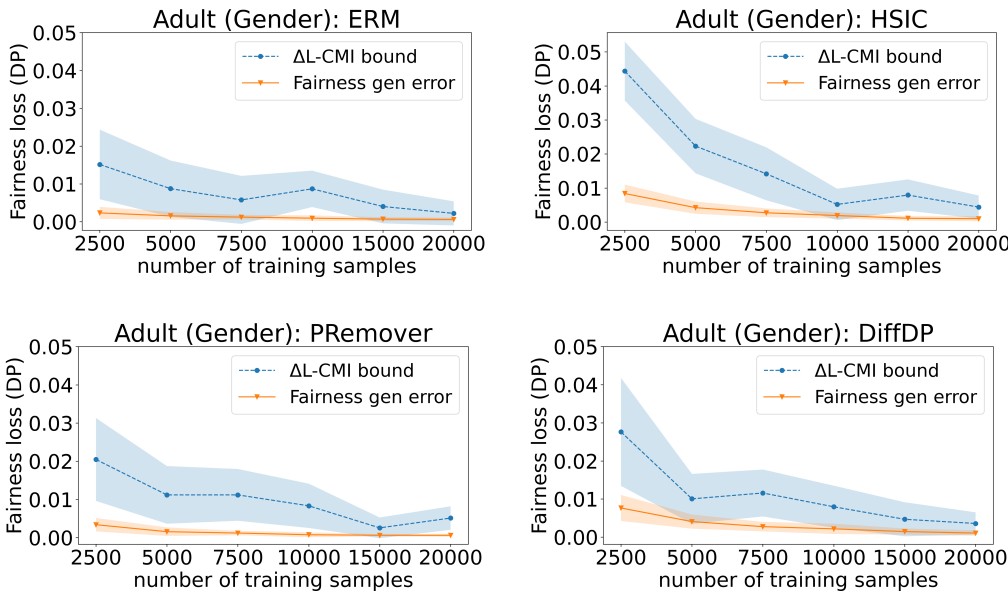

*Figure 5.* Experimental results with Adult dataset (Gender as sensitive attribute) of fairness generalization error and our bound in Theorem 5 (DP) as a function of the total number of training samples $n$.

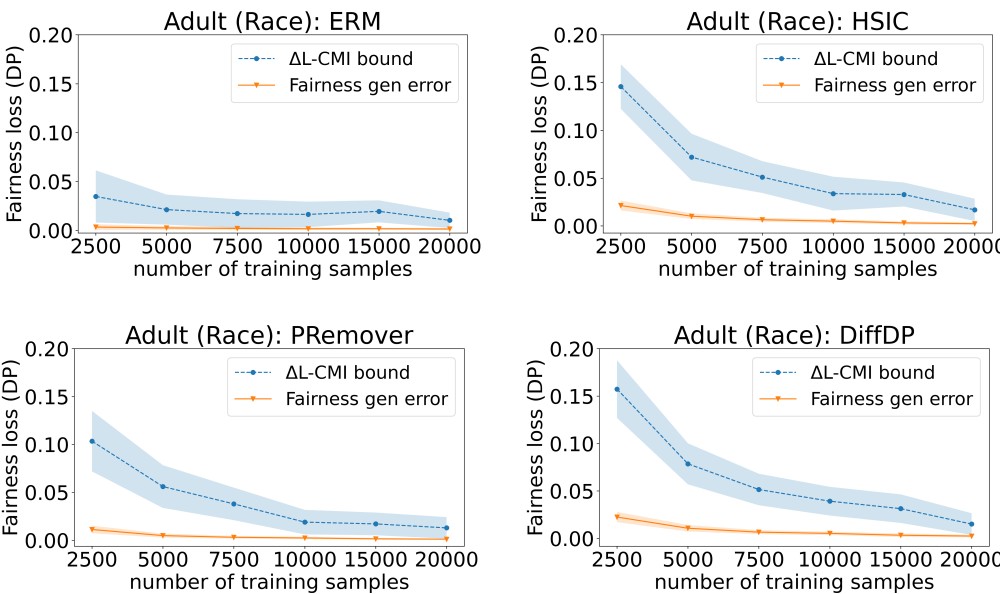

*Figure 6.* Experimental results with Adult dataset (Race as sensitive attribute) of fairness generalization error and our bound in Theorem 5 (DP) as a function of the total number of training samples $n$.

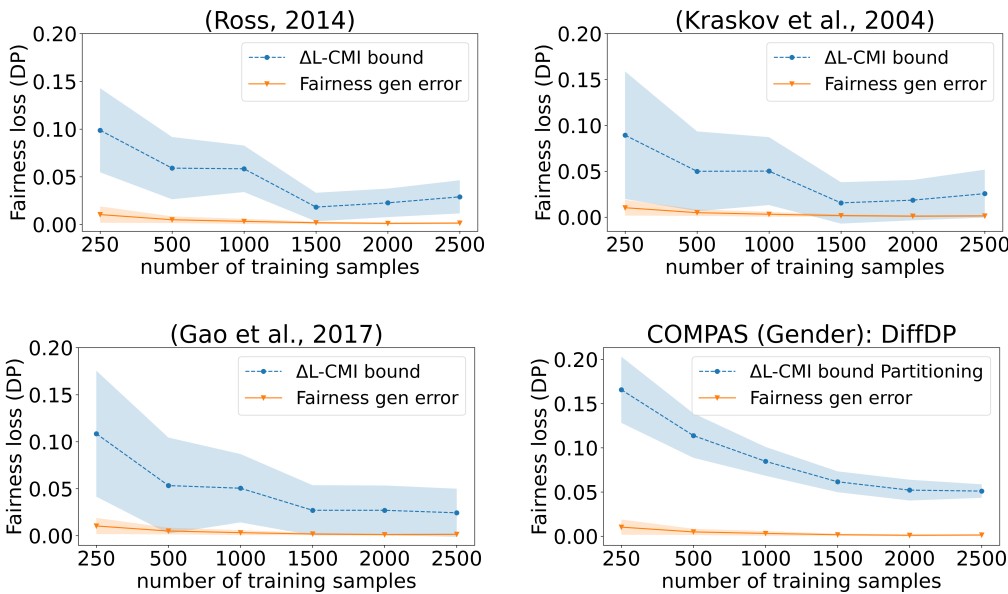

*Figure 7.* Experimental results of our bound Theorem 5 (DP) with different MI estimators using the DiffDP approach on the COMPAS dataset (gender as sensitive attribute) as a function of the total number of training samples $n$.

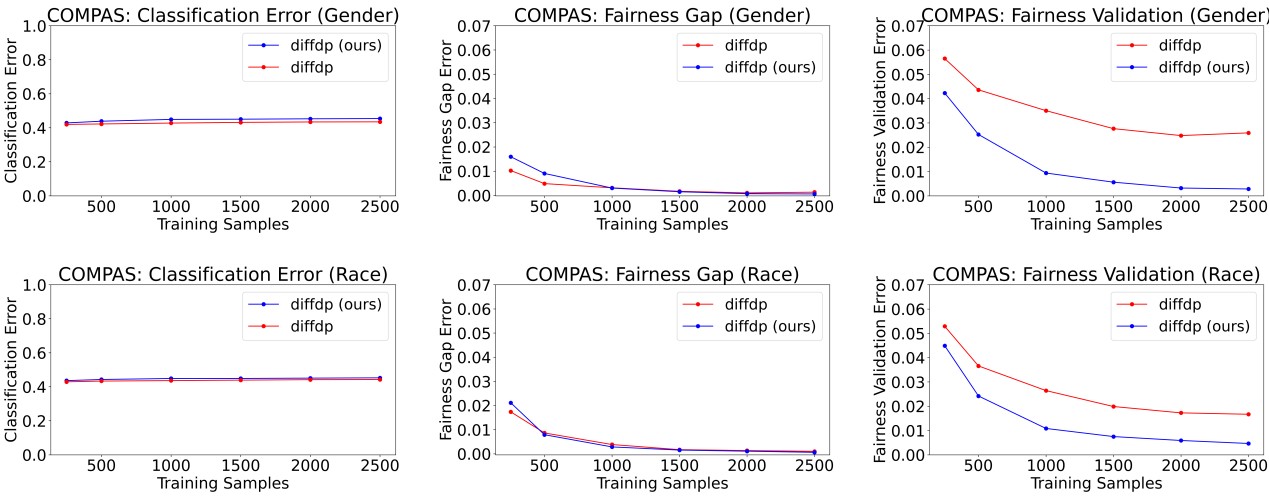

*Figure 8.* Experimental results with COMPAS (gender) dataset of our batch-balancing technique for diffDP as a function of the total number of training samples $n$. We report the mean over $m_1$.

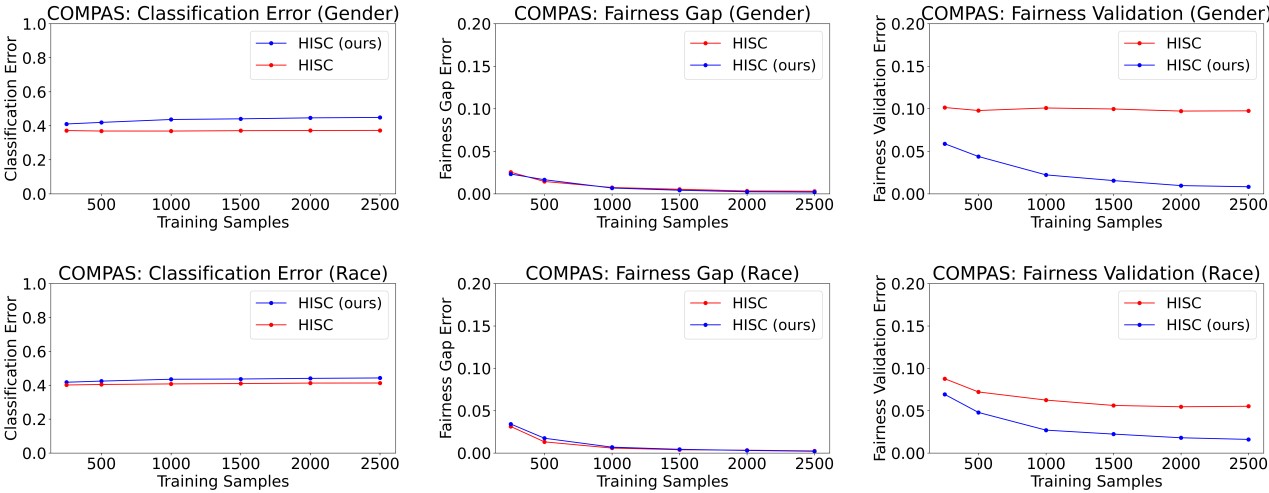

*Figure 9.* Experimental results with COMPAS (gender) dataset of our batch-balancing technique for HISC as a function of the total number of training samples $n$. We report the mean over $m_1$.

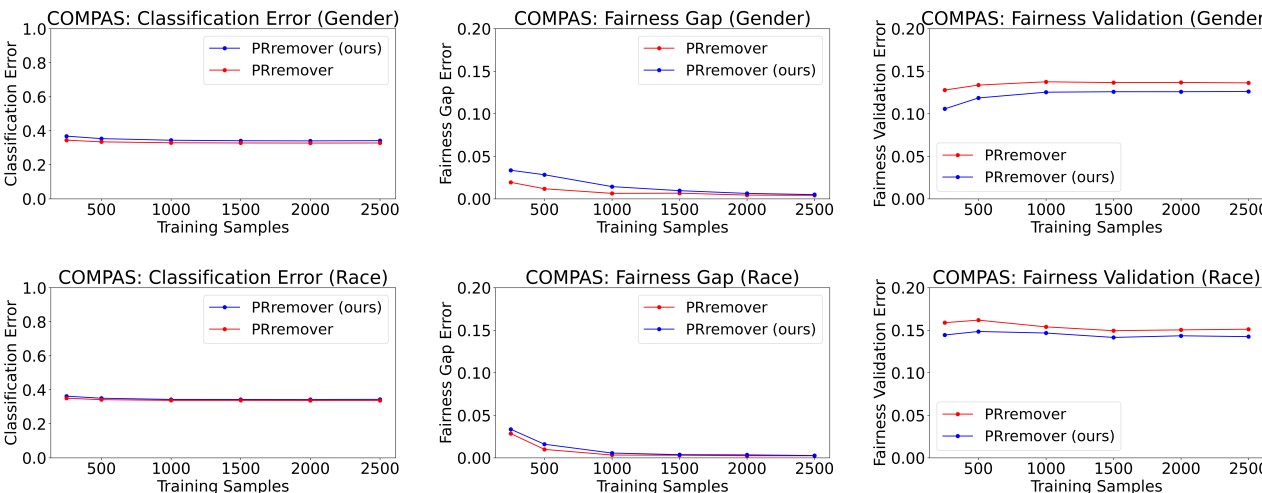

*Figure 10.* Experimental results with COMPAS (gender) dataset of our batch-balancing technique for PRremover as a function of the total number of training samples $n$. We report the mean over $m_1$.

