# OpenReview forum: "Fairness Overfitting in Machine Learning: An Information-Theoretic Perspective"
_ICML.cc/2025/Conference — ICML 2025 poster_

### Official Review · Reviewer_H2i8 · 2025-03-13

**Overall Recommendation:** 3

**Summary:**

This paper proposes new generalization bounds for fair machine learning. Based on the Mutual Information framework, these bounds show that the important factors governing fairness generalization are the size of the different subgroups and the mutual information between the distribution of hypothesis and subsets of the training set on which fairness is evaluated. Tighter versions of the bound based on more involved mesures of mutual information are also derived. Developed for both Demographic Parity and Equalized Odds, the tightness of the proposed bounds is empirically evaluated on several datasets.

**Claims And Evidence:**

The proposed fairness definitions do not seem to match the usual ones in the multiclass case:
- The fairness definition considered in Equation (1) for demographic parity seems to assume binary classes (at least to match the usual demographic parity definition, see the work of Agarwal et al. 2018 for example). However, in the paper, it is assumed that the predictions lie in a range $[0,a]$ which suggests a multi-label predictions setting, it is then not clear what the proposed formula really represents.
- To match the definition of equalized odds, $f$ should be the probability of having a positive label given a sample. While this initially fits the setup proposed in the paper (line 320, 1st column), it is later mentioned that it is possible to extend the proposed approach to the multiclass case by considering $f$ in $[0,a]$ (line 365, second column). It is then not clear what this measure really represents.

**Essential References Not Discussed:**

The related work, relegated to Appendix A, misses several works that addressed the problem of generalization guarantees in fair machine learning, albeit with different proof techniques. A non-exhaustive list:
- Learning Non-Discriminatory Predictors, Woodworth et al., 2017
- A Reductions Approach to Fair Classification, Agarwal et al., 2018
- Randomized Learning and Generalization of Fair and Private Classifiers: from PAC-Bayes to Stability and Differential Privacy, Oneto et al, 2019

**Experimental Designs Or Analyses:**

To improve fairness, the paper proposes, in Section 6, to use a batch balancing approach. This is reminiscing of the work of Roh et al. (FairBatch: Batch Seleection for Model Fairness, 2021) and connections should be discussed.

**Methods And Evaluation Criteria:**

The methods and evaluation criteria appear to be appropriate. This is mainly a theoretical paper and complete proofs are provided.

**Other Comments Or Suggestions:**

- There is a notation mismatch between the main paper and the proof of Lemma 1 ($\ell_E^F$ seems to become $F_E$ for example).
- Line 161, second column, it should be $\tilde{v}_i \neq v_i$.
- In the experiments, the plot at the bottom right of Figure 2 suggests that the actual generalization error is larger than the bound. This is probably due to inverted axis labels.

**Other Strengths And Weaknesses:**

The intuition behind the Mutual Information terms appearing in the different bounds is hard to grasp as there is only very sporadic discussion on what they really capture and when they can be expected to be small or large. From the experiments in Figure 2, they seem rather large since the bound does not converge to $0$ but, instead, seems to reach a plateau when the number of examples exceeds $1500$.

**Questions For Authors:**

1. Is the square-integrability assumption to apply the Efron-Stein inequality respected?

Would this concern be addressed in a satisfactory manner, I could increase my score. Furthermore, would the other reviews or the rebuttal show that I missed or misunderstood some key points, I would reconsider my stance.

**Relation To Broader Scientific Literature:**

Line 191 to 198, it is mentioned that the proposed result is the first to link generalization error to group imbalance. However, this is something that already appears in previous fairness generalization bounds based on the VC dimension where the bound is smallest when all the groups have the same size (for example, see Woodworth et al. 2017 in the list of missing references).

**Theoretical Claims:**

I only skimmed through the proof of Lemma 2 as it is one of the central results of the paper:
- The Efron-Stein inequality (Boucheron et al. 2013, Theorem 3.1) assumes that the function $g$ is square-integrable. However, this assumption is not mentioned in the paper and it is never formally proved that the considered functions respect this assumption.

---

> ### Author Rebuttal · Authors · 2025-03-31
>
> ## Mismatch in Fairness Definitions for the Multiclass Problem
>
>  In our original formulation for the binary case, the prediction function $f =\hat{Y}$ outputs values in $\{0,1\}$ (i.e., a=1). For multiclass problems, we allow $f$ to take values in a bounded range $[0,a]$; however, the choice of $a$ does not affect the upper bound in our fairness guarantees.
>
> To illustrate this in the context of Equalized Odds (EO), consider our use of the Total Variation (TV) loss as a fairness measure. For each true label $y \in \{0,1,2,3\}$, we define the TV loss as follows:
>
> $$ \ell^{F_S}\_E(w,S \mid Y=y) = \frac{1}{2}\sum\_{c=0}^{3}\Bigl|P(\hat{Y}=c\mid Y=y,T=0)-P(\hat{Y}=c\mid Y=y,T=1)\Bigr|, $$
> which in practice is approximated by
> $$ \ell^{F_S}\_E(w,S \mid Y=y) = \frac{1}{2}\sum\_{c=0}^{3}\left|\frac{n\_{0,y,c}}{n\_{0,y}+2}-\frac{n\_{1,y,c}}{n_{1,y}+2}\right|. $$
> We then define an aggregate function over the true labels: $g(z_u) = \sum\_{y\in\{0,1,2,3\}} \ell^{F\_S}\_E(w,z_u \mid Y=y) $. Our analysis shows that
> $$ \sup\_{z\_u,\tilde{z}\_u^i} |g(z\_u)-g(\tilde{z}\_u^i)| \le \frac{2}{\min\_{(t,y)}\{n^{Z\_u}\_{t,y}+2\}}$$
> The key point is that—even if the prediction function $f$ is allowed to take any value in $[0,a]$—changing one sample affects the probability estimates (and hence the TV loss) by at most a fixed amount (i.e., at most 1) regardless of $a$. In other words, while $f$’s output may be scaled by $a$, the impact on the fairness loss (measured in terms of probability differences) remains unchanged.
>
> Thus, our technical arguments extend naturally to the multiclass setting. We acknowledge that our previous explanation did not clearly separate the role of the prediction magnitude (bounded by $a$) from the fairness loss itself, and we appreciate the opportunity to clarify that the fairness definitions (including EO) are applicable to the multiclass case under our bounded prediction assumption.
>
> ## Is the square-integrability assumption to apply the Efron-Stein inequality respected?
> Thank you for carefully reading the proof. You are correct that the Efron-Stein inequality assumes square-integrability, and we acknowledge that this assumption was not explicitly stated in Lemma 2 in the paper. However, in our case, this assumption is naturally satisfied because all the random variables involved are bounded. Specifically, as  $f$ is bounded ($f<a$), the loss function $l$, for which we apply the Efron-Stein inequality through Lemma 2 (Eq. 65  or Eq 82), is also bounded ($l<a$ see Eq. 190-191 ). Since any bounded random variable is square-integrable, this guarantees that the assumption is met. Formally, we have: \
> $$ |l| \leq a. $$
>  To verify square-integrability, we need to show that
> $\mathbb{E}[l^2] < \infty$.
> Since $l$ is bounded. We have
> $$ X^2 \leq a^2$$
> Taking expectations on both sides,
> $$ \mathbb{E}[l^2] \leq \mathbb{E}[a^2] = a^2 < \infty. $$
> Thus, our loss is always square-integrable. Therefore, the conditions for applying the Efron-Stein inequality are fully satisfied in our setting.
>
> We will include this clarification and add the assumption explicitly to Lemma 2 in the revised version of the paper to ensure completeness.  Given that this addresses your concern, we kindly ask you to reconsider your score. We appreciate your openness to revisiting your evaluation and would be happy to discuss any further points if needed.
>
> ## Related work:
>
> Thank you for highlighting the related work on generalization guarantees in fair machine learning. We will address these references in the revision.
> However, while (Woodworth et al., 2017; Agarwal et al., 2018) derive generalization guarantees for loss functions corresponding to DP and EO within specific algorithms, our work targets a more general algorithmic framework in the DP and EO setting—even when using other loss functions. Moreover, their guarantees primarily focus on overall sample complexity, whereas our bounds incorporate not only the total sample size but also additional factors such as the properties of the learning algorithm, the particular loss function, the dataset characteristics, and, importantly, the group balance in the dataset.
>
> More closely related work is Oneto et al. (2019), which derives a generalization bound in Theorem 1 for fairness generalization with randomized algorithms. However, their bound, being a KL-divergence bound, can not be computed and evaluated in practice for any realistic setting. In contrast, our bounds—particularly Theorem 5—are computable, even for modern deep neural networks. This ensures that our results are not only theoretically rigorous but also practical and computable, allowing us to study fairness generalization errors in real-world settings.
> We will incorporate this discussion into the paper and revise our claim accordingly.
>
> Thank you for catching the typos. We will fix all these typos and the flipped axis labels iof the Figure in the final version.

---

> > ### Comment · Reviewer_H2i8 · 2025-04-03
> >
> > Thanks for this rebuttal.
> >
> > Unfortunately, I could not follow the explanation regarding the mismatch between the fairness definition studied in this paper and the standard definitions. More precisely, the loss discussed in the rebuttal seems different to the one present in the paper. Furthermore, my concern is not whether the proposed upper-bounds apply in the multi-class setting but rather what is the connection between the proposed fairness metrics and the ones that can be found in the literature in the multi-class case. They appear to be different.
> >
> > The rebuttal addressed my concern regarding the missing assumption.
> >
> > In light of this, I increased my score to 2.

---

> > > ### Author Response · Authors · 2025-04-04
> > >
> > > Thank you for the follow-up and for pointing out the confusion. We appreciate the opportunity to clarify the distinction.
> > >
> > > We would like first to emphasize that in this paper, we focus mainly on the binary case (i.e., a=1 throughout the paper. we will make that clear in the final version). In the binary setting, we focus on two widely adopted notions: demographic parity (DP) and equalized odds (EO), which correspond to the independence and separation criteria in fairness. We chose to focus on these definitions due to their simplicity, widespread adoption, and analytical tractability in the binary setting [1,2].
> > >
> > > For instance, demographic parity is typically expressed as
> > > $ P(\hat{Y} \mid T = 0) = P(\hat{Y} \mid T = 1) $,
> > > and equalized odds is given by
> > > $ P(\hat{Y} = 1 \mid T = 1, Y = y) = P(\hat{Y} = 0 \mid T = 0, Y = y) $
> > > for $ y \in$ {0, 1}. Although the notation in our paper may appear different at first glance, our fairness losses are mathematically equivalent to these standard definitions, as used in prior work (e.g., [2,6]). This equivalence is what enables our tractable theoretical analysis.
> > >
> > >
> > > Regarding the multi-class extension, extending notions like DP or EO beyond binary outcomes is known to be nontrivial, and there is no universally accepted generalization in the literature [2-5]. Some works rely on information-theoretic quantities such as mutual information [3], others on distance correlation [4], or direct extension based on the binary formulation [5] above.
> > >
> > > In our rebuttal, we presented a multi-class example to illustrate the flexibility of our theoretical tools. The fairness criterion used there aligns with Definition 1 in [5]. Our goal was to show that the theoretical framework developed in the main paper can easily extend beyond the binary case and apply to other fairness notions.
> > >
> > >
> > > To conclude, while our paper primarily focuses on the binary case and provides a detailed theoretical analysis with a novel bounding technique, the flexibility of our framework offers significant value. Our tools can easily be extended to address other fairness definitions in more complex settings, including multi-class cases (as demonstrated in the rebuttal), thus opening avenues for future work. The key contribution remains the new bounding technique and the versatility of the theoretical tools developed, which have the potential to facilitate further progress not only in fairness setting but also in group-based loss settings.
> > >
> > > We hope that we have addressed your concerns and we kindly ask you to reconsider your score.
> > >
> > > [1] Han, Xiaotian, et al. FFB: A fair fairness benchmark for in-processing group fairness methods. In International Conference on Learning Representations, 2024.
> > >
> > > [2] Mroueh, Y. et al. Fair mixup: Fairness via interpolation. In International Conference on Learning Representations,2021.
> > >
> > > [3] Umang Gupta, Aaron M Ferber, Bistra Dilkina, and Greg Ver Steeg. Controllable guarantees for fair outcomes via contrastive information estimation. In Proceedings of the AAAI Conference on Artificial Intelligence, volume 35, pages 7610–7619, 2021.
> > >
> > > [4] Dandan Guo, Chaojie Wang, Baoxiang Wang, and Hongyuan Zha. Learning Fair Representations via Distance Correlation Minimization. IEEE Transactions on Neural Networks and Learning Systems, pages 1–14, 2022.
> > >
> > > [5] Denis et al. Fairness guarantee in multi-class classification. arXiv:2109.13642, 2023.
> > >
> > > [6] Madras, David, et al. Learning adversarially fair and transferable representations. International Conference on Machine Learning. 2018.

---

### Official Review · Reviewer_p9yH · 2025-03-15

**Overall Recommendation:** 4

**Summary:**

The paper considers the generalization of (in terms of empirical violation of) fairness when presented with unseen data. Specifically, the goal is to provide a formal guarantee through information-theoretic fairness generalization bounds with mutual information (MI) and conditional mutual information (CMI). The theoretical and empirical results are provided, and the tightness and practical relevance of the bounds are analyzed across several (group-level) fairness notions, including DP and EOdds.

---

### After Rebuttal

Thank authors for the rebuttal. The response is helpful and to-the-point, which further boosts my confidence in evaluation of an Accept (score 4). I confirm that I also carefully went through comments from other reviewers, as well as the rebuttals therein.

**Claims And Evidence:**

The claims consists of several theoretical bounds, e.g., the relation of fairness generalization and the dependence between hypothesis and input data (Theorem 1), the relation between fairness overfitting and the selection mechanism (Theorem 2), the tightening of presented bounds (Theorems 3--4), the reduction of computational cost (Theorem 5), and the bounds for specific fairness notions (Theorems 6--7). The evidence includes the proofs of theorems and the empirical evaluations.

**Essential References Not Discussed:**

There are no significant missing references.

**Experimental Designs Or Analyses:**

The experimental analyses include different aspects of the evaluation of bounds, including bound tightness, bound-error correlation, and the implication of batch balancing (during training).

**Methods And Evaluation Criteria:**

The method starts from theoretically capturing the bounds followed by improving them, and the further applying to specific group-level fairness notions. The evaluation criteria include the derived $\Delta L$-CMI, with different methods including DiffDP, DiffEodd, DiffEopp, HSIC, PRemover, etc.

**Other Comments Or Suggestions:**

Nothing specific in addition to the comments in above sections.

**Other Strengths And Weaknesses:**

The strength of the paper comes from the organization of materials and the clear presentation of motivation, setting, theoretical analyses, and empirical evaluations.

The paper can be further improved by including some discussion from the side of lower bounds. While I understand that providing a lower bound might involve developing another set of theoretical results and is beyond the scope of current work, having some discussion (e.g., why a lower bound might be nontrivial, whether a lower bound can be shown to be above 0) can shed light on the whole picture and make it even more informative, i.e., the generalization bounds for fairness can be from both sides (instead of only gets upper bounded).

**Questions For Authors:**

Since the impossibility results in previous literature have shown that, in general, not all group-level fairness notions can be achieved at the same time (Chouldechova, 2017, Kleinberg et al., 2017), and also that, the group-level fairness notion EOdds may not be attainable if the data distribution itself does not satisfy certain properties (Tang and Zhang, 2022). I am curious about the possibility of directly deriving the lower bound with the proposed theoretical analyzing framework.

---

Alexandra Chouldechova. 2017. Fair prediction with disparate impact: A study of bias in recidivism prediction instruments. _Big Data_ 5, 2 (2017), 153–163.

Jon Kleinberg, Sendhil Mullainathan, and Manish Raghavan. 2017. Inherent trade-offs in the fair determination of risk scores. In _Proceedings of the 8th Innovations in Theoretical Computer Science Conference (ITCS’17)_.

Zeyu Tang and Kun Zhang. 2022. Attainability and optimality: The equalized odds fairness revisited. In _Proceedings of the Conference on Causal Learning and Reasoning_, Vol. 177. PMLR, 754–786.

**Relation To Broader Scientific Literature:**

The fairness overfitting and generalization bounds can have implications over broader fields where relevant group-level fairness notions are of interest.

**Theoretical Claims:**

The theoretical claims are different generalization bounds for fairness, and the proofs are provided in appendix.

---

> ### Author Rebuttal · Authors · 2025-03-31
>
> We appreciate your thoughtful question about deriving lower bounds within our theoretical framework, especially considering existing challenges in achieving group-level fairness notions like Equalized Odds (EOdds).
> In our work, we focus on understanding how fairness measures observed during training generalize to new, unseen data. While our framework sheds light on the behavior of fairness interventions and their generalization properties, directly deriving explicit lower bounds for the fairness-accuracy trade-off, particularly for specific notions like EOdds, is fundamentally different. In particular, Information-theoretic bounds, including ours, rely on the Donsker-Varadhan variational representation for upper-bound generalization error. Hence, deriving a lower bound would require an alternative variational formulation that directly lower bounds the difference in expectation between the joint distribution and the product of marginals of the specific loss function, which is not trivial.
>
> Regarding the impossibility results, as you've noted, prior research has shown that it's often impossible to satisfy all group-level fairness criteria simultaneously. For instance, Chouldechova (2017) and Kleinberg et al. (2017) discuss inherent trade-offs between different fairness measures. Additionally, Tang and Zhang (2022) highlight that achieving EOdds depends on specific data distribution properties; when these aren't met, deterministic classifiers may struggle to attain EOdds without incorporating randomness into their predictions. The key distinction is that impossibility results establish lower bounds on the population risk for different fairness loss functions, while our contribution provides upper bounds on fairness generalization error. Notably, a model can have low generalization error but still perform poorly in terms of fairness on the population level.
>
> An interesting parallel future direction would be to explore connections between different fairness generalization bounds—for instance, relating DP-fairness generalization error to EO-fairness error. We will incorporate this discussion in the future directions section.

---

### Official Review · Reviewer_mzyB · 2025-03-24

**Overall Recommendation:** 3

**Summary:**

This paper studies fairness generalization error, i.e., how does model fairness extend to new, unseen data. The fairness generalization error is defined (in Eq. (3)) as the discrepancy between the fairness-population risk and the fairness-empirical risk. The authors study this from an information theory perspective, and derive upper bounds for fairness generalization error using two widely used fairness metrics: Demographic Parity (DP) and Equalized Odds (EOd). The proposed fairness error bounds are validated through experiments on two datasets: COMPAS and Adult.

**Claims And Evidence:**

My first question about the paper is in terms of its motivation. The authors illustrated the fairness error in training and testing on COMPAS dataset in Figure 1, but the difference between training and testing error doesn't seem significant, even for ERM training. The authors show more COMPAS results in Figure 2, and in the appendix (which includes more results on COMPAS and Adult dataset), where the fairness generalization error (difference between training and testing fairness error) is not significant.

This raises the question of whether fairness overfitting is a significant issue, at least on the two datasets (COMPAS and Adult) presented. If the fairness generalization error remains small, the motivation for a theoretical study on fairness generalization error bounds may be weaker. Could the authors clarify why such an analysis is necessary given these findings?

**Essential References Not Discussed:**

See in *Theoretical Claims*

**Experimental Designs Or Analyses:**

I understand that the paper primarily focuses on a theoretical study, but the experimental analysis is limited to only two datasets (COMPAS and Adult). Actually, the main paper presents results only on COMPAS, while all Adult dataset results are relegated to the appendix. To better demonstrate the practical applicability of the proposed theory, the authors may consider including a few larger datasets (like CelebA) and varying model architectures in their analysis.

**Methods And Evaluation Criteria:**

The paper focuses on a theoretical study, but it is not quite clear how the proposed theory can be applied to practical estimation. To be specific:

The paper focuses on a theoretical study, but it is not quite clear how the proposed theory can be used in practice. Specifically:

1. the description of the learning algorithm \mathcal{A} and hypothesis W seems vague to me. Could the authors clarify line 108 (left column) - line 072 (right column)? Are there any assumptions or constraints on the learning algorithm AA?

2. The assumption that $|f| \in [0,a]$ is used in many theorems. Could the authors clarify how to determine the value of a in practice for estimating the upper bound?

3. The derived error bounds rely on using different permutations of a subset of training data of size mm. However, there is no clear guideline on how to choose mm in practice. Could the authors elaborate on the discussion of mm in Remark 5?

4. Could the authors discuss the time complexity of computing the bound estimation? Does obtaining the bound require re-training the model for each new permutation?

5. From Figure 2 and more figures in the appendix, the derived bound seems to be the upper bound of the fairness generalization error. However, it's not clear if the derived upper bound is indeed tight. Could the authors elaborate more on how to interpret the experimental results? Or any evidence to support that the bound is tight?

**Other Comments Or Suggestions:**

1. In Eq. (4), is it gen_fairness or \bar gen_fairness?

2. Lemma 2: "such that ... for all j \not = i except on i, i.e., \tilde v_i \not = v_j". Is it \tilde v_i \not = v_i instead?

3. Theorem 1: the second square symbol seems not clear

4. In Figure 2, second row, third column, the authors may consider reporting the correlation coefficient and p-value to validate the linear relationship

**Other Strengths And Weaknesses:**

None

**Questions For Authors:**

My major concerns regarding the paper are about the following:

1. motivation to study fair generalization error bound

2. applicability, parameter setting, efficiency, and validity of using the derived theorems for error bound estimation

3. practical insights from the theory

4. experimental validation of the derived bound

Please check my comments in above sections of Claims And Evidence, Methods And Evaluation Criteria, Theoretical Claims, Experimental Designs Or Analyses for details.

**Relation To Broader Scientific Literature:**

This paper is related to fairness in machine learning, an important topic in trustworthy machine learning.

**Theoretical Claims:**

This paper proposes to derive the upper bound of fairness generalization error. However, it's not quite clear what new insights can be obtained from the theoretical study. For example, in Section 6, "Batch Balancing", the authors mention that balancing training data between different sensitive groups can help improve fairness, but this conclusion has been studied in many existing works, which use different techniques like resampling, reweighting, counter-factual data generation to balance data [1,2,3]. It's not clear what new insights can be obtained from the theoretical analysis.

Building on the points mentioned above: Based on Theorem 5 and 7, it seems that the mutual information term and the number of samples affect the upper bound. The authors may consider exploring potential new/practical insights from these two perspectives. For example, can we practically reduce the mutual information term (maybe through some regularization in training) or increase the training sample number (like data augmentation) to minimize the fairness error bound?

[1] Buda, Mateusz, Atsuto Maki, and Maciej A. Mazurowski. "A systematic study of the class imbalance problem in convolutional neural networks." Neural networks 106 (2018): 249-259.

[2] Jang, Taeuk, Feng Zheng, and Xiaoqian Wang. "Constructing a fair classifier with generated fair data." Proceedings of the AAAI Conference on Artificial Intelligence. Vol. 35. No. 9. 2021.

[3] Sagawa, Shiori, et al. "Distributionally Robust Neural Networks." International Conference on Learning Representations. 2019.

---

> ### Author Rebuttal · Authors · 2025-03-31
>
> We sincerely appreciate the reviewer’s thoughtful feedback on our work. Below, we address each concern point by point.
>
> **Motivation:** We respectfully disagree that there is a lack of motivation to study fairness generalization. As demonstrated in Figure 1, i) Compared to ERM, when fairness interventions are applied, the generalization error can become particularly significant (e.g., HSCI or PRemover), indicating that these techniques introduce generalization challenges that need further investigation. ii) In certain cases—especially with low-data— the fairness generalization error can be as high as 10–30% of the fairness training loss, which is significant. iii) Different fairness methods exhibit varying generalization behaviors, suggesting that multiple factors influence fairness generalization. Understanding these variations is essential for developing principled approaches to mitigating fairness-related overfitting.
>
> **Experiments:** We kindly refer the reviewer to lines 397-404. Estimating our proposed bound follows the well-established protocol (Harutyunyan et al. 2021; Wang & Mao 2023) and particularly Dong et al. (2024), where permutation-based bounds have been proposed for standard generalization error. The main difference is that here: i) we are targeting a fairness loss. ii) MI terms involve a continuous variable.
>
> Q:  $a$ and $m$ in practice? \
> A: $a$ in our paper is the upper bound of the predictor function f. Since in this paper, we consider mainly the binary case {-1,1}, a=1 and does not require any estimation. We will clarify this in the final version. \
> $m$ is a hyperparameter. Like in our experiments, we recommend setting  m=n−1 for practical reasons. ​\
> Q: Re-training model for each permutation? \
> A: No, computing the bound does not require retraining for each permutation. It only involves evaluating mutual information estimates over different subsets, with a computational complexity similar to Dong et al. (2024).
>
> **Insights from theory:**\
> **Theoretical:** In this paper, we take a significant step toward addressing fairness generalization errors by deriving the first rigorous **computable bounds** in the MI and CMI settings, using a novel bounding technique. Our results demonstrate that models generalize differently depending on the fairness approach used, as they memorize the different data attributes (V) at different scales. Furthermore, our bounds shed some light on how data balance influences fairness overfitting. \
> **Practical:** Beyond these theoretical insights, generalization bounds can also inform practical strategies for improving fairness performance. As a proof of concept, our results suggest that balanced representation (1/H(n_0,n_1)) reduces fairness generalization errors. To validate this, we experimented with batch-balancing, demonstrating that this indeed improves generalization further confirming our theoretical finding. Note that from this perspective, our work can be seen as a theoretical guarantee for the previous works on data balancing highlighted by the reviewers. Additionally, for example, our findings suggest that controlling the MI term—such as by introducing gradient noise during training—could further improve fairness generalization.
>
> Q: assumptions on A? \
> A: A learning algorithm is a randomized mapping from the training data $S$ to the model weights $W$. Similar to other information-theoretic studies (e.g., Harutyunyan et al., 2021), we make no assumptions about A.
>
> Q: How to interpret the experimental results? the bound is tight? \
> A: In general, the primary goal of generalization bounds is not solely tightness—achieving this requires strong additional assumptions—but rather to provide theoretical insights into generalization behavior and capture the key factors. Information-theoretic bounds are generally among the tightest in learning theory. For instance, Wang & Mao (2023) (in Figure 2) report bounds that, while sometimes over four times larger than the observed empirical generalization error, remain valid and notably tighter than previous results, as they successfully capture the overall trends. \
> In the context of fairness, using prior bounding techniques would yield overly loose bounds that fail to provide meaningful insights into fairness dynamics (see our discussion on Lemma 1). In contrast, our work introduces a novel bounding technique that not only results in tighter fairness bounds compared to previous techniques—e.g., Theorem 1 is tighter than Lemma 1 by a factor of $1/\sqrt{n}$​—but also reveals fairness-specific properties, such as the influence of class balance (n0,n1​). Empirically, our derived bounds consistently track the observed fairness generalization error across different models and datasets, further validating their effectiveness.
>
> Q: p-value? \
> A: The p-value for the scatter plot is 3.43×10⁻¹⁶, indicating an extremely strong statistical significance further validating our bounds.
>
> We will fix the typos suggested by the reviewer.

---

### Decision · Program_Chairs · 2025-05-01

**Decision:**

Accept (poster)

**Comment:**

It seems like the reviewers are on the positive side on this paper and agree that the paper has merits and is worth being accepted at ICML. The rebuttal definitely seemed to help clarify things for the reviewers. Please incorporate the clarifications into the final version of the paper.    Also please be sure to effectively and clearly clarify in the final paper, the details regarding the clarifications raised by reviewer H2i8